# Diverse Influence Component Analysis: A Geometric Approach to Nonlinear Mixture Identifiability

**Hoang-Son Nguyen** and **Xiao Fu**

School of Electrical Engineering and Computer Science
Oregon State University
{nguyhoa3, xiao.fu}@oregonstate.edu

## Abstract

Latent component identification from unknown *nonlinear* mixtures is a foundational challenge in machine learning, with applications in tasks such as self-supervised learning and causal representation learning. Prior work in *nonlinear independent component analysis* (nICA) has shown that auxiliary signals—such as weak supervision—can support *identifiability* of conditionally independent latent components. More recent approaches explore structural assumptions, e.g., sparsity in the Jacobian of the mixing function, to relax such requirements. In this work, we introduce *Diverse Influence Component Analysis* (DICA), a framework that exploits the convex geometry of the mixing function's Jacobian. We propose a *Jacobian Volume Maximization* (J-VolMax) criterion, which enables latent component identification by encouraging diversity in their influence on the observed variables. Under reasonable conditions, this approach achieves identifiability without relying on auxiliary information, latent component independence, or Jacobian sparsity assumptions. These results extend the scope of identifiability analysis and offer a complementary perspective to existing methods.

## 1 Introduction

*Nonlinear mixture model identification* (NMMI) seeks to uncover latent components transformed by *unknown* nonlinear functions. A typical data model of interest in the context of NMMI is as follows:

$$\boldsymbol{x} = \boldsymbol{f}(\boldsymbol{s}), \; \boldsymbol{s} \in \mathbb{R}^d, \; \boldsymbol{x} \in \mathbb{R}^m, \tag{1}$$

where $\boldsymbol{s} = (s_1, \ldots, s_d) \sim p(\boldsymbol{s})$ is a random vector following a certain distribution $p(\boldsymbol{s})$ with support $\mathcal{S}$, $\boldsymbol{f} : \mathbb{R}^d \to \mathbb{R}^m$ is an unknown, nonlinear mixing function, $\boldsymbol{x} = (x_1, \ldots, x_m) \in \mathbb{R}^m$ is the observed data, and $s_i$ for $i \in [d]$ and $x_j$ for $j \in [m]$ represent the $i$th latent component and the $j$th observed feature, respectively. The function $\boldsymbol{f}$ is modeled a diffeomorphism that maps a latent variable to a $d$-dimensional Riemannian manifold $\mathcal{X}$ embedded in $\mathbb{R}^m$, where $m \geq d$ [1–3]. Given the observations (samples) of $\boldsymbol{x}$, NMMI amounts to recovering $\boldsymbol{s}$ and $\boldsymbol{f}$ up to certain acceptable or inconsequential ambiguities. NMMI and variants play a fundamental role in understanding many machine learning tasks, e.g., latent disentanglement [1, 4], causal representation learning [5, 6], object-centric learning [2], and self-supervised learning [7, 8].

The NMMI task is clearly ill-posed—in general, an infinite number of different $(\boldsymbol{f}, \boldsymbol{s})$ could be found from observations of $\boldsymbol{x}$ under (1). Hence, establishing *identifiability* of $\boldsymbol{f}$ and $\boldsymbol{s}$ becomes a central topic in NMMI. This identifiability challenge was extensively studied under the umbrella of *nonlinear independent component analysis* (nICA) [9–12]. A key take-home point is that nICA poses a much more challenging identification problem relative to ICA (where $\boldsymbol{f}$ is a linear system). That is, even if $s_1, \ldots, s_d$ are statistically independent, the model in (1) is not identifiable [9].

39th Conference on Neural Information Processing Systems (NeurIPS 2025).

In recent years, much progress has been made in understanding identifiability of (1). The line of work [1, 10–12] showed that if $s_1, \ldots, s_d$ are *conditionally* independent given a certain auxiliary variable $\boldsymbol{u}$ (which can be understood as additional side information), then $\boldsymbol{f}$ and $\boldsymbol{s}$ can be recovered to reasonable extents. Another line of work tackles identifiability by assuming that the mixing function $\boldsymbol{f}$ is *structured* other than completely unknown. For example, the works [13–16] make an explicit structural assumption that $\boldsymbol{f}$ is a post-nonlinear mixing function, the work [17] assumes that $\boldsymbol{f}$ is conformal, and the more recent work [18] assumes that $\boldsymbol{f}$ is piecewise linear. Using these structures, the auxiliary information can be circumvented for establishing identifiability.

More recent advances propose to exploit *implicit* structures (other than *explicit* structures like post-nonlinearity) of $\boldsymbol{f}$. Notably, the works [3, 19, 20] utilize structures defined over the Jacobian of $\boldsymbol{f}$ for identifying the model (1). In particular, it was shown that as long as the Jacobian of $\boldsymbol{f}$ follows certain sparsity patterns, then the model (1) is identifiable [2, 3, 21]. Imposing structural constraints on the Jacobian of $\boldsymbol{f}$ is arguably less restrictive compared to assuming explicit parameterization of $\boldsymbol{f}$. Structures of Jacobian reflect how the latent variables $s_1, \ldots, s_d$ affect the observed features $x_1, \ldots, x_m$, making the related assumptions admit intuitive physical meaning. Using structured Jacobian to model such influences is also advocated by several causal representation learning paradigms (see, e.g., *independent mechanism analysis* (IMA) [20] under the *independent causal mechanism* (ICM) principle [22]), and other frameworks, e.g, object-centric representation learning [2, 23, 24].

**Open Question.** The advancements have been encouraging, yet understanding to NMMI identifiability remains to be deepened. In particular, the Jacobian sparsity-based approaches, e.g., [2, 3, 19, 21], provided intriguing ways of establishing identifiability of the model in (1)—without using auxiliary variables, latent component independence, or relatively restrictive structural constraints of $\boldsymbol{f}$. However, the Jacobian sparsity assumptions in these works essentially assume that the observed features are only generated from a subset of $s_1, \ldots, s_d$, which sometimes may not hold. It is tempting to circumvent using such strict sparsity-based assumptions in NMMI.

**Contributions.** In this work, we propose to make use of an alternative condition of $\boldsymbol{f}$'s Jacobian for NMMI. We leverage the fact that, as long as the influences of $\boldsymbol{s}$ imposed on each $x_j$ for $j = 1, \ldots, m$ are *sufficiently diverse*, $\boldsymbol{f}$'s Jacobian exhibits an interesting convex geometry. This geometry is similar to the classical "*sufficiently scattered condition* (SSC)" in the *structured matrix factorization* (SMF) literature [25–29] and, particularly, the more recent developments in *polytopic matrix factorization* (PMF) [30] (also see [31–33]). As a consequence, taking intuition from volume-regularized SMF [25, 30–36], we show that fitting the data model in (1) together with maximizing the learned $\boldsymbol{f}$'s Jacobian volume *provably* recovers $\boldsymbol{f}$ and $\boldsymbol{s}$ up to acceptable ambiguities. The proposed *Jacobian volume maximization* (J-VolMax) approach does not rely on auxiliary variables or statistical independence. More notably, an $\boldsymbol{f}$ with a *dense* Jacobian can still be provably identified under our formulation. These advantages make our identifiability results applicable to a wide range of scenarios that are not covered by the existing literature, substantially expanding the understanding of model identifiability under (1). We tested the proposed approach on synthetic data and in a single-cell transcriptomics application. The results corroborate with our NMMI theory.

**Notation.** Please refer to Appendix A.1 for details.

## 2 Background

**From ICA to nICA.** ICA is arguably one of the most influential latent component identification approaches. Classic ICA [37, 38] assumes that $\boldsymbol{x} = \boldsymbol{A}\boldsymbol{s}$ with a nonsingular $\boldsymbol{A}$ and that

$$p(\boldsymbol{s}) = \prod_{i=1}^{d} p_i(s_i), \tag{2}$$

where at most one $s_i$ is a Gaussian variable. Then, the identifiability of $(\boldsymbol{A}, \boldsymbol{s})$ up to permutation and scaling ambiguities can be established by finding a linear filter to output independent estimates. Unfortunately, the nICA model, i.e., (1) with condition (2), is in general non-identifiable [9].

**nICA with Auxiliary Information.** More recent breakthroughs propose to use the following conditional independence model, i.e.,

$$p(\boldsymbol{s}|\boldsymbol{u}) = \prod_{i=1}^{d} p_i(s_i|\boldsymbol{u}), \tag{3}$$

to establish identifiability of (1). The side information $\boldsymbol{u}$ can be time frame labels [10, 11, 21, 39, 40], observation group indices [41], or view indices [42]. The takeaway from this line of work is that, using

the *variations* of $\boldsymbol{u}$, one can fend against negative effects (e.g., the existence of measure-preserving automorphism (MPA) [6, 43]) leading to non-identifiability under (1). The results are elegant and inspiring. Nonetheless, both $\boldsymbol{u}$ and conditional independence might not always be available.

**nICA with Structured $\boldsymbol{f}$.** Another route to establish identifiability is to exploit prior structural information of $\boldsymbol{f}$. For example, the works [9, 16, 18, 23, 44–48] used conformal, local isometry, close-to-linear, post-nonlinear, piecewise affine structures, and additive structures of $\boldsymbol{f}$, respectively. Recent works, e.g., [14, 15, 18, 23, 48], also showed that some of these structures can be used for dependent component identification. Nonetheless, such *explicit* structures of $\boldsymbol{f}$, e.g., post-nonlinear mixtures, only make sense when they are used for suitable applications (e.g., hyperspectral imaging [15] and audio separation [16]), yet most problems in generative model learning and representation learning may not have such structures.

**Exploiting Jacobian Structures.** Instead of imposing explicit structures directly on $\boldsymbol{f}$, it is also plausible to exploit the structures of the Jacobian of $\boldsymbol{f}$. Note that $[\boldsymbol{J}_{\boldsymbol{f}}(\boldsymbol{s})]_{i,j} = \partial x_i / \partial s_j$ characterizes how $x_i$ is influenced by the change of $s_j$.

The aforementioned IMA approach [20], inspired by the principle of ICM [22], assumes orthogonal columns of $\boldsymbol{J}_{\boldsymbol{f}}(\boldsymbol{s})$ at each point $\boldsymbol{s} \in \mathcal{S}$. But these developments lack comprehensive identifiability characterizations (see [44]). On the other hand, the works [2, 3, 19, 49] assume that $\boldsymbol{J}_{\boldsymbol{f}}(\boldsymbol{s})$ exhibits a certain type of sparsity pattern. These works formulate the NMMI problem as Jacobian sparsity-regularized data fitting problems. For example, the work [2] proposed the following:

$$\min_{\boldsymbol{f}, \boldsymbol{g}} \ \mathbb{E}_{\boldsymbol{x}}\big[\|\boldsymbol{f}(\boldsymbol{g}(\boldsymbol{x})) - \boldsymbol{x}\|_2^2 + \lambda_{\mathrm{sp}} c_{\mathrm{sp}}(\boldsymbol{J}_{\boldsymbol{f}}(\boldsymbol{g}(\boldsymbol{x})))\big], \tag{4}$$

where the first term finds a diffeomorphism $\boldsymbol{f}$ and its "inverse" $\boldsymbol{g}$ (in which $\boldsymbol{g}(\boldsymbol{x})$ is supposed to recover $\boldsymbol{s}$), and the second term $c_{\mathrm{sp}}(\cdot)$ promotes sparsity of $\boldsymbol{J}_{\boldsymbol{f}}$—see [2, 3, 19, 49, 50] for their respective ways of sparsity promotion. The most notable feature of this line of work lies in its relatively relaxed conditions on $\boldsymbol{s}$ for establishing identifiability of the model. For instance, the work [2] showed that identifiability can be established without using auxiliary variables or statistical independence of $\boldsymbol{s}$. On the other hand, $\boldsymbol{J}_{\boldsymbol{f}}$ being sparse means that the observed features in $\boldsymbol{x}$ are only generated by subsets of $\boldsymbol{s}$. This assumption is justifiable in some applications, e.g., object-centric image generation [2, 23, 24], but can be violated in other settings.

## 3 Proposed Approach: Diverse Influence Component Analysis

We present an alternative way to establish identifiability of the nonlinear mixture model in (1), without relying on statistical assumptions of $\boldsymbol{s}$, sparsity of $\boldsymbol{J}_{\boldsymbol{f}}$, or auxiliary information. Our approach is built upon *convex geometry* of $\boldsymbol{J}_{\boldsymbol{f}}(\boldsymbol{s})$ and an underlying connection between NMMI and SMF models [25–29], particularly the recent advancements in [30] and variants in [31–33].

**Preliminaries of Convex Geometry.** In Appendix A.2, we give a brief introduction to the notions (e.g., convex hull, polar set, *maximal volume inscribed ellipsoid* (MVIE)) that we use in our context.

### 3.1 Sufficiently Diverse Influence

**Motivation.** We are interested in understanding how the influences of $\boldsymbol{s}$ on $\boldsymbol{x}$ affect the identifiability of (1), beginning with a close examination of existing Jacobian assumptions. First, the sparsity assumptions on $\boldsymbol{J}_{\boldsymbol{f}}$ in [2, 3, 19, 49–51] embody key principles in generative modeling and causal representation learning. For instance, the ICM principle [22] promotes generative models where latent variables exert distinct influences on observed features [20]. Sparse Jacobian models share this view when the sparsity patterns of $[\boldsymbol{J}_{\boldsymbol{f}}]_{i,:}$ for $i = 1, \ldots, m$ (or $[\boldsymbol{J}_{\boldsymbol{f}}]_{j,:}$ for $j = 1, \ldots, d$) differ sufficiently. However, sparsity is arguably a relatively stringent assumption—the hard constraint that each $x_i$ depends only on a subset of $s_1, \ldots, s_d$ may not always hold in practice. Second, the IMA method [52] reflects the ICM principle differently: by enforcing column orthogonality in $\boldsymbol{J}_{\boldsymbol{f}}$, it assumes that the influences of $s_i$ and $s_j$ on the change of $\boldsymbol{x}$ are mutually perpendicular. This circumvents sparsity-based constraints and permits all $x_i$ to depend on all latent components. Nonetheless, IMA only guarantees local identifiability of (1) (see [44]), while global identifiability remains unresolved. These observations motivate us to develop an alternative framework that flexibly models diverse interactions between $\boldsymbol{s}$ and $\boldsymbol{x}$, while ensuring global identifiability of nonlinear mixtures.

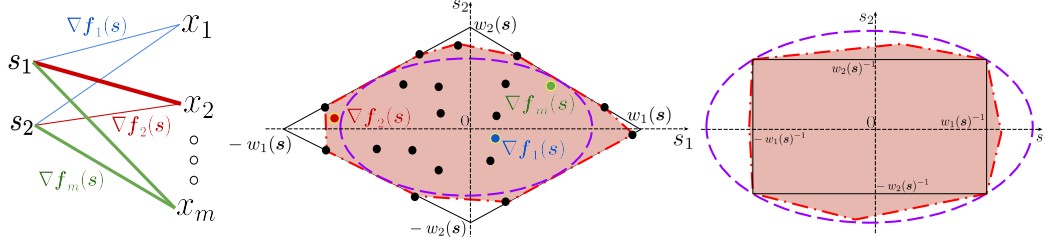

Figure 1: **[Left]** $s$, $x$, and $\nabla f_i(s) \in \mathbb{R}^d$ for $d = 2$, $\forall i \in [m]$; line thickness indicates the magnitude of influence of $s_i$ on $x_j$. **[Middle]** Condition 1 in Assumption 3.1 for $d = 2$: axes represent $\partial f_i/\partial s_1$ and $\partial f_i/\partial s_2 \in \mathbb{R}$; the pink region is $\mathrm{conv}\{\nabla f_1(s), \dots, \nabla f_m(s)\}$, the dashed purple ellipse is $\mathcal{E}(\mathcal{B}_1^{w(s)})$, and the solid black diamond is $\mathcal{B}_1^{w(s)}$. **[Right]** Condition 2 in Assumption 3.1: shaded region shows $\mathrm{conv}\{\nabla f_1(s), \dots, \nabla f_m(s)\}^*$, dashed ellipse shows $\mathcal{E}(\mathcal{B}^{w(s)})$, and solid rectangle shows $(\mathcal{B}_1^{w(s)})^* = \mathcal{B}_\infty^{w(s)}$.

**Diverse Influence.** To formally underpin the intuitive notion of "diverse influence" to an exact definition, we use convex geometry to characterize $\nabla f_1(s), \dots, \nabla f_m(s)$ (i.e., the rows of $J_f$). Note that when the rows of $J_f$ are sufficiently distinct, it means that $s$'s changes render different variations of $x_1, \dots, x_m$. In particular, we will exploit the cases where some observed dimensions $x_i = f_i(s)$ for $i \in [m]$ are affected by the changes of $s_1, \dots, s_d$ in a sufficiently different way.

> **Assumption 3.1** (Sufficiently Diverse Influence (SDI)). At $s \in \mathcal{S}$, there exists an $s$-dependent weighted $L_1$ norm ball $\mathcal{B}_1^{w(s)}$ such that $\nabla f_1(s), \dots, \nabla f_m(s) \in \mathcal{B}_1^{w(s)}$. In addition, the following two conditions hold:
>
> 1. $\mathcal{E}(\mathcal{B}_1^{w(s)}) \subseteq \mathrm{conv}\{\nabla f_1(s), \dots, \nabla f_m(s)\} \subseteq \mathcal{B}_1^{w(s)}$, and
> 2. $\mathrm{conv}\{\nabla f_1(s), \dots, \nabla f_m(s)\}^* \cap \mathrm{bd}(\mathcal{E}(\mathcal{B}_1^{w(s)})^*) = \mathrm{extr}(\mathcal{B}_\infty^{w(s)})$,
>
> where $\mathrm{conv}\{\cdot\}$ denotes the convex hull of a set of vectors, and for a given polytope $\mathcal{P}$, $\mathcal{E}(\mathcal{P})$ is its MVIE, $\mathcal{P}^*$ stands for its polar set, $\mathrm{extr}(\mathcal{P})$ contains its extreme points, and $\mathrm{bd}(\mathcal{P})$ is the polytope's boundary.

The SDI condition describes the scattering geometry of $\nabla f_1(s), \dots, \nabla f_m(s)$, which originates from the *sufficiently scattered condition* (SSC) in the matrix factorization literature. The SSC first appeared in identifiability analysis of *nonnegative matrix factorization* (NMF) [26] and has various forms in the SMF literature; see [25, 27–30, 36]. Here, SDI takes the form of SSC from *polytopic matrix factorization* (PMF) [30]; also see [31–33]. The key difference between SDI and SSC is that the former specifies the row-scattering pattern of a Jacobian $J_f(s)$ at every $s$ over a continuous manifold $\mathcal{S}$, yet SSC only characterizes the latent factors of a data matrix (e.g., $W, H$ in $X = WH$) that do not involve nonlinear functions or derivatives. But their geometries are essentially the same.

**Geometry of SDI.** Fig. 1 [Middle] illustrates Condition 1 of Assumption 3.1 for $d = 2$. One can see that SDI is about the spread of $\{\nabla f_1(s), \dots, \nabla f_m(s)\}$ in an $s$-dependent weighted $L_1$-norm ball $\mathcal{B}_1^{w(s)}$. We note that $\mathcal{B}_1^{w(s)}$ always exists, as long as $\{\nabla f_1(s), \dots, \nabla f_m(s)\}$ are finite. Geometrically, Condition 1 requires that the gradient vectors' convex hull contains the MVIE of $\mathcal{B}_1^{w(s)}$. Condition 2 is imposed on the polar sets $\mathrm{conv}\{\nabla f_1(s), \dots, \nabla f_m(s)\}^*$, $\mathcal{E}(\mathcal{B}_1^{w(s)})^*$ and $\mathcal{B}_\infty^{1/w(s)}$ (note that $\mathcal{B}_\infty^{1/w(s)}$, weighted by $1/w_1(s), \dots, 1/w_d(s)$, is the polar of $\mathcal{B}_1^{w(s)}$). The condition implies that $\mathrm{conv}\{\nabla f_1(s), \dots, \nabla f_m(s)\}^*$ touches the boundary of $\mathcal{E}(\mathcal{B}_1^{w(s)})^*$ at the vertices of $\mathcal{B}_\infty^{1/w(s)}$, which is a vector of the form $[\pm w_1(s)^{-1}, \dots, \pm w_d(s)^{-1}]^\top \in \mathbb{R}^d$. In the original domain, this means that the ellipsoid $\mathcal{E}(\mathcal{B}_1^{w(s)})$ must touch the convex hull $\mathrm{conv}\{\nabla f_1(s), \dots, \nabla f_m(s)\}$ at the facets of $\mathcal{B}_1^{w(s)}$. Fig. 1 [Right] shows the polar set view of Fig. 1 [Middle]. Readers are referred to [30] for a visualization of SSC under PMF, which further clarifies the connection between SSC and SDI.

**Physical Meaning of SDI.** SDI reflects how $s$ *diversely affects* $x_1, \dots, x_m$. Under SDI, there exist some observed features positively influenced by $s_j$ (i.e., $\partial x_i/\partial s_j > 0$), while also some features negatively influenced by $s_j$ (i.e., $\partial x_{i'}/\partial s_j < 0$). This pattern likely holds in many applications with high-dimensional data (i.e. $m \gg d$). This is because each $x_i$ can only either be positively or negatively influenced by $s_j$. As $m$ increases there would be more features that are affected differently by $s_j$.

Note that the positive and negative influences need not be symmetric (i.e., $\partial x_i/\partial s_j \neq -\partial x_{i'}/\partial s_j$ is allowed), as illustrated in Fig. 1 [Middle].

In addition, SDI assumes that each latent component $s_k$ *dominantly* influences at least two observed features, $x_{i_k^+}$ and $x_{i_k^-}$; here, "+" and "-" indicate that the features are positively and negatively affected by $s_k$, respectively. In other words, features satisfying the following should exist:

$$\partial x_{i_k^+}/\partial s_k \gg \partial x_i/\partial s_j, \ \forall j \neq i_k^+ \quad \text{and} \quad \partial x_{i_k^-}/\partial s_k \ll \partial x_{i-}/\partial s_j, \ \forall j \neq i_k^-. \tag{5}$$

**Discussion.** Beyond the basic geometric and physical interpretations, some additional remarks on other aspects of SDI are as follows:

*(i) Dependent $s$ and Dense Jacobian Can Satisfy SDI.* Notably, SDI does not impose statistical assumptions on $s$ (e.g., conditionally independent given auxiliary variables like in [10–12]). In addition, SDI does not rely on the sparsity of $J_f$ as in [2, 3, 19]. In fact, $J_f$ *can be completely dense* and at the same time satisfies SDI (see Fig. 1 [Middle]).

*(ii) SDI Favors $m \gg d$ Cases.* The SDI condition is arguably easier to satisfy when $m \gg d$ (which is justified in many applications, such as image/video generation [2, 40, 53]). It is evident that SDI requires at least $m = 2d$, which corresponds to the case where $\nabla f_1(s), ..., \nabla f_m(s)$ are located at $\pm w_1(s)e_1, ..., \pm w_d(s)e_d$. To see why SDI is in favor of larger $m$'s, consider a case where the rows of $J_f(s)$, i.e., $\nabla f_i(s)$, are drawn from a certain continuous distribution supported on $\mathcal{B}_1^{w(s)}$. Then, a large $m$ means that more realizations of $\nabla f_i(s)$'s are available. Therefore, for a fixed $d$, $m \to \infty$ leads to $\text{conv}\{\nabla f_1(s), ..., \nabla f_m(s)\}$ increasingly covering more of $\mathcal{B}_1^{w(s)}$ and eventually becoming $\mathcal{B}_1^{w(s)}$, which ensures that SDI condition is met. A related note is that a matrix factor $W \in \mathbb{R}^{m \times d}$ with larger $m$ under fixed $d$ would have higher probabilities to satisfy SSC, which was formally studied in [54], using exactly the same insight.

*(iii) SDI Encodes Influence Variations.* Lastly, the SDI condition allows the gradient pattern to vary from point to point on $\mathcal{S}$: at different $s \in \mathcal{S}$, both the ball $\mathcal{B}_1^{w(s)}$ and the position pattern of $\nabla f_1(s), ..., \nabla f_m(s)$ in $\mathcal{B}_1^{w(s)}$ can be different, as long as the gradients are sufficiently spread out in different directions as in Assumption 3.1.

## 3.2 Proposed Learning Criterion

Similar to the NMMI literature, e.g., [3, 10–12, 19], the goal of identifiability-guaranteed learning in this work is to find an invertible function (over the $\mathbb{R}^d$ manifold) $\widehat{f}$ such that $\widehat{s} = \widehat{f}^{-1}(x) = \widehat{f}^{-1} \circ f(s)$ recovers $s$ to a reasonable extent. In practice, we use a neural network (supposedly a universal function representer) $f_\theta$ as our learning function. To ensure the invertibility of $f_\theta : \mathbb{R}^d \to \mathbb{R}^m$ over the manifold $\mathcal{S} \subseteq \mathbb{R}^d$, we use another neural network $g_\phi : \mathbb{R}^m \to \mathbb{R}^d$ and enforce

$$x = f_\theta(g_\phi(x)), \ \forall x \in \mathcal{X}. \tag{6}$$

Note that if the above holds, both $f_\theta$ and $g_\phi$ are invertible over the data-generating $d$-dimensional manifold. Eq. (6) is nothing but a stacked autoencoder, which by itself does not ensure identifiability of the model (1). To utilize SDI for establishing identifiability, we propose the following learning criterion, namely, *Jacobian volume maximization* (J-VolMax):

$$\text{(J-VolMax)} \quad \underset{\theta, \phi}{\text{maximize}} \ \mathbb{E}[\log \det(J_{f_\theta}(g_\phi(x))^\top J_{f_\theta}(g_\phi(x)))] \tag{7a}$$

$$\text{subject to: } ||J_{f_\theta}(g_\phi(x))_{i,:}||_1 \leq C, \ \forall i = 1, ..., m, \tag{7b}$$

$$x = f_\theta(g_\phi(x)), \ \forall x \in \mathcal{X} \tag{7c}$$

The term $\log \det(J_{f_\theta}(g_\phi(x))^\top J_{f_\theta}(g_\phi(x)))$ represents the squared volume of the convex hull spanned by the columns of $J_{f_\theta}$.

Using this criterion, the partial derivatives contained in $J_{f_\theta}(\widehat{s})$ (where $\widehat{s} = g_\phi(x)$) are encouraged to scatter in space—reflecting the belief that the influences of $s$ on different $x_i$'s are diverse.

**Identifiability Result.** Under the model in (1) and Assumption 3.1, we show our main result:

**Theorem 3.2** (Identifiability of J-VolMax). *Denote any optimal solution of Problem* (7) *as* $(\widehat{\boldsymbol{\theta}}, \widehat{\boldsymbol{\phi}})$. *Assume* $\widehat{\boldsymbol{f}} = \boldsymbol{f}_{\widehat{\boldsymbol{\theta}}}$ *and* $\widehat{\boldsymbol{g}} = \boldsymbol{g}_{\widehat{\boldsymbol{\phi}}}$ *are universal function representers. Suppose the model in* (1) *and Assumption 3.1 hold for every* $\boldsymbol{s} \in \mathcal{S}$. *Then, we have* $\widehat{\boldsymbol{s}} = \widehat{\boldsymbol{g}}(\boldsymbol{x}) = \widehat{\boldsymbol{g}} \circ \boldsymbol{f}(\boldsymbol{s})$ *where*

$$[\widehat{\boldsymbol{s}}]_i = [\widehat{\boldsymbol{g}}(\boldsymbol{x})]_i = \rho_i(s_{\boldsymbol{\pi}(i)}), \ \forall i \in [d], \tag{8}$$

*in which* $\boldsymbol{\pi}$ *is a permutation of* $\{1, \ldots, d\}$ *and* $\rho_i(\cdot) : \mathbb{R} \to \mathbb{R}$ *is an invertible function.*

Notice that we used the constraint $||\boldsymbol{J}_{\boldsymbol{f}_{\boldsymbol{\theta}}}(\boldsymbol{g}_{\boldsymbol{\phi}}(\boldsymbol{x}))_{i,:}||_1 \leq C$ in (7) where $C > 0$ is unknown. Under SDI, the ground-truth Jacobian is bounded by $||\boldsymbol{J}_{\boldsymbol{f}}(\boldsymbol{s})_{i,:}||_1 \in \mathcal{B}_1^{\boldsymbol{w}(\boldsymbol{s})}$ with an *unknown* $\mathcal{B}_1^{\boldsymbol{w}(\boldsymbol{s})}$. Nonetheless, using an arbitrary $L_1$-norm ball with radius $C$ in (7) does not affect identifiability of the J-VolMax criterion—the learned $\widehat{\boldsymbol{s}}$ will have a scaling ambiguity anyway.

The proof of the theorem consists of three major steps. First, we recast the nonlinear identifiability problem J-VolMax into its first-order derivative domain as a matrix-finding problem at each $\boldsymbol{s} \in \mathcal{S}$, which is closely related to intermediate steps in matrix factor identification under the PMF model [30]. Second, consequently, we utilize SDI and algebraic properties from volume maximization-based PMF to underpin identifiability under J-VolMax at each $\boldsymbol{s}$ up to an $\boldsymbol{s}$-dependent permutation ambiguity. Third, we invoke continuity of $\boldsymbol{f}$ and its domain $\mathcal{S}$ to unify permutation ambiguity over the entire $\mathcal{S}$. The details can be found in Appendix B.

**Finite-Sample SDI and Identifiability.** In Theorem 3.2, an assumption is that every $\boldsymbol{s}$ in the continuous domain $\mathcal{S}$ has to satisfy SDI, which could be relatively restrictive. To proceed, we show that, as $\boldsymbol{f}$ and the learned functions satisfy certain conditions, having a finite number of $\boldsymbol{s}$ satisfying SDI suffices to establish identifiability up to bounded errors.

To see how we approach this, consider a finite set $\mathcal{S}_N := \{\boldsymbol{s}^{(1)}, ..., \boldsymbol{s}^{(N)}\}$ with $\mathcal{X}_N := \{\boldsymbol{x} \in \mathcal{X} : \boldsymbol{x} = \boldsymbol{f}(\boldsymbol{s}), \forall \boldsymbol{s} \in \mathcal{S}_N\}$ such that Assumption 3.1 is satisfied at each of the $N$ points in $\mathcal{S}_N$. Note that at each $\boldsymbol{s}^{(n)}$, the optimal encoder $\widehat{\boldsymbol{g}}$ recovers $\boldsymbol{s}^{(n)}$ from the observation $\boldsymbol{x}^{(n)}$ up to permutation $\widehat{\boldsymbol{\Pi}}(\boldsymbol{s}^{(n)})$ and an invertible element-wise map $\widehat{\boldsymbol{\rho}}(\boldsymbol{s}^{(n)})$, i.e.,

$$\widehat{\boldsymbol{g}}(\boldsymbol{x}^{(n)}) = \widehat{\boldsymbol{\Pi}}(\boldsymbol{s}^{(n)})\widehat{\boldsymbol{\rho}}(\boldsymbol{s}^{(n)}), \ \forall \boldsymbol{x}^{(n)} \in \mathcal{X}_N, \tag{9}$$

which is a direct result when using the J-VolMax learning criterion; see Lemma C.2. The following result shows that under certain regularity conditions, if the set $\mathcal{S}_N$ contains samples that locate densely enough in space, the learned encoder can recover the ground-truth $\boldsymbol{s}$ up to the same ambiguities as in Theorem 3.2, with a bounded error that decays as $N$ grows.

**Theorem 3.3** (Identifiability under Finite-sample SDI). *Assume that there is a finite set* $\mathcal{S}_N := \{\boldsymbol{s}^{(1)}, ..., \boldsymbol{s}^{(N)}\}$ *with* $\mathcal{X}_N := \{\boldsymbol{x} \in \mathcal{X} : \boldsymbol{x} = \boldsymbol{f}(\boldsymbol{s}), \forall \boldsymbol{s} \in \mathcal{S}_N\}$ *such that the Assumption 3.1 is satisfied at each of the* $N$ *points in* $\mathcal{S}_N$. *Let* $\widehat{\boldsymbol{g}} \in \mathcal{G}$ *be the optimal encoder by J-VolMax criterion* (7), *and* $\overline{\Theta}$ *contains the parameters of the learned encoder and decoder. Further assume that the following regularity conditions hold:*

1. *The functions* $\boldsymbol{g} = \boldsymbol{f}^{-1}$ *and* $\boldsymbol{g}_{\boldsymbol{\phi}}$ *are from classes* $\mathcal{G}'$ *and* $\mathcal{G}$, *respectively, where* $\mathcal{G}' \subseteq \mathcal{G}$.
2. *The functions* $\boldsymbol{f}, \widehat{\boldsymbol{g}}, \widehat{\boldsymbol{\rho}}$ *are Lipschitz continuous with constants* $L_{\boldsymbol{f}}, L_{\widehat{\boldsymbol{g}}}, L_{\widehat{\boldsymbol{\rho}}} > 0$.
3. *There is a* $\gamma > 0$ *such that for any permutation matrix* $\boldsymbol{\Pi} \in \mathcal{P}_d$ *and* $\boldsymbol{\Pi} \neq \widehat{\boldsymbol{\Pi}}(\boldsymbol{s}^{(n)})$ *(from* (9)*),*

$$||\widehat{\boldsymbol{g}}(\boldsymbol{x}^{(n)}) - \boldsymbol{\Pi}\widehat{\boldsymbol{\rho}}(\boldsymbol{s}^{(n)})||_2 \geq \gamma, \ \forall n \in [N].$$

4. *For* $\mathcal{N}^{(n)} = \{\boldsymbol{s} \in \mathcal{S} : ||\boldsymbol{s} - \boldsymbol{s}^{(n)}||_2 < r^{(n)}\}$ *with* $r^{(n)} < \frac{\gamma}{2(L_{\boldsymbol{f}}L_{\widehat{\boldsymbol{g}}} + L_{\widehat{\boldsymbol{\rho}}})}$, *the union of the neighborhoods,* $\mathcal{N} := \bigcup_{n=1}^{N} \mathcal{N}^{(n)}$, *is a connected subset of* $\mathcal{S}$ *and* $V(\boldsymbol{s}; \overline{\Theta})$ *is optimal for any* $\boldsymbol{s} \in \mathcal{N}$.
5. *The points* $\boldsymbol{s}^{(1)}, \ldots, \boldsymbol{s}^{(N)} \in \mathcal{S}_N$ *densely locate in* $\mathcal{S}$ *such that*

$$\mathbb{P}(\boldsymbol{s} \in \mathcal{N}) / \mathbb{P}(\boldsymbol{s} \in \mathcal{S} \setminus \mathcal{N}) > G_{\max} / G_{\min} > 1, \tag{10}$$

*where* $G_{\min}, G_{\max}$ *are bi-Lipschitz constants of the Jacobian volume surrogate: for any parameters* $\Theta_1, \Theta_2$ *in* $(\mathcal{F}, \mathcal{G})$,

$$G_{\min}||\Theta_1 - \Theta_2||_2 \leq |V(\boldsymbol{s}; \Theta_1) - V(\boldsymbol{s}; \Theta_2)| \leq G_{\max}||\Theta_1 - \Theta_2||_2, \ \forall \boldsymbol{s} \in \mathcal{S}. \tag{11}$$

*Then, $\widehat{\boldsymbol{g}}(\boldsymbol{x}^{(n)}) = \widehat{\boldsymbol{\Pi}}\widehat{\boldsymbol{\rho}}(\boldsymbol{s}^{(n)}), \forall n \in [N]$ for a constant permutation matrix $\widehat{\boldsymbol{\Pi}} \in \mathcal{P}_d$. Furthermore, with probability at least $1 - \delta$,*

$$\mathbb{E}_{\boldsymbol{s}\sim p(\boldsymbol{s})}[||\widehat{\boldsymbol{g}}(\boldsymbol{x}) - \widehat{\boldsymbol{\Pi}}\widehat{\boldsymbol{\rho}}(\boldsymbol{s})||_2] = \mathcal{O}\left((L_{\boldsymbol{f}}L_{\widehat{\boldsymbol{g}}} + L_{\widehat{\boldsymbol{\rho}}})\mathcal{R}_N(\mathcal{G}) + \sqrt{\ln(1/\delta)/N}\right), \quad (12)$$

*where $\mathcal{R}_N(\mathcal{G})$ is the empirical Rademacher complexity of the encoder class.*

The theorem implies that if the number of $\{\boldsymbol{s}^{(n)}\}_{n=1}^N$ satisfying SDI is sufficiently large, the identification error defined in (12) can be made arbitrarily small. Note that when $\mathcal{G}$ is not a universal function class—that is, its capacity is controlled, e.g., through bounded norms or Lipschitz constants—its empirical Rademacher complexity satisfies $\mathcal{R}_N(\mathcal{G}) = \mathcal{O}(1/\sqrt{N})$ [55]. Consequently, with probability at least $1-\delta$, the overall error rate is $\mathcal{O}(\sqrt{\ln(1/\delta)}+1/\sqrt{N}) = \tilde{\mathcal{O}}(1/\sqrt{N})$. The detailed proof is provided in Appendix C.

In Theorem 3.3, Condition 1 means that the learning function class can faithfully represent $\boldsymbol{g}$. Condition 2 requires that $\boldsymbol{f}$ and the learned functions have bounded continuity constants, which is a mild assumption as the learning functions are often represented using continuous function classes (e.g., neural networks). Condition 3 assumes that at each point $\boldsymbol{s}^{(n)} \in \mathcal{S}_N$, there is a unique optimal permutation between $\boldsymbol{s}^{(n)}$ and $\widehat{\boldsymbol{g}}(\boldsymbol{x}^{(n)})$. That is, $\gamma = 0$ holds for the optimal permutation, but an error of $\gamma > 0$ occurs for all other permutations. Such a condition naturally holds under sufficient continuity of the functions. Condition 4 translates to the fact that the points $\boldsymbol{s}^{(1)}, ..., \boldsymbol{s}^{(N)} \in \mathcal{S}_N$ locate densely to each other so that their neighborhoods $\mathcal{N}^{(n)}$ (with radius $r^{(n)}$) overlap. Intuitively, if each $\mathcal{N}^{(n)}$ has a unified permutation for all $\boldsymbol{s}$ within $\mathcal{N}^{(n)}$, such overlapping leads to the same permutation for the union of all $\mathcal{N}^{(n)}$'s. Condition 5 means that there are enough points $\boldsymbol{s}^{(1)}, ..., \boldsymbol{s}^{(n)}$ and they are densely positioned in $\mathcal{S}$, such that $\mathcal{N}$ covers a sufficiently large fraction of $\mathcal{S}$. This required fraction depends on the bi-Lipschitz constants with respect to $\Theta$ of Jacobian volume surrogate $V(\boldsymbol{s}; \Theta)$.

## 4  Related Works

**IMA, Sparse Jacobian, and Object-Centric Learning.** As mentioned in "Motivation" (see Sec. 3.1), IMA [20] and sparse Jacobian-based methods (e.g., [2, 19, 49, 50]) are conceptually related to our work. Additional remarks on connections between the SDI condition and IMA, Jacobian sparsity, anchor features [49], and object-centric methods [2, 23, 24], can be found in Appendix D.

**SSC and Matrix Factorization.** Volume-regularization approaches are popular in SMF models—i.e., $\boldsymbol{X} = \boldsymbol{W}\boldsymbol{H}$ with structural constraints (e.g., boundedness and nonnegativity) on $\boldsymbol{W}$ and/or $\boldsymbol{H}$ [27, 30–33]—where SSC-like conditions are imposed on the matrix factors to ensure identifiability of $\boldsymbol{W}$ and $\boldsymbol{H}$. The work [25] was the first to employ an SSC defined over the nonnegative orthant (originated from [26]) to show that latent-factor volume minimization identifies the corresponding SMF model. There, SSC requires that the columns/rows of a factor matrix widely spread in the nonnegative orthant. This line of work later was generalized to SSC in different norm balls [30, 31]. The SDI condition can be viewed as an extension of SSC in [30] (also see a similar SSC in [31, 33]) into the Jacobian domain over a continuous manifold. Although NMMI and SMF contexts differ substantially, some mathematical properties resulted from SSC (coupled with a volume-maximizing criterion) in [30] provide key stepping stones for identifiability analysis in this work.

**Maximum Likelihood Estimation (MLE) and InfoMax for ICA.** Both InfoMax [56] and MLE [57] handle ICA via optimizing the Jacobian volume in their formulations. Nonetheless, the Jacobian volume arises in their learning criteria as a result of information-theoretic or statistical estimation-based derivation (e.g., being surrogate of entropy). Model identifiability under nonlinear mixture models has not been established under these frameworks yet.

## 5  Numerical Results

**Implementation**[1]**.** Given $L$ realizations of $\boldsymbol{x}$, i.e., $\{\boldsymbol{x}^{(1)}, \boldsymbol{x}^{(2)}, ..., \boldsymbol{x}^{(L)}\}$, we use multi-layer perceptrons (MLPs) to parametrize $\boldsymbol{f}_{\boldsymbol{\theta}}$ and $\boldsymbol{g}_{\boldsymbol{\phi}}$. We implement a regularized version of J-VolMax in (7) and train for $T$ iterations via a warm-up heuristic with the number of warm-up epochs is $T_{\text{w}} < T$. Our

---

[1]Our implementation code is available here: https://github.com/hsnguyen24/dica

loss function $\mathcal{L}_t$ at the $t$th epoch is

$$\min_{\boldsymbol{\theta},\boldsymbol{\phi}} \mathcal{L}_t := \frac{1}{L} \sum_{n=1}^{L} \left( ||\boldsymbol{x}^{(n)} - \boldsymbol{f}_{\boldsymbol{\theta}}(\boldsymbol{g}_{\boldsymbol{\phi}}(\boldsymbol{x}^{(n)}))||_2^2 - \lambda_{\mathrm{vol}}(t) \times c_{\mathrm{vol}} + \lambda_{\mathrm{norm}} \times c_{\mathrm{norm}}(t) \right). \quad (13)$$

In the above, $c_{\mathrm{vol}}$ is defined as

$$c_{\mathrm{vol}} := \log \det(\boldsymbol{J}_{\boldsymbol{f}_{\boldsymbol{\theta}}}(\boldsymbol{g}_{\boldsymbol{\phi}}(\boldsymbol{x}^{(n)}))^\top \boldsymbol{J}_{\boldsymbol{f}_{\boldsymbol{\theta}}}(\boldsymbol{g}_{\boldsymbol{\phi}}(\boldsymbol{x}^{(n)}))), \quad (14)$$

and $\lambda_{\mathrm{vol}}(t) := \frac{\lambda_{\mathrm{vol}}}{T_{\mathrm{w}}} \min\{t, T_{\mathrm{w}}\}$; i.e., $\lambda_{\mathrm{vol}}(t)$ linearly increases from $t=0$ to a chosen $\lambda_{\mathrm{vol}} > 0$ at the end of warm-up phase, i.e., $t = T_{\mathrm{w}}$. Note that the explicit form of (14) might impose computational challenges when $m$ and $d$ are large. Hence, for high-dimensional datasets, one can use a trace-based surrogate of $c_{\mathrm{vol}}$, reducing the per-iteration flops from $\mathcal{O}(m^3)$ to $\mathcal{O}(m^2)$; see Appendix E.1.

The term $c_{\mathrm{norm}}(t)$ implements the norm constraint in (7b) via

$$c_{\mathrm{norm}}(t) := \begin{cases} ||\boldsymbol{J}_{\boldsymbol{f}_{\boldsymbol{\theta}}}(\boldsymbol{g}_{\boldsymbol{\phi}}(\boldsymbol{x}^{(n)}))||_1 & \text{if } t \leq T_{\mathrm{w}} \\ \mathtt{Softplus}\{||\boldsymbol{J}_{\boldsymbol{f}_{\boldsymbol{\theta}}}(\boldsymbol{g}_{\boldsymbol{\phi}}(\boldsymbol{x}^{(n)}))||_1 - C\} & \text{if } t > T_{\mathrm{w}} \end{cases}, \quad (15)$$

with $\lambda_{\mathrm{norm}} > 0$. The element-wise function $\mathtt{Softplus}(z) = \ln(1 + e^z) \in \mathbb{R}$ is used as a smooth approximation of the hinge loss (see [58]). This term serves as a soft version of the norm constraint (7b) in J-VolMax; that is, it penalizes large $L_1$ norm values but does not put a hard constraint on their upper bound; see [59].

After the warm-up period, $C$ is set to be the average of $||\boldsymbol{J}_{\boldsymbol{f}_{\boldsymbol{\theta}}}(\boldsymbol{g}_{\boldsymbol{\phi}}(\boldsymbol{x}^{(n)}))||_1$ in the last 10 epochs, to avoid that the $c_{\mathrm{norm}}$ term disproportionally skews the optimization process. More discussions on implementation can be found in Appendix E.1.

**Evaluation.** Following standard practice in NMMI [8, 12], we calculate both the mean Pearson correlation coefficient ($MCC$) and the mean coefficient of determination $R^2$ over pairs of $d$ latent components, after fixing the permutation ambiguity via the Hungarian algorithm [60]. While $MCC$ can only measure linear correlation, the $R^2$ score can measure nonlinear dependence. A perfect $R^2$ score means there is a mapping between the estimated latent component and the ground truth.

**Baselines.** We use the unregularized autoencoder (`Base`), autoencoder with the IMA contrast [20, 52] (`IMA`), and the sparsity-regularized loss (`Sparse`) as in (4). The $L_1$-norm regularization with weight $\lambda_{\mathrm{sp}} > 0$ is used to promote Jacobian sparsity, as in [19, 50]. To present the best performance of all the baselines under the considered settings, we tune their hyperparameters in a separately generated validation dataset with known ground-truth latent components. The hyperparameters $\lambda_{\mathrm{vol}}, \lambda_{\mathrm{norm}},$ and $\lambda_{\mathrm{sp}}$ are chosen by grid search from $\{10^{-2}, 10^{-3}, 10^{-4}, 10^{-5}\}$ on the same validation set.

## 5.1 Synthetic Data Experiments

**Data Generation.** We generate three types of mixtures to test our method. The generating process and ground-truth latent variables are unknown to our algorithms and the baselines. We use such controlled generation for constructing SDI-satisfying $\boldsymbol{f}$'s. In all simulations, we use two fully-connected ReLU neural networks with one hidden layer of 64 neurons to represent $\boldsymbol{f}_{\boldsymbol{\theta}}$ and $\boldsymbol{g}_{\boldsymbol{\phi}}$.

_**Mixture A**_: We use a linear mixture as the first checkpoint of our theory. Here, the latent components $\boldsymbol{s}$ are sampled from a normal distribution $p(\boldsymbol{s}) = \mathcal{N}(\boldsymbol{0}, \boldsymbol{\Sigma})$, where the covariance $\boldsymbol{\Sigma}$ is drawn from a Wishart distribution $W_p(\boldsymbol{I}, d)$. This way, the components of $\boldsymbol{s}$ are dependent. The observed mixtures are created by generating a matrix $\boldsymbol{A} \in \mathbb{R}^{m \times d}$ to form $\boldsymbol{x} = \boldsymbol{A}\boldsymbol{s}$. The matrix $\boldsymbol{A}$ can be generated to approximately satisfy the SDI condition by sampling $m$ points in $\mathbb{R}^d$ randomly from an inflated $L_2$ ball, and then projecting those points onto the chosen weighted norm ball $\mathcal{B}_1^{\boldsymbol{w}(\boldsymbol{s})}$ to create the $m$ rows of $\boldsymbol{A}$; see [30] for a similar way to generate matrices that approximate SSC.

_**Mixture B**_: On top of $\boldsymbol{z} = \boldsymbol{A}\boldsymbol{s}$ as in Mixture A, we also apply an element-wise nonlinear distortion to create nonlinear mixtures, i.e., $\boldsymbol{x} = \boldsymbol{f}(\boldsymbol{z})$ and $[\boldsymbol{f}(\boldsymbol{z})]_i = a\cos(z_i) + z_i$, where $a \sim U(0.5, 1.0)$. Notice that $\boldsymbol{J}_{\boldsymbol{f}}(\boldsymbol{s}) = \mathrm{Diag}(1 - a\sin(\boldsymbol{A}\boldsymbol{s}))\boldsymbol{A}$ under this construction. Therefore, with $\boldsymbol{A}$ approximately satisfying SDI and the nonlinear distortion weight $a$ being sufficiently small, $\boldsymbol{J}_{\boldsymbol{f}}(\boldsymbol{s})$ also approximately satisfies the SDI.

_**Mixture C**_: This mixture is generated by $\boldsymbol{x} = \boldsymbol{f}(\boldsymbol{s})$, where $\boldsymbol{s} \sim U(-1, 1)$ and the nonlinear mixing function is $\boldsymbol{f}(\cdot) = (f_1(\cdot), ..., f_m(\cdot)) \in \mathbb{R}^m$. The functions $f_k : \mathbb{R}^d \to \mathbb{R}, k = 1, ..., m$ are MLPs

Table 1: Nonlinear $R^2$ and $MCC$ scores (mean $\pm$ std., over 10 random trials).

| Model | $(d, m)$ | Mixture A | | Mixture B | | Mixture C | |
|---|---|---|---|---|---|---|---|
| | | $R^2$ | $MCC$ | $R^2$ | $MCC$ | $R^2$ | $MCC$ |
| DICA | $(2, 30)$ | $0.92 \pm 0.16$ | $0.93 \pm 0.15$ | $0.97 \pm 0.06$ | $0.98 \pm 0.03$ | $0.95 \pm 0.13$ | $0.97 \pm 0.09$ |
| | $(3, 40)$ | $0.92 \pm 0.11$ | $0.94 \pm 0.08$ | $0.99 \pm 0.02$ | $0.99 \pm 0.01$ | $0.90 \pm 0.12$ | $0.95 \pm 0.07$ |
| | $(4, 50)$ | $0.98 \pm 0.06$ | $0.98 \pm 0.04$ | $0.92 \pm 0.10$ | $0.95 \pm 0.07$ | $0.87 \pm 0.12$ | $0.92 \pm 0.07$ |
| | $(5, 60)$ | $0.92 \pm 0.10$ | $0.93 \pm 0.09$ | $0.89 \pm 0.13$ | $0.93 \pm 0.08$ | $0.87 \pm 0.11$ | $0.93 \pm 0.06$ |
| Sparse | $(2, 30)$ | $0.79 \pm 0.19$ | $0.83 \pm 0.12$ | $0.88 \pm 0.18$ | $0.82 \pm 0.12$ | $0.87 \pm 0.12$ | $0.92 \pm 0.07$ |
| | $(3, 40)$ | $0.71 \pm 0.15$ | $0.78 \pm 0.13$ | $0.72 \pm 0.11$ | $0.80 \pm 0.09$ | $0.71 \pm 0.11$ | $0.83 \pm 0.07$ |
| | $(4, 50)$ | $0.65 \pm 0.13$ | $0.72 \pm 0.11$ | $0.71 \pm 0.10$ | $0.81 \pm 0.06$ | $0.64 \pm 0.12$ | $0.79 \pm 0.07$ |
| | $(5, 60)$ | $0.63 \pm 0.10$ | $0.71 \pm 0.07$ | $0.67 \pm 0.07$ | $0.78 \pm 0.05$ | $0.60 \pm 0.09$ | $0.76 \pm 0.06$ |
| Base | $(2, 30)$ | $0.88 \pm 0.13$ | $0.86 \pm 0.11$ | $0.81 \pm 0.17$ | $0.88 \pm 0.11$ | $0.77 \pm 0.12$ | $0.87 \pm 0.08$ |
| | $(3, 40)$ | $0.69 \pm 0.08$ | $0.74 \pm 0.07$ | $0.68 \pm 0.12$ | $0.80 \pm 0.09$ | $0.62 \pm 0.14$ | $0.77 \pm 0.09$ |
| | $(4, 50)$ | $0.54 \pm 0.12$ | $0.63 \pm 0.09$ | $0.63 \pm 0.15$ | $0.76 \pm 0.10$ | $0.48 \pm 0.08$ | $0.68 \pm 0.06$ |
| | $(5, 60)$ | $0.42 \pm 0.09$ | $0.53 \pm 0.07$ | $0.54 \pm 0.06$ | $0.71 \pm 0.04$ | $0.47 \pm 0.06$ | $0.66 \pm 0.04$ |
| IMA | $(2, 30)$ | $0.86 \pm 0.14$ | $0.92 \pm 0.10$ | $0.84 \pm 0.13$ | $0.91 \pm 0.07$ | $0.84 \pm 0.14$ | $0.91 \pm 0.07$ |
| | $(3, 40)$ | $0.70 \pm 0.10$ | $0.83 \pm 0.06$ | $0.69 \pm 0.13$ | $0.82 \pm 0.09$ | $0.67 \pm 0.07$ | $0.81 \pm 0.04$ |
| | $(4, 50)$ | $0.70 \pm 0.10$ | $0.83 \pm 0.07$ | $0.68 \pm 0.09$ | $0.81 \pm 0.06$ | $0.56 \pm 0.11$ | $0.74 \pm 0.07$ |
| | $(5, 60)$ | $0.66 \pm 0.05$ | $0.80 \pm 0.04$ | $0.63 \pm 0.10$ | $0.79 \pm 0.06$ | $0.53 \pm 0.08$ | $0.72 \pm 0.05$ |

Table 2: Nonlinear $R^2$ scores for different ratios $m/d$ (mean $\pm$ std., over 10 random trials).

| $(d, m)$ | DICA | IMA | Sparse | Base |
|---|---|---|---|---|
| $(3, 40)$ | $0.90 \pm 0.10$ | $0.63 \pm 0.15$ | $0.80 \pm 0.13$ | $0.63 \pm 0.10$ |
| $(3, 30)$ | $0.89 \pm 0.07$ | $0.63 \pm 0.12$ | $0.77 \pm 0.12$ | $0.63 \pm 0.14$ |
| $(3, 20)$ | $0.82 \pm 0.17$ | $0.60 \pm 0.12$ | $0.74 \pm 0.13$ | $0.60 \pm 0.10$ |
| $(3, 10)$ | $0.71 \pm 0.17$ | $0.56 \pm 0.09$ | $0.75 \pm 0.12$ | $0.63 \pm 0.16$ |
| $(3, 5)$ | $0.64 \pm 0.09$ | $0.66 \pm 0.16$ | $0.78 \pm 0.16$ | $0.61 \pm 0.08$ |
| $(3, 3)$ | $0.55 \pm 0.07$ | $0.60 \pm 0.09$ | $0.58 \pm 0.16$ | $0.51 \pm 0.09$ |

whose layer weights are randomly generated. In addition, we pick $f_1, f_2, ..., f_{\lfloor m/2 \rfloor}$ and modify them as follows: *a)* randomly choose $d - 1$ elements from $[d]$; and *b)* add an input layer that down-scales the chosen $d - 1$ latent component $s_j$ by $\tilde{s}_j = \alpha_j \beta_j s_j$, where $\alpha_j \sim U(0.001, 0.002)$ and $\mathbb{P}(\beta_j = 1) = \mathbb{P}(\beta_j = -1) = 1/2$. This way, the first half of $\nabla f_k(\boldsymbol{s})$'s create a subset of gradients that are dominated by one latent component (cf. "Physical Meaning of SDI" in Section 3.1).

We should mention that, unlike Mixtures A and B, we have less control on generating SDI-satisfying mixtures under Mixture C. Hence, this type of nonlinear mixture can be used to test the model robustness of our approach.

**Results.** Table 1 shows both the $MCC$ and the nonlinear $R^2$ scores of all four methods, namely, DICA, IMA, Sparse, and Base, on Mixtures A, B, C. We observe that for Mixtures A and B, both scores attained by DICA are mostly above $0.92$ (except for $R^2$ under $(d, m) = (5, 60)$). In contrast, IMA, Sparse, and Base do not perform as well. Importantly, the performance of DICA does not significantly degrade as $m$ and $d$ grows, but the baselines appear to struggle under larger $m$'s and $d$'s. For the more challenging Mixture C, DICA's performance scores still have a substantial margin over the baselines, and the scores remain largely the same when $(d, m)$ varies.

**Ablation on $m/d$.** We examine the performance of DICA when varying the ratio $m/d$. Per our discussion (cf. Sec. 3), SDI favors $m \gg d$ cases. As the ratio $m/d$ approaches 2, SDI is less likely to be satisfied. In Table 2, we use Mixture C to test the performance of DICA where we fix $d = 3$ and vary $m$. Consistent with our analysis, the performance of DICA deteriorates as $m$ approaches $2d$. For $m < 2d$, the performance of DICA is similar to vanilla autoencoder—showing that the effect of volume regularization vanishes in these cases. This result confirms that DICA is suitable for learning low-dimensional latent components from high-dimensional data, e.g., image representation learning.

## 5.2 Inferring Transcription Factor Activities from Single-cell Gene Expressions

In this experiment, we apply J-VolMax (7) to inferring transcription factors (TFs) activities in single-cell transcriptomics analysis [61]. Specifically, we employ J-VolMax to infer the ground-truth mRNA concentrations of TFs from gene expression data. In this context, our single-cell gene expressions are from a bio-realistic generator, namely, SERGIO [62]; see the use of SERGIO in the NMMI literature [41, 63]. The mRNA concentration levels $\boldsymbol{s} \in \mathbb{R}^d$ of the TFs are governed from a chemical Langevin

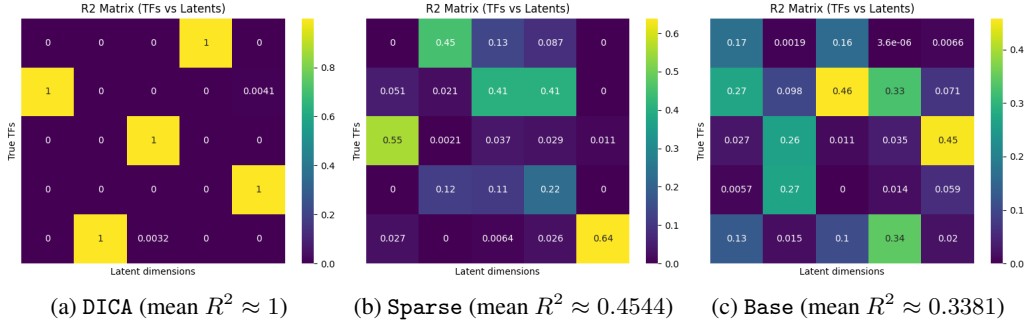

Figure 2: Heatmap of $R^2$ scores between estimated components and ground-truth mRNA concentrations of TFs.

equation, and the gene expressions (i.e., samples of $x \in \mathbb{R}^m$) are generated by a gene regulatory mechanism $f : \mathbb{R}^d \mapsto \mathbb{R}^m$ applied onto $s$.

We use the TRRUST dataset of TF-gene pairs of mouse [64]. It is a manually curated dataset that includes high-confidence interactions between TFs and their target genes. These interactions are either activating or repressing. All entries are supported by experimental evidence from multiple published studies. We sample a sub-network from the large gene regulatory network provided by TRRUST, via which $m = 178$ genes are governed by $d = 5$ master regulator TFs. More details of the data generation process using SERGIO is in Appendix E.3.

Fig. 2 shows average $R^2$ scores (from 10 trials) for each $s_1, ..., s_5$ up to best permutation, as obtained by DICA, Sparse, and Base. One can see that DICA successfully infers the ground-truth mRNA concentration level $s_1, ..., s_5$ of each TF (up to a permutation and invertible mapping), but Sparse and Base are not able to attain reasonable $R^2$ scores. Furthermore, the proposed method can disentangle the influences of different TFs. Specifically, each learned latent component is only matched to a unique ground-truth TF with $R^2$ score $\approx 1$, yet the $R^2$ score is approximately 0 when matching with other TFs. In contrast, Base could not disentangle the TFs at all. Similarly, Sparse could only mildly disentangle two TFs (i.e., the 3th and the 5th TFs). Additionally, the $MCC$ score of DICA reaches $\approx 1$, whereas Sparse and Base output $MCC \approx 0.6635$ and $MCC \approx 0.5721$, respectively.

Some remarks regarding this experiment is as follows. Note that the data generation process follows biology-driven mechanisms in SERGIO, and thus we do not have a way to enforce SDI. However, as the TFs are believed to have diverse influences on the gene expressions (e.g., due to the activating/repressing effects on genes by the TFs), it is reasonable to assume that SDI approximately holds in this application—which perhaps explains why DICA works well. On the other hand, some TFs may influence many gene expressions ([65]), and many cross-interactions between TFs and genes exist ([66]). Hence, the assumption that the Jacobian of $f$ is strictly sparse might be stringent. Therefore, the sparse Jacobian approach may not be a good fit, as its performance indicates.

## 6 Conclusion

We introduced DICA, a new paradigm for nonlinear mixture learning with identifiability guarantees. Via the insight of convex geometry, we showed that, as long as the latent components have sufficiently different influences on the observed features so that a geometric condition, namely, the SDI condition, is satisfied, the mixture model is identifiable via fitting the generative model and maximizing the Jacobian of the learning function simultaneously. This J-VolMax criterion ensures identifiability without relying on many assumptions in the NMMI literature, e.g., availability of auxiliary information, independence of latent components, or the sparsity of the mixing function's Jacobian, thus offering a valuable alternative to existing NMMI methods. The experiments supported our theory.

**Limitations and Future Works.** We noticed a number of limitations. First, the DICA framework has a $\log \det$ term to compute, which poses a challenging optimization problem. This is particularly hard as this term has to be evaluated at every realization of $s$. How to optimize the J-VolMax criterion efficiently is an interesting yet challenging direction to consider. Second, the identifiability analysis was done under ideal conditions where noise is absent, one has access to unlimited samples, and the learner class is assumed to perfectly include the target function. While J-VolMax is empirically shown to be quite robust in experiments, a more thorough identifiability analysis under practical conditions would close the gap, e.g., by analyzing sample complexity under noise and model mismatches.

**Acknowledgment**: This work is supported in part by the National Science Foundation (NSF) CAREER Award ECCS-2144889. The authors would like to thank Sagar Shrestha for insightful discussions on the identifiability analysis, and the anonymous reviewers for constructive feedbacks.

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

# Supplementary Material of "Diverse Influence Component Analysis: A Geometric Approach to Nonlinear Mixture Identifiability"

## A  Preliminaries

### A.1  Notation

- For $m \in \mathbb{N}$, $[m]$ is a shorthand for $\{1, 2, ..., m\}$.

- $x, \boldsymbol{x}$ and $\boldsymbol{X}$ denotes a scalar, a vector, and a matrix, respectively.

- $\boldsymbol{X}_{i,j}$, $[\boldsymbol{X}]_{i,j}$ and $x_{i,j}$ denotes the entry at $i$th row and $j$th column of $\boldsymbol{X}$. $\boldsymbol{X}_{i:}$ and $\boldsymbol{X}_{:j}$ denotes the $i$th row and $j$th column of $\boldsymbol{X}$.

- For a vector, $\|\cdot\|_p$ denotes $L_p$ vector norm; for a matrix, $\|\cdot\|_p$ denotes the entry-wise $L_p$ matrix norm and $\|\cdot\|$ denotes the spectral norm.

- The Jacobian of a function $\boldsymbol{f} : \mathbb{R}^d \to \mathbb{R}^m$ at point $\boldsymbol{s}$ is a matrix $\boldsymbol{J}_{\boldsymbol{f}}(\boldsymbol{s})$ of partial derivatives such that $[\boldsymbol{J}_{\boldsymbol{f}}(\boldsymbol{s})]_{i,j} = \partial f_i / \partial s_j$; $\nabla f(\boldsymbol{s}) = [\partial f(\boldsymbol{s}) / \partial s_1 \ldots \partial f(\boldsymbol{s}) / \partial s_d]^\top$ denotes the gradient of scalar function $f : \mathbb{R}^d \to \mathbb{R}$; hence, $[\boldsymbol{J}_{\boldsymbol{f}}(\boldsymbol{s})]_{i:} = \nabla f_i(\boldsymbol{s})^\top$.

- $\mathrm{conv}\{\cdot\}$ denotes the convex hull of a set of vectors.

- $\mathcal{B}_p(R)$ being the unweighted $L_p$-norm ball with radius $R > 0$, and $\mathcal{B}_p$ is shorthand for the unweighted unit $L_p$-norm ball (i.e., $R = 1$).

- $\mathcal{B}_1^{\boldsymbol{w}(\boldsymbol{s})} = \{\boldsymbol{y} \in \mathbb{R}^d : \sum_{k=1}^d w_k(\boldsymbol{s})^{-1} |y_k| \leq 1\}$ is an $L_1$-norm ball weighted by $1/w_1(\boldsymbol{s}), ..., 1/w_d(\boldsymbol{s}) > 0$.

- $\mathcal{E}(P)$ is the maximum volume inscribed ellipsoid (MVIE) of an origin-centered convex polytope $P$.

- Polar of a set $\mathcal{C}$ is $\mathcal{C}^* = \{\boldsymbol{x} \in \mathbb{R}^d : \boldsymbol{x}^\top \boldsymbol{y} \leq 1, \ \forall \boldsymbol{y} \in \mathcal{C}\}$.

- $\mathrm{bd}(P)$ is the boundary of a set $P$.

- $\mathrm{extr}(P)$ is the set of extreme points (i.e., vertices) of a polytope $P$.

- $\mathcal{P}_d$ is the set of $d \times d$ permutation matrices.

- $\overline{\Theta} = [\overline{\boldsymbol{\theta}}, \overline{\boldsymbol{\phi}}]$ is shorthand for the vector of parameters of parametrized encoder $\boldsymbol{f}_{\overline{\boldsymbol{\theta}}}$ and decoder $\boldsymbol{g}_{\overline{\boldsymbol{\phi}}}$.

- $V(\boldsymbol{s}; \overline{\Theta}) := \log\det(\boldsymbol{J}_{\boldsymbol{f}_{\overline{\boldsymbol{\theta}}}}(\boldsymbol{g}_{\overline{\boldsymbol{\phi}}}(\boldsymbol{x}))^\top \boldsymbol{J}_{\boldsymbol{f}_{\overline{\boldsymbol{\theta}}}}(\boldsymbol{g}_{\overline{\boldsymbol{\phi}}}(\boldsymbol{x})))$ is shorthand for the log-determinant of Jacobian matrix of $\overline{\Theta} := [\overline{\boldsymbol{\theta}}, \overline{\boldsymbol{\phi}}]$, evaluated at $\boldsymbol{x} = \boldsymbol{f}(\boldsymbol{s})$; this is used as the Jacobian volume surrogate in J-VolMax objective (7a).

### A.2  Background on Convex Geometry

In this section, we briefly introduce the key notions in convex geometry that will be used in the paper.

**Convex Hull.** Given a set $S \subset \mathbb{R}^d$, its convex hull $\mathrm{conv}\{S\}$ is the set of all convex combinations of vectors in $S$, i.e.,

$$\mathrm{conv}\{S\} = \left\{ \sum_{i=1}^m \alpha_i \boldsymbol{x}_i : \boldsymbol{x}_i \in S, \alpha_i \geq 0, \sum_{i=1}^m \alpha_i = 1 \right\}.$$

**Maximum Volume Inscribed Ellipsoid (MVIE).** An MVIE (sometimes called *John ellipsoid*) of a bounded convex set $C$ is the ellipsoid of maximum volume that lies inside $C$, which can be represented as

$$\mathcal{E}(C) = \{\boldsymbol{B}\boldsymbol{u} + \boldsymbol{d} : \|\boldsymbol{u}\|_2 \leq 1\}$$

for $\boldsymbol{B} \in \mathbb{R}^{d \times d}, \boldsymbol{d} \in \mathbb{R}^d$, and $\boldsymbol{B}$ is a symmetric positive-definite matrix (i.e., $\boldsymbol{B} \in \mathbb{S}_{++}^d$). $\mathcal{E}(C)$ can be obtained from solving

$$\max_{\boldsymbol{B} \in \mathbb{S}_{++}^d, \boldsymbol{d} \in \mathbb{R}^d} \log\det \boldsymbol{B} \quad \text{s.t.} \quad \sup_{\boldsymbol{u}:\|\boldsymbol{u}\|_2 \leq 1} I_C(\boldsymbol{B}\boldsymbol{u} + \boldsymbol{d}) \leq 0,$$

with $I_C(\cdot)$ being the indicator function of whether a point is in $C$. Note that every non-empty bounded convex set has an unique MVIE. See [67, Section 8.4.2] for more details. Note that the MVIE has the following linear scaling property:

**Proposition A.1.** *If $T : \mathbb{R}^d \mapsto \mathbb{R}^d$ is an invertible linear map and $C$ is a non-empty compact convex set, then the MVIE of $T(C)$ is the image under $T$ of the MVIE of $C$, i.e.*

$$\mathcal{E}(T(C)) = T(\mathcal{E}(C)).$$

*Proof.* See [67, Section 8.4.3]. $\qquad\square$

In the following, we give examples of MVIEs of several sets that are relevant to our context.

**Proposition A.2.** *The followings are true about a convex set and its MVIE:*

1. *The MVIE of an $L_1$-norm ball $\mathcal{B}_1(R)$ in $\mathbb{R}^d$ is $\{\boldsymbol{x} : ||\boldsymbol{x}||_2 \leq \frac{R}{\sqrt{d}}\}$;*

2. *The MVIE of an $L_\infty$-norm ball $\mathcal{B}_\infty(R)$ in $\mathbb{R}^d$ is $\{\boldsymbol{x} : ||\boldsymbol{x}||_2 \leq R\}$;*

3. *The MVIE of an $L_2$-norm ball $\mathcal{B}_2(R)$ is itself.*

**Polytope.** Intuitively, a polytope $P$ is a bounded geometric object with flat sides (called *facets*). A polytope $P$ containing the origin can be represented either by intersections of $f$ half-spaces as

$$P = \{\boldsymbol{x} \in \mathbb{R}^d : \boldsymbol{a}_i^\top \boldsymbol{x} \leq 1, i = 1, ..., f\},$$

or by convex hull of $k$ vertices as

$$P = \mathrm{conv}\{\boldsymbol{v}_1, ..., \boldsymbol{v}_k\}.$$

For examples, $L_p$-norm ball $\mathcal{B}_p$ and simplex $\Delta = \{\boldsymbol{x} \in \mathbb{R}^d : x_i \geq 0, \sum_{i=1}^d x_i = 1\}$ are polytopes.

**Polar Set.** The polar of a set $C \subset \mathbb{R}^d$ is

$$C^* = \{\boldsymbol{x} \in \mathbb{R}^d : \boldsymbol{x}^\top \boldsymbol{y} \leq 1, \forall \boldsymbol{y} \in C\}.$$

There are some interesting properties of polar sets:

**Proposition A.3.** *For two sets $A, B \subset \mathbb{R}^d$ and $\alpha > 0$, the followings holds:*

1. *$A \subseteq B$ implies $B^* \subseteq A^*$,*

2. *$(A \cup B)^* = A^* \cap B^*$,*

3. *$(\alpha A)^* = \frac{1}{\alpha} A^*$.*

We give examples of polar sets of several sets that are related to our context:

**Proposition A.4.** *The followings are true about a set and its polar set:*

1. *Polar set of a weighted $L_1$-norm ball $\mathcal{B}_1^{\boldsymbol{w}} = \{\boldsymbol{y} \in \mathbb{R}^d : \sum_{k=1}^d w_k^{-1}|y_k| \leq 1\}$ is the reciprocally weighted $L_\infty$-norm ball $\mathcal{B}_\infty^{\boldsymbol{w}^{-1}} = \{\boldsymbol{y} \in \mathbb{R}^d : \max_{k=1,...,d}\{w_k|y_k|\} \leq 1\}$, and vice versa;*

2. *Polar set of a $L_2$-norm ball $\mathcal{B}_2(R)$ with radius $R > 0$ is a $L_2$-norm ball $\mathcal{B}_2(\frac{1}{R})$ with radius $\frac{1}{R}$, and vice versa.*

For the details of Proposition A.3 and Proposition A.4, we refer the readers to [68, Chapter 2].

# B   Proof of Theorem 3.2

The proof consists of two parts. In the first part, i.e., Lemma B.2, we show that any optimal solution of J-VolMax problem (7) must identify a ground-truth latent vector $\boldsymbol{s} \in \mathcal{S}$, up to $\boldsymbol{s}$-dependent permutation and invertible component-wise transformation. Then, in the second part, namely, Theorem 3.2, we leverage the continuity and full-rank structure of the Jacobian matrices at different $\boldsymbol{s}$ to argue that the permutation of latent components should be identical for all $\boldsymbol{s} \in \mathcal{S}$, instead of being $\boldsymbol{s}$-dependent as initially shown in Lemma B.2. That gives us the desired identifiability result.

Before we begin the proof, we invoke the following lemma from [30]:

**Lemma B.1.** *Let $\boldsymbol{H} \in \mathbb{R}^{d \times d}$ be a matrix satisfying*

$$\|\boldsymbol{H}^\top \boldsymbol{u}\|_2 \le \sqrt{d}, \ \forall \boldsymbol{u} \in \mathrm{extr}(\mathcal{B}_\infty).$$

*Then, $|\det \boldsymbol{H}| \le 1$, with equality occurs if and only if $\boldsymbol{H}$ is a real orthogonal matrix.*

*Proof.* See [30, Theorem 3]. □

**Lemma B.2.** *Denote any optimal solution of Problem* (7) *as $(\widehat{\boldsymbol{\theta}}, \widehat{\boldsymbol{\phi}})$. Assume $\widehat{\boldsymbol{f}} = \boldsymbol{f}_{\widehat{\boldsymbol{\theta}}}$ and $\widehat{\boldsymbol{g}} = \boldsymbol{g}_{\widehat{\boldsymbol{\phi}}}$ are universal function representers. Suppose the model in* (1) *and Assumption 3.1 hold every point $\boldsymbol{s} \in \mathcal{S}$. Then, the Jacobian at the point $\boldsymbol{s} \in \mathcal{S}$ of the function $\boldsymbol{h}^\star = \widehat{\boldsymbol{g}} \circ \boldsymbol{f}$ will satisfy*

$$\boldsymbol{J}_{\boldsymbol{h}^\star}(\boldsymbol{s}) = \boldsymbol{D}(\boldsymbol{s})\boldsymbol{\Pi}(\boldsymbol{s}),$$

*where $\boldsymbol{D}(\boldsymbol{s})$ is an invertible diagonal matrix with $[\boldsymbol{D}(\boldsymbol{s})]_{i,i}$ dependent on $i$-th component $s_i$ only, and $\boldsymbol{\Pi}(\boldsymbol{s})$ is a permutation matrix dependent on the point $\boldsymbol{s}$, almost everywhere.*

*Proof.* Define a differentiable function $\boldsymbol{h} : \mathcal{S} \mapsto \mathcal{S}$ as a composition $\boldsymbol{h} = \widehat{\boldsymbol{g}} \circ \boldsymbol{f}$. Our ultimate goal is to show that the Jacobian $\boldsymbol{J}_{\boldsymbol{h}}(\boldsymbol{s})$ has a permutation-like support structure, which is equivalent to the result statement.

We begin our analysis by considering a data point $\boldsymbol{x} = \boldsymbol{f}(\boldsymbol{s})$ for $\boldsymbol{s} \in \mathcal{S}$. Let $\widehat{\boldsymbol{s}} = \widehat{\boldsymbol{g}}(\boldsymbol{x})$. By definition of function $\boldsymbol{h}$,

$$\boldsymbol{f}(\boldsymbol{s}) = \boldsymbol{x} = \boldsymbol{f}_{\widehat{\boldsymbol{\theta}}}(\widehat{\boldsymbol{s}}) = \boldsymbol{f}_{\widehat{\boldsymbol{\theta}}}(\boldsymbol{g}_{\widehat{\boldsymbol{\phi}}}(\boldsymbol{x})) = \boldsymbol{f}_{\widehat{\boldsymbol{\theta}}}(\boldsymbol{h}(\boldsymbol{s})) \tag{16}$$

Then, by chain rule,

$$\boldsymbol{J}_{\boldsymbol{f}}(\boldsymbol{s}) = \boldsymbol{J}_{\boldsymbol{f}_{\widehat{\boldsymbol{\theta}}}}(\boldsymbol{h}(\boldsymbol{s}))\boldsymbol{J}_{\boldsymbol{h}}(\boldsymbol{s}) = \boldsymbol{J}_{\boldsymbol{f}_{\widehat{\boldsymbol{\theta}}}}(\widehat{\boldsymbol{s}})\boldsymbol{J}_{\boldsymbol{h}}(\boldsymbol{s}). \tag{17}$$

We now argue that $\boldsymbol{J}_{\boldsymbol{h}}(\boldsymbol{s})$ is a full rank matrix. By definition of $\boldsymbol{h}$,

$$\boldsymbol{J}_{\boldsymbol{h}}(\boldsymbol{s}) = \boldsymbol{J}_{\boldsymbol{g}_{\widehat{\boldsymbol{\phi}}}}(\boldsymbol{x})\boldsymbol{J}_{\boldsymbol{f}}(\boldsymbol{s}). \tag{18}$$

By definition of the ground-truth mixing process, $\boldsymbol{f}$ is an injective function, implying that $\boldsymbol{J}_{\boldsymbol{f}}(\boldsymbol{s})$ is full rank. We now investigate $\boldsymbol{J}_{\boldsymbol{g}_{\widehat{\boldsymbol{\phi}}}}(\boldsymbol{x})$. Due to the data-fitting constraint $\boldsymbol{x} = \boldsymbol{f}_{\widehat{\boldsymbol{\theta}}}(\boldsymbol{g}_{\widehat{\boldsymbol{\phi}}}(\boldsymbol{x}))$ and the chain rule, we have

$$\boldsymbol{J}_{\boldsymbol{f}_{\widehat{\boldsymbol{\theta}}}}(\widehat{\boldsymbol{s}})\boldsymbol{J}_{\boldsymbol{g}_{\widehat{\boldsymbol{\phi}}}}(\boldsymbol{x}) = \boldsymbol{I}. \tag{19}$$

Note that $\boldsymbol{J}_{\boldsymbol{f}_{\widehat{\boldsymbol{\theta}}}}(\widehat{\boldsymbol{s}})$ is full rank, due to the optimality of the problem (7). Therefore, $\boldsymbol{J}_{\boldsymbol{g}_{\widehat{\boldsymbol{\phi}}}}(\boldsymbol{x})$ must also be full rank. As a result, $\boldsymbol{J}_{\boldsymbol{h}}(\boldsymbol{s})$ is full rank, and hence $\boldsymbol{h}$ is indeed a locally invertible function at $\boldsymbol{s}$, by the inverse function theorem [69, Theorem 6.26].

Now, we can rewrite (17) as

$$(\boldsymbol{J}_{\boldsymbol{h}}(\boldsymbol{s})^{-1})^\top \boldsymbol{J}_{\boldsymbol{f}}(\boldsymbol{s})^\top = \boldsymbol{J}_{\boldsymbol{f}_{\widehat{\boldsymbol{\theta}}}}(\widehat{\boldsymbol{s}})^\top. \tag{20}$$

By the inverse function theorem [69, Theorem 6.26] and the chain rule, we also have $\boldsymbol{J}_{\boldsymbol{h}}(\boldsymbol{s})^{-1} = \boldsymbol{J}_{\boldsymbol{h}^{-1}}(\widehat{\boldsymbol{s}})$, which results in

$$(\boldsymbol{J}_{\boldsymbol{h}^{-1}}(\widehat{\boldsymbol{s}}))^\top \boldsymbol{J}_{\boldsymbol{f}}(\boldsymbol{s})^\top = \boldsymbol{J}_{\boldsymbol{f}_{\widehat{\boldsymbol{\theta}}}}(\widehat{\boldsymbol{s}})^\top. \tag{21}$$

We note that there is an intrinsic scaling ambiguity of the Jacobian matrices here: with an diagonal matrix $\boldsymbol{D}_{\boldsymbol{w}}(\boldsymbol{s}) = \mathrm{diag}(1/w_1(\boldsymbol{s}), ..., 1/w_d(\boldsymbol{s}))$ whose entries are dependent on the unknown $\boldsymbol{w}(\boldsymbol{s})$, we have

$$\boldsymbol{J}_{\boldsymbol{f}_{\widehat{\boldsymbol{\theta}}}}(\widehat{\boldsymbol{s}})^\top = (\boldsymbol{J}_{\boldsymbol{h}^{-1}}(\widehat{\boldsymbol{s}}))^\top \boldsymbol{D}_{\boldsymbol{w}}(\boldsymbol{s})^{-1}\boldsymbol{D}_{\boldsymbol{w}}(\boldsymbol{s})\boldsymbol{J}_{\boldsymbol{f}}(\boldsymbol{s})^\top. \tag{22}$$

We use the shorthand $\boldsymbol{z}_i := \boldsymbol{D}_{\boldsymbol{w}}(\boldsymbol{s})\nabla f_i(\boldsymbol{s})$, which is the $i$-th column of $\boldsymbol{D}_{\boldsymbol{w}}(\boldsymbol{s})\boldsymbol{J}_{\boldsymbol{f}}(\boldsymbol{s})^\top$. Notice that $\boldsymbol{D}_{\boldsymbol{w}}(\boldsymbol{s})\boldsymbol{J}_{\boldsymbol{f}}(\boldsymbol{s})^\top$ is a matrix whose columns lie inside the unit $L_1$-norm ball $\mathcal{B}_1$, i.e.

$$\boldsymbol{z}_1, ..., \boldsymbol{z}_m \in \mathcal{B}_1. \tag{23}$$

By combining Proposition A.1 and Assumption 3.1, we have a rescaled version of the SDI condition:

$$\mathcal{B}_2(\tfrac{1}{\sqrt{d}}) \subset \text{conv}\{\boldsymbol{z}_1, ..., \boldsymbol{z}_m\} \subset \mathcal{B}_1, \text{ and} \tag{24}$$

$$\text{conv}\{\boldsymbol{z}_1, ..., \boldsymbol{z}_m\}^* \cap \text{bd}(\mathcal{B}_2(\sqrt{d})) = \text{extr}(\mathcal{B}_\infty). \tag{25}$$

To proceed, let $\boldsymbol{H} := (\boldsymbol{J}_{\boldsymbol{h}^{-1}}(\widehat{\boldsymbol{s}}))^\top \boldsymbol{D}_{\boldsymbol{w}}(\boldsymbol{s})^{-1}$. To show that $\boldsymbol{J}_{\boldsymbol{h}}(\boldsymbol{s})$ has the support structure of a permutation matrix as desired, we can prove that such structure occurs in $\boldsymbol{H}$. Now, we consider the constraint (7b),

$$||\boldsymbol{J}_{\boldsymbol{f}_{\widehat{\boldsymbol{\theta}}}}(\widehat{\boldsymbol{s}})_{i,:}||_1 \leq C, \forall i \in [m]. \tag{26}$$

Combining (26) with the equation (22), we have

$$\|\tfrac{1}{C}\boldsymbol{H}\boldsymbol{z}_i\|_1 \leq 1, \ \forall i = 1, ..., m. \tag{27}$$

Observe that (27) can be further rewritten as the following set of $2^d$ explicit inequalities over all possible signs of each entry of $\tfrac{1}{C}\boldsymbol{H}\boldsymbol{z}_i$:

$$\boldsymbol{z}_i^\top (\tfrac{1}{C}\boldsymbol{H}^\top \boldsymbol{u}) \leq 1, \forall \boldsymbol{u} \in \{\pm 1\}^d. \tag{28}$$

Note that $\{\pm 1\}^d = \text{extr}(\mathcal{B}_\infty)$. This allows us to reveal from (28) that

$$\boldsymbol{z}_i^\top (\tfrac{1}{C}\boldsymbol{H}^\top \boldsymbol{u}) \leq 1, \forall \boldsymbol{u} \in \text{extr}(\mathcal{B}_\infty). \tag{29}$$

Hence, by definition of the polar set of a polytope, we have

$$\tfrac{1}{C}\boldsymbol{H}^\top \boldsymbol{u} \in \text{conv}\{\boldsymbol{z}_1, ..., \boldsymbol{z}_m\}^*, \forall \boldsymbol{u} \in \text{extr}(\mathcal{B}_\infty). \tag{30}$$

To proceed, recall from the rescaled SDI assumption in (24) that $\mathcal{B}_2(\tfrac{1}{\sqrt{d}}) \subset \text{conv}\{\boldsymbol{z}_1, ..., \boldsymbol{z}_m\}$. By Proposition A.3 of polar sets,

$$\text{conv}\{\boldsymbol{z}_1, ..., \boldsymbol{z}_m\}^* \subset \mathcal{B}_2(\sqrt{d}). \tag{31}$$

Since $\tfrac{1}{C}\boldsymbol{H}^\top \boldsymbol{u} \in \text{conv}\{\boldsymbol{z}_1, ..., \boldsymbol{z}_m\}^*$ and $\text{conv}\{\boldsymbol{z}_1, ..., \boldsymbol{z}_m\}^* \subset \mathcal{B}_2(\sqrt{d})$, we have $\tfrac{1}{C}\boldsymbol{H}^\top \boldsymbol{u} \in \mathcal{B}_2(\sqrt{d})$, i.e. $\|\tfrac{1}{C}\boldsymbol{H}^\top \boldsymbol{u}\|_2 \leq \sqrt{d}$ for any $\boldsymbol{u} \in \text{extr}(\mathcal{B}_\infty)$. By Lemma B.1, we can see that

$$|\det \boldsymbol{H}| \leq C, \tag{32}$$

with equality holding if and only if $\tfrac{1}{C}\boldsymbol{H}$ is an orthogonal matrix. Here, we note that the J-VolMax criterion would lead to an $\boldsymbol{H}$ with maximal determinant, so then $|\det \boldsymbol{H}| = C$.

Since $\tfrac{1}{C}\boldsymbol{H}$ is an orthogonal matrix and $\|\boldsymbol{u}\|_2 = \sqrt{d}$ for any $\boldsymbol{u} \in \text{extr}(\mathcal{B}_\infty)$, we have $\tfrac{1}{C}\boldsymbol{H}^\top \boldsymbol{u} \in \text{bd}(\mathcal{B}_2(\sqrt{d}))$. Also, recall $\tfrac{1}{C}\boldsymbol{H}^\top \boldsymbol{u} \in \text{conv}\{\boldsymbol{z}_1, ..., \boldsymbol{z}_m\}^*$. Hence, $\tfrac{1}{C}\boldsymbol{H}^\top \boldsymbol{u} \in \text{conv}\{\boldsymbol{z}_1, ..., \boldsymbol{z}_m\}^* \cap \text{bd}(\mathcal{B}_2(\sqrt{d}))$. Since

$$\text{conv}\{\boldsymbol{z}_1, ..., \boldsymbol{z}_m\}^* \cap \text{bd}(\mathcal{B}_2(\sqrt{d})) = \text{extr}(\mathcal{B}_\infty), \tag{33}$$

we have

$$\tfrac{1}{C}\boldsymbol{H}^\top \boldsymbol{u} \in \text{extr}(\mathcal{B}_\infty), \ \forall \boldsymbol{u} \in \text{extr}(\mathcal{B}_\infty), \tag{34}$$

which implies

$$\|\tfrac{1}{C}\boldsymbol{H}_{:,i}\|_1 = 1, \ \forall i = 1, ..., d. \tag{35}$$

Hence, $|\det \boldsymbol{H}| = C$ if and only if both $\|\tfrac{1}{C}\boldsymbol{H}_{:,i}\|_1 = 1$ and $\tfrac{1}{C}\boldsymbol{H}$ is an orthogonal matrix. Then, we are able to conclude that

$$|\det \boldsymbol{H}| = C \tag{36}$$

if and only if $\tfrac{1}{C}\boldsymbol{H}$ is a signed permutation matrix. Equivalently,

$$\log |\det \boldsymbol{H}| \leq \log C \tag{37}$$

with equality holds if and only if $\frac{1}{C}\boldsymbol{H}$ is a signed permutation matrix. The arguments in Eqs. (26)-(37) are from analytical tools first developed in the PMF work [30]. Similar steps were used in [31, 33] for other models as well.

Suppose that there exists an optimal solution $(\overline{\boldsymbol{\theta}}, \overline{\boldsymbol{\phi}})$ with some estimated source $\overline{\boldsymbol{s}} = \boldsymbol{g}_{\overline{\boldsymbol{\phi}}}(\boldsymbol{x})$ such that the mapping $\overline{\boldsymbol{h}} = \boldsymbol{g}_{\overline{\boldsymbol{\phi}}} \circ \boldsymbol{f}$ is not a composition of a permutation and an element-wise invertible transformation with strictly positive probability, i.e., for $\overline{\boldsymbol{H}} = (\boldsymbol{J}_{\overline{\boldsymbol{h}}^{-1}}(\overline{\boldsymbol{s}}))^{\top} \boldsymbol{D}_{\boldsymbol{w}}(\boldsymbol{s})^{-1}$, we have

$$\mathbb{P}(\log|\det \overline{\boldsymbol{H}}| < \log C) > 0, \tag{38}$$

where the probability is over $p_{\boldsymbol{s}}$. This implies

$$\mathbb{E}[\log\det(\boldsymbol{J}_{\boldsymbol{f}_{\overline{\boldsymbol{\theta}}}}(\overline{\boldsymbol{s}})^{\top}\boldsymbol{J}_{\boldsymbol{f}_{\overline{\boldsymbol{\theta}}}}(\overline{\boldsymbol{s}}))] \tag{39}$$

$$= \mathbb{E}[\log\det(\boldsymbol{J}_{\overline{\boldsymbol{h}}^{-1}}(\overline{\boldsymbol{s}})^{\top}\boldsymbol{J}_{\boldsymbol{f}}(\boldsymbol{s})^{\top}\boldsymbol{J}_{\boldsymbol{f}}(\boldsymbol{s})\boldsymbol{J}_{\overline{\boldsymbol{h}}^{-1}}(\overline{\boldsymbol{s}}))] \tag{40}$$

$$= \mathbb{E}[\log\det(\boldsymbol{J}_{\overline{\boldsymbol{h}}^{-1}}(\overline{\boldsymbol{s}})^{\top}\boldsymbol{D}_{\boldsymbol{w}}(\boldsymbol{s})^{-1}\boldsymbol{D}_{\boldsymbol{w}}(\boldsymbol{s})\boldsymbol{J}_{\boldsymbol{f}}(\boldsymbol{s})^{\top}\boldsymbol{J}_{\boldsymbol{f}}(\boldsymbol{s})\boldsymbol{D}_{\boldsymbol{w}}(\boldsymbol{s})\boldsymbol{D}_{\boldsymbol{w}}(\boldsymbol{s})^{-1}\boldsymbol{J}_{\overline{\boldsymbol{h}}^{-1}}(\overline{\boldsymbol{s}}))] \tag{41}$$

$$= \mathbb{E}[\log\det(\overline{\boldsymbol{H}}\boldsymbol{D}_{\boldsymbol{w}}(\boldsymbol{s})\boldsymbol{J}_{\boldsymbol{f}}(\boldsymbol{s})^{\top}\boldsymbol{J}_{\boldsymbol{f}}(\boldsymbol{s})\boldsymbol{D}_{\boldsymbol{w}}(\boldsymbol{s})\overline{\boldsymbol{H}}^{\top})] \tag{42}$$

$$= \mathbb{E}[\log(\det(\overline{\boldsymbol{H}})^2\det(\boldsymbol{D}_{\boldsymbol{w}}(\boldsymbol{s})\boldsymbol{J}_{\boldsymbol{f}}(\boldsymbol{s})^{\top}\boldsymbol{J}_{\boldsymbol{f}}(\boldsymbol{s})\boldsymbol{D}_{\boldsymbol{w}}(\boldsymbol{s})))] \tag{43}$$

$$= \mathbb{E}[2\log|\det\overline{\boldsymbol{H}}| + \log\det(\boldsymbol{D}_{\boldsymbol{w}}(\boldsymbol{s})\boldsymbol{J}_{\boldsymbol{f}}(\boldsymbol{s})^{\top}\boldsymbol{J}_{\boldsymbol{f}}(\boldsymbol{s})\boldsymbol{D}_{\boldsymbol{w}}(\boldsymbol{s}))] \tag{44}$$

$$\overset{(a)}{=} \int_{\boldsymbol{s}\in\mathcal{A}} p(\boldsymbol{s})[2\log|\det\overline{\boldsymbol{H}}| + \log\det(\boldsymbol{D}_{\boldsymbol{w}}(\boldsymbol{s})\boldsymbol{J}_{\boldsymbol{f}}(\boldsymbol{s})^{\top}\boldsymbol{J}_{\boldsymbol{f}}(\boldsymbol{s})\boldsymbol{D}_{\boldsymbol{w}}(\boldsymbol{s}))]d\boldsymbol{s} \tag{45}$$

$$+ \int_{\boldsymbol{s}\in\mathcal{S}\backslash\mathcal{A}} p(\boldsymbol{s})[2\log|\det\overline{\boldsymbol{H}}| + \log\det(\boldsymbol{D}_{\boldsymbol{w}}(\boldsymbol{s})\boldsymbol{J}_{\boldsymbol{f}}(\boldsymbol{s})^{\top}\boldsymbol{J}_{\boldsymbol{f}}(\boldsymbol{s})\boldsymbol{D}_{\boldsymbol{w}}(\boldsymbol{s}))]d\boldsymbol{s} \tag{46}$$

$$\overset{(b)}{<} \int_{\boldsymbol{s}\in\mathcal{A}} p(\boldsymbol{s})[2\log C + \log\det(\boldsymbol{D}_{\boldsymbol{w}}(\boldsymbol{s})\boldsymbol{J}_{\boldsymbol{f}}(\boldsymbol{s})^{\top}\boldsymbol{J}_{\boldsymbol{f}}(\boldsymbol{s})\boldsymbol{D}_{\boldsymbol{w}}(\boldsymbol{s}))]d\boldsymbol{s} \tag{47}$$

$$+ \int_{\boldsymbol{s}\in\mathcal{S}\backslash\mathcal{A}} p(\boldsymbol{s})[2\log C + \log\det(\boldsymbol{D}_{\boldsymbol{w}}(\boldsymbol{s})\boldsymbol{J}_{\boldsymbol{f}}(\boldsymbol{s})^{\top}\boldsymbol{J}_{\boldsymbol{f}}(\boldsymbol{s})\boldsymbol{D}_{\boldsymbol{w}}(\boldsymbol{s}))]d\boldsymbol{s} \tag{48}$$

$$= \int_{\boldsymbol{s}\in\mathcal{S}} p(\boldsymbol{s})[2\log C + \log\det(\boldsymbol{D}_{\boldsymbol{w}}(\boldsymbol{s})\boldsymbol{J}_{\boldsymbol{f}}(\boldsymbol{s})^{\top}\boldsymbol{J}_{\boldsymbol{f}}(\boldsymbol{s})\boldsymbol{D}_{\boldsymbol{w}}(\boldsymbol{s}))]d\boldsymbol{s} \tag{49}$$

$$= \mathbb{E}[2\log C + \log\det(\boldsymbol{D}_{\boldsymbol{w}}(\boldsymbol{s})\boldsymbol{J}_{\boldsymbol{f}}(\boldsymbol{s})^{\top}\boldsymbol{J}_{\boldsymbol{f}}(\boldsymbol{s})\boldsymbol{D}_{\boldsymbol{w}}(\boldsymbol{s}))], \tag{50}$$

$$= \mathbb{E}[\log(C^2\det(\boldsymbol{D}_{\boldsymbol{w}}(\boldsymbol{s})\boldsymbol{J}_{\boldsymbol{f}}(\boldsymbol{s})^{\top}\boldsymbol{J}_{\boldsymbol{f}}(\boldsymbol{s})\boldsymbol{D}_{\boldsymbol{w}}(\boldsymbol{s})))], \tag{51}$$

where (a) and (b) involve two sets $\mathcal{A} = \{\boldsymbol{s} : \log|\det\overline{\boldsymbol{H}}| < C\}$ and $\mathcal{S}\backslash\mathcal{A} = \{\boldsymbol{s} : \log|\det\overline{\boldsymbol{H}}| = C\}$.

Consider an invertible continuous element-wise function $\tilde{\boldsymbol{h}} : \mathcal{S} \mapsto \mathcal{S}$ with $\tilde{\boldsymbol{s}} := \tilde{\boldsymbol{h}}^{-1}(\boldsymbol{s}) \in \mathcal{S}$, $\tilde{\boldsymbol{h}}(\tilde{\boldsymbol{s}}) = \boldsymbol{s}$, and $\boldsymbol{J}_{\tilde{\boldsymbol{h}}}(\tilde{\boldsymbol{s}}) = \boldsymbol{J}_{\tilde{\boldsymbol{h}}}(\tilde{\boldsymbol{h}}^{-1}(\boldsymbol{s})) = C\boldsymbol{D}_{\boldsymbol{w}}(\boldsymbol{s})$. This function $\tilde{\boldsymbol{h}}$ merely rescales and maps each latent component by an element-wise invertible transformation. Define

$$\tilde{\boldsymbol{f}}(\tilde{\boldsymbol{s}}) := \boldsymbol{f}(\tilde{\boldsymbol{h}}(\tilde{\boldsymbol{s}})), \tag{52}$$

whose Jacobian is

$$\boldsymbol{J}_{\tilde{\boldsymbol{f}}}(\tilde{\boldsymbol{s}}) = \boldsymbol{J}_{\boldsymbol{f}}(\boldsymbol{s})\boldsymbol{J}_{\tilde{\boldsymbol{h}}}(\tilde{\boldsymbol{s}}). \tag{53}$$

We show that $\tilde{\boldsymbol{f}}(\tilde{\boldsymbol{s}})$ is a feasible solution to J-VolMax, i.e. satisfying (7b) and (7c). First, note that

$$\|\boldsymbol{J}_{\tilde{\boldsymbol{f}}}(\tilde{\boldsymbol{s}})_{i,:}\|_1 = \|C\boldsymbol{D}_{\boldsymbol{w}}(\boldsymbol{s})\nabla f_i(\boldsymbol{s})\|_1 \leq C. \tag{54}$$

Second, observe that

$$\tilde{\boldsymbol{f}}(\tilde{\boldsymbol{s}}) = \boldsymbol{f}(\tilde{\boldsymbol{h}}(\tilde{\boldsymbol{s}})) = \boldsymbol{f}(\tilde{\boldsymbol{h}}(\tilde{\boldsymbol{h}}^{-1}(\boldsymbol{s}))) = \boldsymbol{f}(\boldsymbol{s}) = \boldsymbol{x}. \tag{55}$$

Hence, there exists a feasible solution $\boldsymbol{x} = \tilde{\boldsymbol{f}}(\tilde{\boldsymbol{s}})$ to J-VolMax (7) such that

$$\mathbb{E}[\log\det(\boldsymbol{J}_{\boldsymbol{f}_{\overline{\boldsymbol{\theta}}}}(\overline{\boldsymbol{s}})^{\top}\boldsymbol{J}_{\boldsymbol{f}_{\overline{\boldsymbol{\theta}}}}(\overline{\boldsymbol{s}}))] < \mathbb{E}[\log\det(\boldsymbol{J}_{\tilde{\boldsymbol{f}}}(\tilde{\boldsymbol{s}})^{\top}\boldsymbol{J}_{\tilde{\boldsymbol{f}}}(\tilde{\boldsymbol{s}}))]. \tag{56}$$

This contradicts the optimality of $(\overline{\boldsymbol{\theta}}, \overline{\boldsymbol{\phi}})$ to J-VolMax problem in (7). We hence conclude that any optimal solution $(\boldsymbol{\theta}^{\star}, \boldsymbol{\phi}^{\star})$ of (7) would have the composite function $\boldsymbol{h}^{\star} = \boldsymbol{g}_{\boldsymbol{\phi}^{\star}} \circ \boldsymbol{f}$ satisfying

$$\boldsymbol{J}_{\boldsymbol{h}^{\star}}(\boldsymbol{s}) = \boldsymbol{D}(\boldsymbol{s})\boldsymbol{\Pi}(\boldsymbol{s}) \text{ a.e.}, \tag{57}$$

where $\boldsymbol{D}(\boldsymbol{s})$ is an invertible diagonal matrix such that with the $i^{th}$ diagonal entries dependent on $s_i$ due to the definition of a Jacobian matrix, and $\boldsymbol{\Pi}(\boldsymbol{s})$ is a permutation matrix that depends on $\boldsymbol{s}$. $\qquad\square$

**Theorem 3.2** (Identifiability of J-VolMax). *Denote any optimal solution of Problem* (7) *as* $(\widehat{\boldsymbol{\theta}}, \widehat{\boldsymbol{\phi}})$. *Assume* $\widehat{\boldsymbol{f}} = \boldsymbol{f}_{\widehat{\boldsymbol{\theta}}}$ *and* $\widehat{\boldsymbol{g}} = \boldsymbol{g}_{\widehat{\boldsymbol{\phi}}}$ *are universal function representers. Suppose the model in* (1) *and Assumption 3.1 hold for every* $\boldsymbol{s} \in \mathcal{S}$. *Then, we have* $\widehat{\boldsymbol{s}} = \widehat{\boldsymbol{g}}(\boldsymbol{x}) = \widehat{\boldsymbol{g}} \circ \boldsymbol{f}(\boldsymbol{s})$ *where*

$$[\widehat{\boldsymbol{s}}]_i = [\widehat{\boldsymbol{g}}(\boldsymbol{x})]_i = \rho_i(s_{\boldsymbol{\pi}(i)}), \ \forall i \in [d], \tag{8}$$

*in which* $\boldsymbol{\pi}$ *is a permutation of* $\{1, \ldots, d\}$ *and* $\rho_i(\cdot) : \mathbb{R} \to \mathbb{R}$ *is an invertible function.*

*Proof of Theorem 3.2.* Define a differentiable function $\boldsymbol{h}^\star : \mathcal{S} \mapsto \mathcal{S}$ as a composition $\boldsymbol{h} = \widehat{\boldsymbol{g}} \circ \boldsymbol{f}$. By Lemma B.2, we have that the Jacobian at the point $\boldsymbol{s} \in \mathcal{S}$ of the function $\boldsymbol{h}^\star = \widehat{\boldsymbol{g}} \circ \boldsymbol{f}$ will satisfy

$$\boldsymbol{J}_{\boldsymbol{h}^\star}(\boldsymbol{s}) = \boldsymbol{D}(\boldsymbol{s})\boldsymbol{\Pi}(\boldsymbol{s}), \tag{58}$$

where $\boldsymbol{D}(\boldsymbol{s})$ is an invertible diagonal matrix with $[\boldsymbol{D}(\boldsymbol{s})]_{i,i}$ dependent on $i$-th component $s_i$ only, and $\boldsymbol{\Pi}(\boldsymbol{s})$ is a permutation matrix dependent on the point $\boldsymbol{s}$, *almost everywhere*.

First, we want to leverage continuity of $\boldsymbol{h}^\star$ to show that $\boldsymbol{J}_{\boldsymbol{h}^\star}(\boldsymbol{s}) = \boldsymbol{D}(\boldsymbol{s})\boldsymbol{\Pi}(\boldsymbol{s})$ actually holds everywhere on $\mathcal{S}$. This is because if there exists $\overline{\boldsymbol{s}} \in \mathcal{S}$ such that $\boldsymbol{J}_{\boldsymbol{h}^\star}(\overline{\boldsymbol{s}}) \neq \boldsymbol{D}(\overline{\boldsymbol{s}})\boldsymbol{\Pi}(\overline{\boldsymbol{s}})$, then due to the full-rank property of $\boldsymbol{J}_{\boldsymbol{h}^\star}(\overline{\boldsymbol{s}})$, there exist a $j \in [d]$ such that the column $[\boldsymbol{J}_{\boldsymbol{h}^\star}(\overline{\boldsymbol{s}})]_{:j}$ has at least two non-zero elements. However, the continuity of the Jacobian $\boldsymbol{J}_{\boldsymbol{h}^\star}$ implies that there exists an open neighborhood of $\overline{\boldsymbol{s}}$, where the column $[\boldsymbol{J}_{\boldsymbol{h}^\star}(\overline{\boldsymbol{s}})]_{:j}$ has at least two non-zero elements. This contradicts the fact that $\boldsymbol{J}_{\boldsymbol{h}^\star}(\boldsymbol{s}) = \boldsymbol{D}(\boldsymbol{s})\boldsymbol{\Pi}(\boldsymbol{s})$ almost everywhere.

Next, it remains to show that $\boldsymbol{\Pi}(\boldsymbol{s})$ is constant (say $\boldsymbol{\Pi} \in \mathcal{P}_d$) for all $\boldsymbol{s} \in \mathcal{S}$. The reason is that if the non-zero elements in $\boldsymbol{\Pi}(\boldsymbol{s})$ switched locations, then there would exist a point in $\overline{\boldsymbol{s}} \in \mathcal{S}$ where $\boldsymbol{J}_{\boldsymbol{h}^\star}(\overline{\boldsymbol{s}})$ is singular, which contradicts the full-rank property of $\boldsymbol{J}_{\boldsymbol{h}^\star}(\overline{\boldsymbol{s}})$. Hence, $\boldsymbol{\Pi}(\boldsymbol{s}) = \boldsymbol{\Pi}, \forall \boldsymbol{s} \in \mathcal{S}$; see a similar argument in [12, Theorem 1].

Finally, let $\boldsymbol{\pi}$ denote the permutation of $\{1, \ldots, d\}$ corresponding to the permutation matrix $\boldsymbol{\Pi}$, i.e., let $\boldsymbol{r} = [1, 2, \ldots, d]^\top$, then $\boldsymbol{\pi}(i) = \boldsymbol{\Pi}_{i:}\boldsymbol{r}$. Then, $\boldsymbol{J}_{\boldsymbol{h}^\star}(\boldsymbol{s}) = \boldsymbol{D}(\boldsymbol{s})\boldsymbol{\Pi}$ implies that

$$\frac{\partial h_i^\star(\boldsymbol{s})}{\partial s_{\boldsymbol{\pi}(i)}} = \frac{\partial \widehat{s}_i}{\partial s_{\boldsymbol{\pi}(i)}} \neq 0 \quad \text{and} \quad \frac{\partial \widehat{s}_i}{\partial s_{\boldsymbol{\pi}(j)}} = 0, \forall j \neq i$$

Hence, there exist scalar functions $\rho_1, \ldots, \rho_d$ such that

$$\widehat{s}_i = \rho_i(s_{\boldsymbol{\pi}(i)}).$$

Note that $\rho_1, \ldots, \rho_d$ are invertible functions by the invertibility of $\boldsymbol{h}^\star$, which was shown in Lemma B.2. In conclusion, the estimated unmixer $\boldsymbol{g}_{\boldsymbol{\phi}^\star}$ satisfies $\boldsymbol{g}_{\boldsymbol{\phi}^\star} \circ \boldsymbol{f}$ being an invertible element-wise transformation and permutation. $\square$

## C  Proof of Theorem 3.3

Before diving into proving Theorem 3.3, we first show three cornerstone lemmas of the proof. These lemmas help characterize the detailed conditions under which an optimal solution of J-VolMax criterion can recover the latent components up to a constant permutation and invertible element-wise transformations, i.e.,

$$\widehat{\boldsymbol{g}}(\boldsymbol{x}^{(n)}) = \widehat{\boldsymbol{\Pi}}\widehat{\boldsymbol{\rho}}(\boldsymbol{s}^{(n)}), \forall \boldsymbol{s}^{(n)} \in \mathcal{S}_N. \tag{59}$$

The formal result is given in Lemma C.3. Intuitively, (59) holds if the points $\boldsymbol{s}^{(n)}$ in set $\mathcal{S}_N$ are sampled densely and closely enough from $p_{\boldsymbol{s}}$ with a sufficiently large $N$, together with some regularity conditions on the ground-truth and the learnable function classes. To show (59), we employ a three-step argument, Lemma C.1, Lemma C.2, and Lemma C.3.

Firstly, we show that for any feasible solution of J-VolMax that attains maximal Jacobian volume at each $\boldsymbol{s}^{(n)} \in \mathcal{S}_N$, it must give an estimated latent component vector that is permutation and invertible element-wise transformation of the ground truth $\boldsymbol{s}^{(n)}$. In fact, due to continuity, this holds at every point $\boldsymbol{s}$ in an union of neighborhoods $U_N = \cup_{n=1}^N U^{(n)}$ centered on each $\boldsymbol{s}^{(n)}$. The result is stated in the following Lemma C.1, and proof is a straightforward adaptation of Lemma B.2.

**Lemma C.1.** *Denote any feasible solution of J-VolMax problem (7) as $\overline{\Theta} := [\overline{\theta}, \overline{\phi}]$, learned from classes of learnable encoders and decoders $\mathcal{F}, \mathcal{G}$ that include the function classes of ground-truth encoders and decoders $\mathcal{F}', \mathcal{G}'$. Assume that there is an unknown finite set $\mathcal{S}_N := \{s^{(1)}, ..., s^{(N)}\} \subset \mathcal{S}$ with unknown $\mathcal{X}_N := \{x^{(n)} \in \mathcal{X} : x^{(n)} = f(s^{(n)}), \forall s^{(n)} \in \mathcal{S}_N\} \subset \mathcal{X}$ such that the Assumption 3.1 is satisfied at each of the $N$ points in $\mathcal{S}_N$. If the estimated latent components $\overline{s}^{(n)} = g_{\overline{\phi}}(x^{(n)})$ are not permutation and invertible element-wise transformation of ground-truth $s^{(n)}$, then there exists another feasible solution with parameters $\tilde{\Theta}$ such that*

$$V(s^{(n)}; \overline{\Theta}) < V(s^{(n)}; \tilde{\Theta}), \ \forall s^{(n)} \in \mathcal{S}_N \tag{60}$$

*Furthermore, there exists an union $U_N = \cup_{n=1}^N U^{(n)} \subseteq \mathcal{S}$ of open ball neighborhoods $U^{(n)} = \{s \in \mathcal{S} : ||s - s^{(n)}||_2 < d^{(n)}\}$ centered on $s^{(n)} \in \mathcal{S}_N$ such that*

$$V(s; \overline{\Theta}) < V(s; \tilde{\Theta}), \ \forall s \in U_N. \tag{61}$$

*Proof.* For the feasible solution $\overline{f} = f_{\overline{\theta}}$ and $\overline{g} = g_{\overline{\phi}}$, define a differentiable function $\overline{h} : \mathcal{S} \mapsto \mathcal{S}$ as a composition $\overline{h} = \overline{g} \circ f$, and consider a data point $x^{(n)} = f(s^{(n)}) \in \mathcal{X}_N$ for $s^{(n)} \in \mathcal{S}_N$. Let $\overline{s}^{(n)} = \overline{g}(x^{(n)})$. Following the same argument (16)-(36) as in Lemma B.2, for $\overline{H} := (J_{\overline{h}^{-1}}(\overline{s}^{(n)})^\top D_w(s^{(n)}))^{-1}$, we can similarly derive that $\log|\det \overline{H}| \leq \log C$ at any point $s^{(n)} \in \mathcal{S}_N$, with equality holds if and only if $\frac{1}{C}\overline{H}$ is a signed permutation matrix, i.e. when $\overline{h}$ is a composition of a permutation and invertible element-wise transformations.

We will show in the following that if $\overline{h}$ is not a composition of a permutation and an element-wise invertible transformation, then the Jacobian volume $V(s^{(n)}; \overline{\Theta})$ attained at $s^{(n)}$ of the feasible solution $\overline{\Theta}$ is strictly smaller than the Jacobian volume of another feasible solution; hence, the feasible solution $\overline{\Theta}$ does not reach maximal Jacobian volume at $s^{(n)}$.

Since $\overline{h} = \overline{g} \circ f$ is not a composition of permutation and component-wise invertible mappings,

$$\log|\det \overline{H}| < \log C, \ \forall s^{(n)} \in \mathcal{S}_N. \tag{62}$$

This implies

$$\log \det(J_{f_{\overline{\theta}}}(\overline{s}^{(n)})^\top J_{f_{\overline{\theta}}}(\overline{s}^{(n)}))$$
$$= \log \det(J_{\overline{h}^{-1}}(\overline{s}^{(n)})^\top J_f(s^{(n)})^\top J_f(s^{(n)}) J_{\overline{h}^{-1}}(\overline{s}^{(n)}))$$
$$= \log \det(J_{\overline{h}^{-1}}(\overline{s}^{(n)})^\top D_w(s^{(n)})^{-1} D_w(s^{(n)}) J_f(s^{(n)})^\top J_f(s^{(n)}) D_w(s^{(n)}) D_w(s^{(n)})^{-1} J_{\overline{h}^{-1}}(\overline{s}^{(n)}))$$
$$= \log \det(\overline{H} D_w(s^{(n)}) J_f(s^{(n)})^\top J_f(s) D_w(s^{(n)}) \overline{H}^\top)$$
$$= \log[(\det \overline{H})^2 \det(D_w(s^{(n)}) J_f(s^{(n)})^\top J_f(s^{(n)}) D_w(s^{(n)}))]$$
$$< \log(C^2 \det(D_w(s^{(n)}) J_f(s^{(n)})^\top J_f(s^{(n)}) D_w(s^{(n)}))), \ \forall s^{(n)} \in \mathcal{S}_N. \tag{63}$$

Consider an invertible continuous element-wise function $\tilde{h} : \mathcal{S} \mapsto \mathcal{S}$ with $\tilde{s}^{(n)} := \tilde{h}^{-1}(s^{(n)}) \in \mathcal{S}$, $\tilde{h}(\tilde{s}^{(n)}) = s^{(n)}$, and $J_{\tilde{h}}(\tilde{s}^{(n)}) = J_{\tilde{h}}(\tilde{h}^{-1}(s^{(n)})) = CD_w(s^{(n)})$. This function $\tilde{h}$ merely rescales and maps each latent component by an element-wise invertible transformation. Define

$$\tilde{f}(\tilde{s}^{(n)}) := f(\tilde{h}(\tilde{s}^{(n)})), \tag{64}$$

whose Jacobian is

$$J_{\tilde{f}}(\tilde{s}^{(n)}) = J_f(s^{(n)}) J_{\tilde{h}}(\tilde{s}^{(n)}). \tag{65}$$

We show that $\tilde{f}(\tilde{s}^{(n)})$ is a feasible solution to J-VolMax (7), i.e. (7b) and (7c) hold. First, note that

$$||J_{\tilde{f}}(\tilde{s}^{(n)})_{i,:}||_1 = ||CD_w(s^{(n)})\nabla f_i(s^{(n)})||_1 \leq C, \ \forall i \in [m]. \tag{66}$$

Second, observe that

$$\tilde{f}(\tilde{s}^{(n)}) = f(\tilde{h}(\tilde{s}^{(n)})) = f(\tilde{h}(\tilde{h}^{-1}(s^{(n)}))) = f(s^{(n)}) = x^{(n)}. \tag{67}$$

Hence, there exists a feasible solution $\boldsymbol{x} = \tilde{\boldsymbol{f}}(\tilde{\boldsymbol{s}}^{(n)})$ to J-VolMax such that

$$\log \det(\boldsymbol{J}_{\boldsymbol{f}_{\overline{\boldsymbol{\theta}}}}(\overline{\boldsymbol{s}}^{(n)})^\top \boldsymbol{J}_{\boldsymbol{f}_{\overline{\boldsymbol{\theta}}}}(\overline{\boldsymbol{s}}^{(n)})) < \log \det(\boldsymbol{J}_{\tilde{\boldsymbol{f}}}(\tilde{\boldsymbol{s}}^{(n)})^\top \boldsymbol{J}_{\tilde{\boldsymbol{f}}}(\tilde{\boldsymbol{s}}^{(n)})), \ \forall \boldsymbol{s}^{(n)} \in \mathcal{S}_N. \tag{68}$$

Equivalently, for $\tilde{\Theta} = [\tilde{\boldsymbol{\theta}}, \tilde{\boldsymbol{\phi}}]$ being the parameters corresponding to $\tilde{\boldsymbol{f}}$ and $\tilde{\boldsymbol{s}}^{(n)}$,

$$V(\boldsymbol{s}^{(n)}; \overline{\Theta}) < V(\boldsymbol{s}^{(n)}; \tilde{\Theta}), \ \forall \boldsymbol{s}^{(n)} \in \mathcal{S}_N. \tag{69}$$

Finally, due to continuity of the involved functions, there exists a neighborhood $U^{(n)} = \{\boldsymbol{s} \in \mathcal{S} : ||\boldsymbol{s} - \boldsymbol{s}^{(n)}||_2 < d^{(n)}\}$ centered at $\boldsymbol{s}^{(n)}$ such that

$$V(\boldsymbol{s}; \overline{\Theta}) < V(\boldsymbol{s}; \tilde{\Theta}), \ \forall \boldsymbol{s} \in U^{(n)}, \tag{70}$$

which holds for all $U^{(1)}, ..., U^{(n)}$. Hence, with $U_N = \cup_{n=1}^{N} U^{(n)}$,

$$V(\boldsymbol{s}; \overline{\Theta}) < V(\boldsymbol{s}; \tilde{\Theta}), \ \forall \boldsymbol{s} \in U_N. \tag{71}$$

$\square$

Following Lemma C.1, we show that if the SDI-satisfying finite set $\mathcal{S}_N$ are sampled dense enough such that $U_N$ covers a significant part of $\mathcal{S}$, then any optimal solution of J-VolMax criterion must recover the ground-truth latent components $\boldsymbol{s}^{(n)}$, up to $\boldsymbol{s}^{(n)}$-dependent permutation and invertible element-wise transformations. The proof leverages the result from the previous Lemma C.1 to show a contradiction that if a feasible solution $\overline{\Theta}$ does not give us ground-truth latents up to aforementioned ambiguities, there must exist another feasible solution $\tilde{\Theta}$ that is strictly better than $\overline{\Theta}$ in terms of J-VolMax objective (7a), i.e.,

$$\mathbb{E}[V(\boldsymbol{s}; \tilde{\Theta})] > \mathbb{E}[V(\boldsymbol{s}; \overline{\Theta})]. \tag{72}$$

The result is formally given in the following Lemma C.2.

**Lemma C.2.** *Denote any optimal solution of J-VolMax problem* (7) *as $\widehat{\Theta} := [\widehat{\boldsymbol{\theta}}, \widehat{\boldsymbol{\phi}}]$, learned from classes of learnable encoders and decoders $\mathcal{F}, \mathcal{G}$ that include the function classes of ground-truth encoders and decoders $\mathcal{F}', \mathcal{G}'$. Suppose there is an unknown finite set $\mathcal{S}_N := \{\boldsymbol{s}^{(1)}, ..., \boldsymbol{s}^{(N)}\} \subset \mathcal{S}$ with unknown $\mathcal{X}_N := \{\boldsymbol{x}^{(n)} \in \mathcal{X} : \boldsymbol{x}^{(n)} = \boldsymbol{f}(\boldsymbol{s}^{(n)}), \forall \boldsymbol{s}^{(n)} \in \mathcal{S}_N\} \subset \mathcal{X}$ such that the Assumption 3.1 is satisfied at each of the $N$ points in $\mathcal{S}_N$.*

*For the union of neighborhoods $U_N = \cup_{n=1}^{N} U^{(n)} = \cup_{n=1}^{N}\{\boldsymbol{s} \in \mathcal{S} : ||\boldsymbol{s} - \boldsymbol{s}^{(n)}||_2 < d^{(n)}\}$ within which $V(\boldsymbol{s}; \widehat{\Theta})$ is optimal, assume that $\mathcal{S}_N$ is sampled densely enough from $p_{\boldsymbol{s}}$ with sufficiently large $N$ such that*

$$\frac{\mathbb{P}(\boldsymbol{s} \in U_N)}{\mathbb{P}(\boldsymbol{s} \in \mathcal{S} \setminus U_N)} > \frac{G_{\max}}{G_{\min}} > 1, \tag{73}$$

*where $G_{\min}, G_{\max}$ are bi-Lipschitz constants of the Jacobian volume surrogate with respect to the parameter, i.e.,*

$$G_{\min}||\Theta_1 - \Theta_2||_2 \le |V(\boldsymbol{s}; \Theta_1) - V(\boldsymbol{s}; \Theta_2)| \le G_{\max}||\Theta_1 - \Theta_2||_2, \ \forall \boldsymbol{s} \in \mathcal{S}, \tag{74}$$

*for any parameters $\Theta_1, \Theta_2$ in $\mathcal{F}, \mathcal{G}$. Then, the optimal encoder $\widehat{\boldsymbol{g}} = \boldsymbol{g}_{\widehat{\boldsymbol{\phi}}}$ gives*

$$\widehat{\boldsymbol{g}}(\boldsymbol{x}^{(n)}) = \widehat{\boldsymbol{\Pi}}(\boldsymbol{s}^{(n)}) \widehat{\boldsymbol{\rho}}(\boldsymbol{s}^{(n)}), \ \forall \boldsymbol{s}^{(n)} \in \mathcal{S}_N, \tag{75}$$

*where $\widehat{\boldsymbol{\Pi}}(\boldsymbol{s}^{(n)})$ is a $\boldsymbol{s}^{(n)}$-dependent permutation matrix, and $\widehat{\boldsymbol{\rho}}(\boldsymbol{s}^{(n)}) = [\widehat{\rho}_1(s_1), ..., \widehat{\rho}_d(s_d)]^\top$ is a vector of $d$ invertible element-wise transformations on each component of $\boldsymbol{s}^{(n)}$.*

*Proof.* Suppose that there exists an optimal solution $\overline{\Theta} = [\overline{\boldsymbol{\theta}}, \overline{\boldsymbol{\phi}}]$ of J-VolMax such that the mapping $\overline{\boldsymbol{h}} = \boldsymbol{g}_{\overline{\boldsymbol{\theta}}} \circ \boldsymbol{f}$ is not a composition of permutation and element-wise invertible transformations. By Lemma C.1, within a certain union of neighborhoods $U_N$ such that (73) is satisfied, there exists another feasible solution $\tilde{\Theta} = [\tilde{\boldsymbol{\theta}}, \tilde{\boldsymbol{\phi}}]$ such that,

$$V(\boldsymbol{s}; \overline{\Theta}) < V(\boldsymbol{s}; \tilde{\Theta}), \ \forall \boldsymbol{s} \in U_N. \tag{76}$$

We now show that: since the solution $\overline{\Theta}$ does not reach maximal Jacobian volume at each point $s$ within $U_N$ as in (76), $\overline{\Theta}$ cannot reach maximal expected Jacobian volume over all of $\mathcal{S}$, i.e.,

$$\mathbb{E}[V(s; \overline{\Theta})] < \mathbb{E}[V(s; \tilde{\Theta})], \tag{77}$$

and therefore $\overline{\Theta}$ is actually not an optimal solution of J-VolMax, which raises a contradiction.

To begin, notice that the open neighborhood $U^{(n)} = \{s \in \mathcal{S} : ||s - s^{(n)}|| < d^{(n)}\}$ has non-zero measure, then their union $U_N = \cup_{n=1}^{N} U^{(n)}$ also has non-zero measure. This allows us to define

$$C_{U_N}(\overline{\Theta}) := \int_{s \in U_N} V(s; \overline{\Theta}) p(s) ds, \tag{78}$$

$$C_{\mathcal{S} \setminus U_N}(\overline{\Theta}) := \int_{s \in \mathcal{S} \setminus U_N} V(s; \overline{\Theta}) p(s) ds, \tag{79}$$

as the part over $U_N$ and the part over $\mathcal{S} \setminus U_N$ of the objective value $\mathbb{E}[V(s; \overline{\Theta})]$, i.e., $C_{U_N}(\overline{\Theta}) + C_{\mathcal{S} \setminus U_N}(\overline{\Theta}) = \mathbb{E}[V(s; \overline{\Theta})]$. Similarly, define

$$C_{U_N}(\tilde{\Theta}) := \int_{s \in U_N} V(s; \tilde{\Theta}) p(s) ds, \tag{80}$$

$$C_{\mathcal{S} \setminus U_N}(\tilde{\Theta}) := \int_{s \in \mathcal{S} \setminus U_N} V(s; \tilde{\Theta}) p(s) ds, \tag{81}$$

with $C_{U_N}(\tilde{\Theta}) + C_{\mathcal{S} \setminus U_N}(\tilde{\Theta}) = \mathbb{E}[V(s; \tilde{\Theta})]$.

Now, observe that

$$C_{U_N}(\tilde{\Theta}) - C_{U_N}(\overline{\Theta}) = \int_{s \in U_N} (V(s; \tilde{\Theta}) - V(s; \overline{\Theta})) p(s) ds \tag{82}$$

$$= \int_{s \in U_N} |V(s; \tilde{\Theta}) - V(s; \overline{\Theta})| p(s) ds \tag{83}$$

$$\geq \int_{s \in U_N} G_{\min} ||\tilde{\Theta} - \overline{\Theta}||_2 p(s) ds \tag{84}$$

$$= G_{\min} ||\tilde{\Theta} - \overline{\Theta}||_2 \int_{s \in U_N} p(s) ds \tag{85}$$

$$= G_{\min} ||\tilde{\Theta} - \overline{\Theta}||_2 \mathbb{P}(s \in U_N), \tag{86}$$

where (83) is due to the fact that $V(s; \tilde{\Theta}) > V(s; \overline{\Theta}), \forall s \in U_N$ as from Lemma C.1, and (84) is obtained by applying bi-Lipschitz continuity of $V(s; \cdot)$ as assumed in (74). Similarly, we have

$$|C_{\mathcal{S} \setminus U_N}(\tilde{\Theta}) - C_{\mathcal{S} \setminus U_N}(\overline{\Theta})| \leq \int_{s \in \mathcal{S} \setminus U_N} |V(s; \tilde{\Theta}) - V(s; \overline{\Theta})| p(s) ds \tag{87}$$

$$\leq \int_{s \in \mathcal{S} \setminus U_N} G_{\max} ||\tilde{\Theta} - \overline{\Theta}||_2 p(s) ds \tag{88}$$

$$= G_{\max} ||\tilde{\Theta} - \overline{\Theta}||_2 \int_{s \in \mathcal{S} \setminus U_N} p(s) ds \tag{89}$$

$$= G_{\max} ||\tilde{\Theta} - \overline{\Theta}||_2 \mathbb{P}(s \in \mathcal{S} \setminus U_N), \tag{90}$$

where (87) is obtained via triangle inequality, and (88) is from the bi-Lipschitz continuity of $V(s; \cdot)$ as in the assumption (74).

Combining (86), (90) with the assumption that $\mathcal{S}_N = \{s^{(1)}, ..., s^{(N)}\}$ is sampled densely and closely enough over $\mathcal{S}$ via $p_s$ such that

$$\frac{\mathbb{P}(s \in U_N)}{\mathbb{P}(s \in \mathcal{S} \setminus U_N)} > \frac{G_{\max}}{G_{\min}} > 1, \tag{91}$$

we have the following chain of inequalities:

$$C_{U_N}(\tilde{\Theta}) - C_{U_N}(\overline{\Theta}) \geq G_{\min} ||\tilde{\Theta} - \overline{\Theta}||_2 \mathbb{P}(s \in U_N) \tag{92}$$

$$> G_{\max}||\tilde{\Theta} - \overline{\Theta}||_2 \mathbb{P}(s \in \mathcal{S} \setminus U_N) \tag{93}$$

$$\geq |C_{\mathcal{S} \setminus U_N}(\tilde{\Theta}) - C_{\mathcal{S} \setminus U_N}(\overline{\Theta})| \tag{94}$$

$$\geq C_{\mathcal{S} \setminus U_N}(\overline{\Theta}) - C_{\mathcal{S} \setminus U_N}(\tilde{\Theta}). \tag{95}$$

Therefore, the following strict inequality holds:

$$C_{U_N}(\tilde{\Theta}) + C_{\mathcal{S} \setminus U_N}(\tilde{\Theta}) > C_{U_N}(\overline{\Theta}) + C_{\mathcal{S} \setminus U_N}(\overline{\Theta}), \tag{96}$$

or equivalently,

$$\mathbb{E}[V(s; \tilde{\Theta})] > \mathbb{E}[V(s; \overline{\Theta})]. \tag{97}$$

That is to say, $\tilde{\Theta} = [\tilde{\theta}, \tilde{\phi}]$ is indeed a feasible solution of J-VolMax that is strictly better than $\overline{\Theta} = (\overline{\theta}, \overline{\phi})$ in terms of expected Jacobian volume in J-VolMax objective (7a). This contradicts the assumed optimality of $(\overline{\theta}, \overline{\phi})$ to J-VolMax problem (7).

We hence conclude that any optimal solution with encoder $\widehat{g}$, the Jacobian at every point $s^{(n)} \in \mathcal{S}_N$ of the function $h^\star = \widehat{g} \circ f$ must satisfy

$$J_{h^\star}(s^{(n)}) = D(s^{(n)})\Pi(s^{(n)}),$$

where $D(s^{(n)})$ is an invertible diagonal matrix with $[D(s^{(n)})]_{i,i}$ dependent on $i$-th component $s_i^{(n)}$ only, and $\Pi(s^{(n)})$ is a permutation matrix dependent on the point $s^{(n)}$. $\qquad\square$

In the following Lemma C.3, we show that under further regularity conditions, the permutation ordering in the result (75) of Lemma C.2,

$$\widehat{g}(x^{(n)}) = \widehat{\Pi}(s^{(n)})\widehat{\rho}(s^{(n)}), \ \forall s^{(n)} \in \mathcal{S}_N,$$

become a constant permutation independent of $s^{(n)}$, i.e., $\widehat{\Pi}(s^{(n)}) = \widehat{\Pi}, \ \forall s^{(n)} \in \mathcal{S}_N$. That is,

$$\widehat{g}(x^{(n)}) = \widehat{\Pi}\widehat{\rho}(s^{(n)}), \ \forall s^{(n)} \in \mathcal{S}_N.$$

---

**Lemma C.3.** *Assume that there is a finite set $\mathcal{S}_N := \{s^{(1)}, ..., s^{(N)}\}$ with $\mathcal{X}_N := \{x \in \mathcal{X} : x = f(s), \forall s \in \mathcal{S}_N\}$ such that the Assumption 3.1 is satisfied at each of the $N$ points in $\mathcal{S}_N$. Denote $\widehat{g} \in \mathcal{G}$ as the optimal encoder learned by J-VolMax. Suppose that*

$$\widehat{g}(x^{(n)}) = \widehat{\Pi}(s^{(n)})\widehat{\rho}(s^{(n)}), \ \forall s^{(n)} \in \mathcal{S}_N \tag{98}$$

*for a $s^{(n)}$-dependent permutation $\widehat{\Pi}$, and additionally these three regularity conditions hold:*

1. *The functions $f, \widehat{g}, \widehat{\rho}$ are Lipschitz continuous with constants $L_f, L_{\widehat{g}}, L_{\widehat{\rho}} > 0$.*
2. *There is a constant $\gamma > 0$ such that for any permutation matrix $\Pi \in \mathcal{P}_d$ and $\Pi \neq \widehat{\Pi}(s^{(n)})$,*

$$||\widehat{g}(x^{(n)}) - \Pi\widehat{\rho}(s^{(n)})||_2 \geq \gamma, \ \forall n \in [N]. \tag{99}$$

3. *For $\mathcal{N}^{(n)} = \{s \in \mathcal{S} : ||s - s^{(n)}||_2 < r^{(n)}\}$ with $r^{(n)} < \frac{\gamma}{2(L_f L_{\widehat{g}} + L_{\widehat{\rho}})}$, the union set of the neighborhoods, $\mathcal{N} := \bigcup_{n=1}^N \mathcal{N}^{(n)}$, is a connected subset of $\mathcal{S}$.*

*Then, the permutation ordering $\widehat{\Pi}(s^{(n)})$ in (98) of the estimated latent components is constant, i.e., $\widehat{\Pi}(s^{(n)}) = \widehat{\Pi}$ for a fixed permutation $\widehat{\Pi} \in \mathcal{P}_d$. Consequently,*

$$\widehat{g}(x^{(n)}) = \widehat{\Pi}\widehat{\rho}(s^{(n)}), \forall n \in [N]. \tag{100}$$

---

*Proof.* Define a function $\ell_{\Pi}(s)$ parametrized by a permutation matrix $\Pi \in \mathcal{P}_d$ as

$$\ell_{\Pi}(s) := ||\widehat{g}(x) - \Pi\widehat{\rho}(s)||_2. \tag{101}$$

From the Lipschitz continuity of $f, \widehat{g}, \widehat{\rho}$, we have the corresponding Lipschitz continuity property of $\ell_{\Pi}(\cdot)$: for any $s, s' \in \mathcal{S}$,

$$\left|\ell_{\Pi}(s) - \ell_{\Pi}(s')\right| = \left|||\widehat{g}(f(s)) - \Pi\widehat{\rho}(s)||_2 - ||\widehat{g}(f(s')) - \Pi\widehat{\rho}(s')||_2\right| \tag{102}$$

$$\leq ||(\widehat{g}(f(s)) - \Pi\widehat{\rho}(s)) - (\widehat{g}(f(s')) - \Pi\widehat{\rho}(s'))||_2 \tag{103}$$

$$= ||(\widehat{g}(f(s)) - \widehat{g}(f(s'))) + (\Pi\widehat{\rho}(s') - \Pi\widehat{\rho}(s))||_2 \tag{104}$$

$$\leq ||\widehat{g}(f(s)) - \widehat{g}(f(s'))||_2 + ||\Pi\widehat{\rho}(s) - \Pi\widehat{\rho}(s')||_2 \tag{105}$$

$$= ||\widehat{g}(f(s)) - \widehat{g}(f(s'))||_2 + ||\widehat{\rho}(s) - \widehat{\rho}(s')||_2 \tag{106}$$

$$\leq L_{\widehat{g}}L_f ||s - s'||_2 + L_{\widehat{\rho}}||s - s'||_2 \tag{107}$$

$$= (L_{\widehat{g}}L_f + L_{\widehat{\rho}})||s - s'||_2, \tag{108}$$

where we applied the triangle inequality in (103) and (105), the fact that $\Pi \in \mathcal{P}_d$ is an orthogonal matrix in (106), and the Lipschitz continuity of $f, \widehat{g}, \widehat{\rho}$ in (107). Then, we have the following chain of inequalities for any $s \in \mathcal{N}^{(n)}$:

$$\ell_{\widehat{\Pi}(s^{(n)})}(s) \leq \ell_{\widehat{\Pi}(s^{(n)})}(s^{(n)}) + (L_{\widehat{g}}L_f + L_{\widehat{\rho}})||s - s^{(n)}||_2 \tag{109}$$

$$< \ell_{\widehat{\Pi}(s^{(n)})}(s^{(n)}) + (L_{\widehat{g}}L_f + L_{\widehat{\rho}})r^{(n)} \tag{110}$$

$$= (L_{\widehat{g}}L_f + L_{\widehat{\rho}})r^{(n)}, \tag{111}$$

where we have used the Lipschitz property of $\ell_{\widehat{\Pi}(s^{(n)})}$ for (109), the defined radius of $\mathcal{N}^{(n)}$ for (110), and the fact that $\ell_{\widehat{\Pi}(s^{(n)})}(s^{(n)}) = 0$ due to (98) for (111). Similarly, for any $s \in \mathcal{N}^{(n)}$ and any permutation matrix $\Pi \neq \widehat{\Pi}(s^{(n)})$,

$$\ell_{\Pi}(s) \geq \ell_{\Pi}(s^{(n)}) - (L_{\widehat{g}}L_f + L_{\widehat{\rho}})||s - s^{(n)}||_2 \tag{112}$$

$$> \ell_{\Pi}(s^{(n)}) - (L_{\widehat{g}}L_f + L_{\widehat{\rho}})r^{(n)} \tag{113}$$

$$> \gamma - (L_{\widehat{g}}L_f + L_{\widehat{\rho}})r^{(n)}, \tag{114}$$

where we again used the Lipschitz property of $\ell_{\widehat{\Pi}(s^{(n)})}$ for (112), the defined radius of $\mathcal{N}^{(n)}$ for (113), and the error gap of non-optimal permutations in assumption (99) for (114). Combining (111) and (114) with the assumption $r^{(n)} < \frac{\gamma}{2(L_{\widehat{g}}L_f + L_{\widehat{\rho}})}$, we have

$$\ell_{\widehat{\Pi}(s^{(n)})}(s) < (L_{\widehat{g}}L_f + L_{\widehat{\rho}})r^{(n)} < \gamma - (L_{\widehat{g}}L_f + L_{\widehat{\rho}})r^{(n)} < \ell_{\Pi}(s). \tag{115}$$

This implies that for any points $s$ in the $s^{(n)}$-centered neighborhood $s \in \mathcal{N}^{(n)}$, the optimal permutation for $\ell_{\Pi}(s)$ is still the optimal permutation $\widehat{\Pi}(s^{(n)})$ at the center point $s^{(n)}$; that is,

$$\widehat{\Pi}(s^{(n)}) = \arg\min_{\Pi \in \mathcal{P}_d} ||\widehat{g}(x) - \Pi\widehat{\rho}(s)||_2, \ \forall s \in \mathcal{N}^{(n)}. \tag{116}$$

As a result, $\widehat{\Pi}(\cdot) : \mathcal{N}_N \mapsto \mathcal{P}_d$, which maps a latent vector $s \in \mathcal{N}$ to a permutation matrix in $\mathcal{P}_d$, is a locally constant mapping over the union set $\mathcal{N}$. Since the locally constant $\widehat{\Pi}(\cdot)$ maps to a discrete space $\mathcal{P}_d$, the map $\widehat{\Pi}(\cdot)$ is indeed a continuous map. Combining the continuity of $\widehat{\Pi}(\cdot)$, a map from $\mathcal{N}$ into a discrete space $\mathcal{P}_d$, with the fact that $\mathcal{N}$ is connected, we can conclude that $\widehat{\Pi}(\cdot)$ is indeed constant. Hence, $\widehat{\Pi}(s) = \widehat{\Pi}$ for any $s \in \mathcal{N}$, which also implies $\widehat{\Pi}(s^{(n)}) = \widehat{\Pi}$ for any $n \in [N]$.

In conclusion, there is a single permutation matrix $\widehat{\Pi} \in \mathcal{P}_d$ such that

$$\widehat{g}(x^{(n)}) = \widehat{\Pi}\widehat{\rho}(s^{(n)}), \forall n \in [N]. \tag{117}$$

$\square$

Lastly, we combine the results from Lemma C.1, C.2, C.3 with a Rademacher complexity-based generalization bound to derive Theorem 3.3.

**Theorem 3.3** (Identifiability under Finite-sample SDI). *Assume that there is a finite set $\mathcal{S}_N := \{s^{(1)}, ..., s^{(N)}\}$ with $\mathcal{X}_N := \{x \in \mathcal{X} : x = f(s), \forall s \in \mathcal{S}_N\}$ such that the Assumption 3.1 is satisfied at each of the $N$ points in $\mathcal{S}_N$. Let $\widehat{g} \in \mathcal{G}$ be the optimal encoder by J-VolMax criterion (7), and $\overline{\Theta}$ contains the parameters of the learned encoder and decoder. Further assume that the following regularity conditions hold:*

1. *The functions $g = f^{-1}$ and $g_\phi$ are from classes $\mathcal{G}'$ and $\mathcal{G}$, respectively, where $\mathcal{G}' \subseteq \mathcal{G}$.*
2. *The functions $f, \widehat{g}, \widehat{\rho}$ are Lipschitz continuous with constants $L_f, L_{\widehat{g}}, L_{\widehat{\rho}} > 0$.*

3. *There is a $\gamma > 0$ such that for any permutation matrix $\mathbf{\Pi} \in \mathcal{P}_d$ and $\mathbf{\Pi} \neq \widehat{\mathbf{\Pi}}(\boldsymbol{s}^{(n)})$ (from (9)),*

$$||\widehat{\boldsymbol{g}}(\boldsymbol{x}^{(n)}) - \mathbf{\Pi}\widehat{\boldsymbol{\rho}}(\boldsymbol{s}^{(n)})||_2 \geq \gamma, \ \forall n \in [N].$$

4. *For $\mathcal{N}^{(n)} = \{\boldsymbol{s} \in \mathcal{S} : ||\boldsymbol{s} - \boldsymbol{s}^{(n)}||_2 < r^{(n)}\}$ with $r^{(n)} < \frac{\gamma}{2(L_f L_{\widehat{g}} + L_{\widehat{\rho}})}$, the union of the neighborhoods, $\mathcal{N} := \bigcup_{n=1}^{N} \mathcal{N}^{(n)}$, is a connected subset of $\mathcal{S}$ and $V(\boldsymbol{s}; \overline{\Theta})$ is optimal for any $\boldsymbol{s} \in \mathcal{N}$.*

5. *The points $\boldsymbol{s}^{(1)}, \ldots, \boldsymbol{s}^{(N)} \in \mathcal{S}_N$ densely locate in $\mathcal{S}$ such that*

$$\mathbb{P}(\boldsymbol{s} \in \mathcal{N}) / \mathbb{P}(\boldsymbol{s} \in \mathcal{S} \setminus \mathcal{N}) > G_{\max} / G_{\min} > 1, \tag{10}$$

*where $G_{\min}, G_{\max}$ are bi-Lipschitz constants of the Jacobian volume surrogate: for any parameters $\Theta_1, \Theta_2$ in $(\mathcal{F}, \mathcal{G})$,*

$$G_{\min}||\Theta_1 - \Theta_2||_2 \leq |V(\boldsymbol{s}; \Theta_1) - V(\boldsymbol{s}; \Theta_2)| \leq G_{\max}||\Theta_1 - \Theta_2||_2, \ \forall \boldsymbol{s} \in \mathcal{S}. \tag{11}$$

*Then, $\widehat{\boldsymbol{g}}(\boldsymbol{x}^{(n)}) = \widehat{\mathbf{\Pi}}\widehat{\boldsymbol{\rho}}(\boldsymbol{s}^{(n)}), \forall n \in [N]$ for a constant permutation matrix $\widehat{\mathbf{\Pi}} \in \mathcal{P}_d$. Furthermore, with probability at least $1 - \delta$,*

$$\mathbb{E}_{\boldsymbol{s} \sim p(\boldsymbol{s})}[||\widehat{\boldsymbol{g}}(\boldsymbol{x}) - \widehat{\mathbf{\Pi}}\widehat{\boldsymbol{\rho}}(\boldsymbol{s})||_2] = \mathcal{O}\left((L_f L_{\widehat{g}} + L_{\widehat{\rho}})\mathcal{R}_N(\mathcal{G}) + \sqrt{\ln(1/\delta)/N}\right), \tag{12}$$

*where $\mathcal{R}_N(\mathcal{G})$ is the empirical Rademacher complexity of the encoder class.*

*Proof.* Given that the J-VolMax criterion (7) provides us with an optimal solution $(\widehat{\boldsymbol{f}}, \widehat{\boldsymbol{g}})$ reaching maximal Jacobian volume at each $\boldsymbol{s}^{(n)}$, such that the estimated latent components $\widehat{\boldsymbol{g}}(\boldsymbol{x}^{(n)}) = \widehat{\boldsymbol{s}}^{(n)}$ identifies $\boldsymbol{s}^{(n)} \in \mathcal{S}_N$, up to permutation and component-wise transformation, i.e.,

$$\widehat{\boldsymbol{g}}(\boldsymbol{x}^{(n)}) = \widehat{\mathbf{\Pi}}\widehat{\boldsymbol{\rho}}(\boldsymbol{s}^{(n)}), \ \forall n \in [N], \tag{118}$$

we now show a finite-sample analysis on how well $\widehat{\boldsymbol{g}}$ can estimate the ground-truth latent sources over all $\boldsymbol{s} \in \mathcal{S}$, up to the said permutation $\widehat{\mathbf{\Pi}}$ and the invertible element-wise mappings $\widehat{\boldsymbol{\rho}}(\cdot)$ associated with $\boldsymbol{s}^{(n)} \in \mathcal{S}_N$. To proceed, let us define a loss function

$$\ell(\boldsymbol{g}, (\boldsymbol{x}, \boldsymbol{s})) = ||\boldsymbol{g}(\boldsymbol{x}) - \widehat{\mathbf{\Pi}}\widehat{\boldsymbol{\rho}}(\boldsymbol{s})||_2, \tag{119}$$

which is $L_\ell$-Lipschitz, with the constant depending on Lipschitz continuity of $\widehat{\boldsymbol{g}}, \boldsymbol{f}, \widehat{\boldsymbol{\rho}}$ as $L_\ell = L_{\widehat{g}} L_f + L_{\widehat{\rho}}$ (as derived in Lemma C.3). The defined loss function can be assumed to be upper-bounded by a finite constant as $\ell(\boldsymbol{g}, (\boldsymbol{x}, \boldsymbol{s})) \leq M$. Note that the encoder from J-VolMax $\widehat{\boldsymbol{g}}$ minimizes the following empirical risk:

$$\widehat{\mathcal{L}}(\boldsymbol{g}) := \frac{1}{N} \sum_{n=1}^{N} \ell(\boldsymbol{g}, (\boldsymbol{x}^{(n)}, \boldsymbol{s}^{(n)})), \tag{120}$$

i.e., $\widehat{\mathcal{L}}(\widehat{\boldsymbol{g}}) = 0$. We are now in a position to use a generalization bound that characterizes the difference between the given empirical risk $\widehat{\mathcal{L}}(\phi)$ and the population risk

$$\mathcal{L}(\boldsymbol{g}) := \mathbb{E}_{\boldsymbol{s} \sim p_s}[\ell(\boldsymbol{g}, (\boldsymbol{x}, \boldsymbol{s}))] = \mathbb{E}_{\boldsymbol{s} \sim p_s}[||\boldsymbol{g}(\boldsymbol{x}) - \widehat{\mathbf{\Pi}}\widehat{\boldsymbol{\rho}}(\boldsymbol{s})||_2]. \tag{121}$$

Applying an (empirical) Rademacher complexity-based generalization bound [70, Theorem 26.5] with the contraction lemma [70, Lemma 26.9] gives the following with probability at least $1 - \delta$:

$$\mathcal{L}(\boldsymbol{g}) \leq \widehat{\mathcal{L}}(\boldsymbol{g}) + 2L_\ell \mathcal{R}_N(\mathcal{G}) + 4M\sqrt{\frac{2\ln(4/\delta)}{N}}, \ \forall \boldsymbol{g} \in \mathcal{G}. \tag{122}$$

As a result, the following bound holds for any optimal solution $\widehat{\boldsymbol{g}} \in \mathcal{G}$: with probability at least $1 - \delta$,

$$\mathcal{L}(\widehat{\boldsymbol{g}}) = \mathbb{E}_{\boldsymbol{s} \sim p_s}[||\widehat{\boldsymbol{g}}(\boldsymbol{x}) - \widehat{\mathbf{\Pi}}\widehat{\boldsymbol{\rho}}(\boldsymbol{s})||_2] \leq 2L_\ell \mathcal{R}_N(\mathcal{G}) + 4M\sqrt{\frac{2\ln(4/\delta)}{N}}. \tag{123}$$

$\square$

# D  Additional Remarks on Related Works

**IMA.** The IMA framework also exploits influence diversity [20]. There, the ground-truth decoder $\boldsymbol{f}$ is assumed to have an orthogonal Jacobian (e.g., Möbius transformations [20]), which reflects linearly uncorrelated influences from the latent components. Similar orthogonal Jacobian-based regularization have found empirical successes for disentanglement problems in computer vision [71]. However, IMA does not provide identifiability (only *local* identifiability was shown [44]).

**Sparse Jacobian-based NMMI.** Our proposed SDI condition naturally subsumes some cases of sparse Jacobian conditions employed in [2, 3, 19, 49]. A notable example is when $m = 2d$, the SDI assumption boils down to a sparsity pattern in $\boldsymbol{J_f}$ where each row touches a corner of the weighted $L_1$ ball $\mathcal{B}_1^{\boldsymbol{w}(s)}$—and thus the gradients are scaled unit vectors (with $d - 1$ zeros). Nonetheless, when $m > 2d$, completely dense $\boldsymbol{J_f}$ can also satisfy SDI. We note that although the $L_1$ regularization on $\boldsymbol{J_f}$ appears in both J-VolMax and sparse Jacobian criteria (see [3, 19, 50]), the reasons of having this regularization are very different: the latter use the $L_1$ norm as a surrogate to attain Jacobian sparsity, but our method uses the regularization to confine $\nabla f_i(\boldsymbol{s})$'s in an $L_1$-norm ball (which is not a proxy for sparsity).

**Object-centric Representation Learning (OCRL).** Many OCRL works also share the perspective of modeling influences—particularly the influences of latent variables onto objects/concepts in the observed domain. While the goal of NMMI is to recover each individual latent variable, OCRL [2, 23, 24, 48] seeks a weaker form of identifiability—identifying blocks of variables corresponding to observed objects/concepts. When the block size becomes 1, the goal of OCRL becomes latent variable identification as in NMMI. Notably, [2, 23, 24] propose specific forms of $\boldsymbol{f}$ (which is called "decoder" in the OCRL literature) for block-wise identifiability. These structures often imply sparsity in Jacobian or higher-derivatives of $\boldsymbol{f}$. For example, the *compositional generator* in [2] and *additive decoder* in [23] are associated with structured sparse $\boldsymbol{J_f}$'s. Similar to DICA, the work [2] uses a Jacobian-based regularizer in their training loss. The work [23] imposes the additive decoder structure at the model architecture level. Interestingly, the additive decoder structure in [23] was shown to have the same disentanglement effects as a block-diagonal Hessian penalty proposed in [51].

# E  Experiment Details

## E.1  Further Details on Implementation and Evaluation

**On Warm-up Heuristic.** The use of warm-up heuristic with regularization schedulers help alleviate the numerical instability that comes with optimizing $\log \det$ of Jacobian of the decoding neural network $\boldsymbol{f_\theta}$, which can quickly explode if not controlled. Specifically, by gradually introducing the $\log \det$ term, we prioritize optimizing for data reconstruction in the warm-up period, since it is a hard constraint in the J-VolMax formulation (7c). Moreover, by minimizing $||\boldsymbol{J_{f_\theta}}(\boldsymbol{g_\phi}(\boldsymbol{x}^{(n)}))||_1$ during warm-up, we prevent the Jacobian from exploding as a result of maximizing $c_{\mathrm{vol}}(t)$ while keeping the norm term $||\boldsymbol{J_{f_\theta}}(\boldsymbol{g_\phi}(\boldsymbol{x}^{(n)}))||_1$ at a reasonable magnitude. This also encourages a smoother encoder $\boldsymbol{f_\theta}$ with small enough Lipschitz constant; recall that this is important for identifiability of J-VolMax under finite SDI-satisfying samples, as pointed out by Theorem 3.3.

**Efficient Computation of Jacobian Volume Regularizer.** As mentioned in Section 5, the computational cost of the log-det volume surrogate $c_{\mathrm{vol}}$ in (14) is $\mathcal{O}(m^3)$. This might make $c_{\mathrm{vol}}$ impractical to use in high-dimensional data setting, where the dimension of the Jacobian matrix is large. An alternative volume surrogate is to use [34, 72]

$$\hat{c}_{\mathrm{tr}} = \mathrm{Tr}((d\boldsymbol{I} - \boldsymbol{1}\boldsymbol{1}^\top)\boldsymbol{J_{f_\theta}}(\boldsymbol{g_\phi}(\boldsymbol{x}^{(n)})^\top \boldsymbol{J_{f_\theta}}(\boldsymbol{g_\phi}(\boldsymbol{x}^{(n)})) \tag{124}$$

$$= \sum_{i=1}^{d-1} \sum_{j=i+1}^{d} \left\lVert \frac{\partial \boldsymbol{f_\theta}(\boldsymbol{g_\phi}(\boldsymbol{x}^{(n)}))}{\partial \hat{s}_i^{(n)}} - \frac{\partial \boldsymbol{f_\theta}(\boldsymbol{g_\phi}(\boldsymbol{x}^{(n)}))}{\partial \hat{s}_j^{(n)}} \right\rVert_2^2 \tag{125}$$

which corresponds to the sum of Euclidean distances between all pairs of partial derivative vectors $\partial \boldsymbol{f_\theta}/\partial \hat{s}_i$. Maximizing $c_{\mathrm{tr}}$ would force the partial derivatives to be spread out, akin to the effect from maximizing log-det surrogate $c_{\mathrm{vol}}$, while reducing the computational cost to $\mathcal{O}(m^2)$. In Table 3, we test the wall-clock time needed for backpropagation per epoch of the logdet-based and trace-based regularization, using a setting similar to Mixture C in our synthetic experiment (with $d = 3, m = 40$).

Table 3: Compare performance and wall-clock time for gradient computation (per epoch) of trace-based and logdet-based surrogate for Jacobian volume.

|  | Logdet-based DICA | Trace-based DICA | Sparse | Base |
|---|---|---|---|---|
| $R^2$ score | $0.94 \pm 0.08$ | $0.91 \pm 0.07$ | $0.79 \pm 0.12$ | $0.63 \pm 0.04$ |
| Time (ms) | $31.31 \pm 0.30$ | $24.89 \pm 0.26$ | $18.95 \pm 0.18$ | $6.90 \pm 0.18$ |

One can see that while trace-based DICA formulation is approximately $25\%$ faster than logdet-based DICA, $R^2$ score only slightly decreases. Lastly, we remark that there are other methods for reducing the computational cost for optimizing Jacobian determinant of a neural network, such as [73].

**Evaluation with $R^2$.** To evaluate the $R^2$ score, we use nonlinear kernel ridge regression with radial basis function kernel [74], which is an universal function approximator. We train the regression model on a train set, and use the prediction of the trained regressor on a test set to calculate the coefficient of determination. This gives us the nonlinear $R^2$ score.

**Compute Resources.** All experiments use one NVIDIA A40 48GB GPU, hosted on a server using Intel Xeon Gold 6148 CPU @ 2.40GHz with 260GB of RAM.

### E.2 Synthetic Simulations

For each of the simulation, we generate 30000 samples from the described synthetic data generation processes, of which $90\%$ are for training and $10\%$ are for evaluating the $MCC$ and $R^2$ scores. We use two ReLU fully-connected neural networks with one 64-neuron hidden layer for the encoder and the decoder. The autoencoder is trained via Adam method [75] with learning rate $10^{-4}$ for 200 iterations, among which the first 20 epochs are for warm-up. The regularization hyperparameters are chosen by validating from $\{10^{-2}, 10^{-3}, 10^{-4}, 10^{-5}\}$, resulting in $\lambda_{\text{vol}} = 10^{-4}, \lambda_{\text{norm}} = 10^{-4}, \lambda_{\text{sp}} = 10^{-4}$. The neural networks are initialized via He initialization with uniform distribution [76].

### E.3 Single-cell Transcriptomics Analysis

**Data Generation with SERGIO Simulator.** We extract top 5 TFs that regulate the most number of genes in TRRUST database. Furthermore, we only keep $30\%$ of genes with a single regulator, so as to have a more challenging mixture; this gives us $m = 178$ gene expressions.

To construct the gene regulatory mechanism $\boldsymbol{f}$ for SERGIO generation, we use the extracted high-confident interactions from TRRUST as primary regulating edges with coefficients randomly sampled from $U(1.5, 1.8)$ if the TF is the gene's activator, and from $U(-1.8, -1.5)$ if the TF is the gene's repressor. In addition, to model the potential spurious cross-talks interactions between the TFs and the genes, we add secondary weak regulating edges with coefficients with random signs and their magnitudes uniformly sampled from $U(0.1, 1.0)$.

We simulate 20000 cells of the same cell type, and use the gene expression data of these cells as the observation dataset to train with J-VolMax learning criterion.

**Training.** We use two ReLU fully-connected neural networks with one 64-neuron hidden layer for encoder and decoder. We use Adam optimizer [75] with learning rate $10^{-4}$, and train the autoencoder for 4000 epochs, with the first 1000 epochs for warm-up. The regularization hyperparameters are chosen by validation from $\{10^{-2}, 10^{-3}, 10^{-4}, 10^{-5}\}$ to be $\lambda_{\text{vol}} = 10^{-3}, \lambda_{\text{norm}} = 10^{-4}, \lambda_{\text{sp}} = 10^{-4}$. The neural networks are initialized using He initialization with normal distribution [76].

## F  Additional Experiment: Unsupervised Concept Discovery in MNIST

**Setting.** In this section, we explore further experiments beyond nonlinear mixture model identification tasks, to demonstrate potential applicability of DICA in disentanglement and representation learning. Specifically, we apply DICA to the MNIST dataset [77] to train an autoencoder using J-VolMax loss function. We use convolutional neural networks (CNNs) for both encoder and decoder, with the architectures reported in Table 4. We choose $d = 10$ as the latent dimension, and the observation dimension is $m = 32 \times 32 = 1024$. The regularization hyperparameters are $\lambda_{\text{vol}} = 10^{-4}, \lambda_{\text{sp}} = 10^{-4}$, and we optimizing using Adam with learning rate $10^{-3}$ for 100 iterations, of which the first 50

Table 4: Architectures of encoder and decoder used in MNIST experiment (all use stride 2, pad 1, out_pad 1)

| *Encoder* | *Decoder* |
|---|---|
| **Input:** $\boldsymbol{x} \in \mathbb{R}^{32 \times 32 \times 1}$ | **Input:** $\widehat{\boldsymbol{s}} \in \mathbb{R}^{10 \times 1 \times 1}$ |
| $3 \times 3$ Conv, 256 ReLU | $2 \times 2$ ConvTrans, 32 ReLU |
| $3 \times 3$ Conv, 128 ReLU | $3 \times 3$ ConvTrans, 64 ReLU |
| $3 \times 3$ Conv, 64 ReLU | $3 \times 3$ ConvTrans, 128 ReLU |
| $3 \times 3$ Conv, 32 ReLU | $3 \times 3$ ConvTrans, 256 ReLU |
| $2 \times 2$ Conv, 10 | $3 \times 3$ ConvTrans, 1 |

epochs are for warm-up. To prevent numerical instability regarding the log-determinant of Jacobian due to high-dimensional data of $m = 1024$, instead of optimizing $\log \det(\cdot)$ directly, we optimize $\log \det(\cdot + \tau \boldsymbol{I})$ with $\tau > 0$, so as to avoid zero determinant that leads to exploding log-determinant.

To visualize how the learned latent factors affect the observed images, we encode a test image corresponding to a digit to achieve a latent vector $\widehat{\boldsymbol{s}} = [s_1, ..., s_{10}] \in \mathbb{R}^{10}$. Then, we vary each component $s_i$ in the range of $\pm 4$ std to achieve a new latent vector $\widehat{\boldsymbol{s}}_{\text{new}}$ with a certain component being increased/decreased. The new latent vector $\widehat{\boldsymbol{s}}_{\text{new}}$ are used as input of the trained decoder to obtain a new image $\boldsymbol{x}_{\text{new}}$. Therefore, we can examine how a specific latent component $s_i$ can affect the observed image $\boldsymbol{x}_{\text{new}}$.

**Results.** We reported four visualizations corresponding to varying $s_8, s_9, s_1, s_7$ from images with digit $3, 0, 2, 5$, correspondingly in Fig. 3. One can observe that as a learned latent component increases/decreases, the semantic meaning (i.e., digit) of the image changes in a relatively uniform manner towards a new digit. This suggests that the latent components induced by J-VolMax are somewhat correlated with semantic meanings of the images.

We note that not all latent components we obtained correspond to a clear semantic meaning, and the variation of latent components can sometimes significantly distort the images beyond normal hand-written images. We hypothesize that this is due to the used autoencoder architecture not being suitable for disentanglement purposes, as well as the optimization algorithm not well-designed for this task, since we directly implement J-VolMax criterion without adaptations for image data. Nonetheless, the preliminary results are encouraging, and we speculate that with better-designed architecture and algorithm, the J-VolMax criterion can significantly improve in the challenging task of disentangling meaningful latent components of image data.

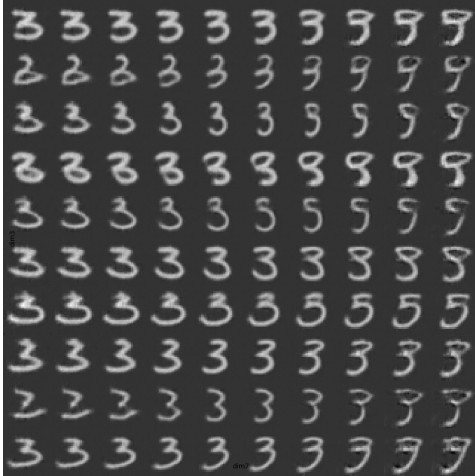

(a) [Anchor digit 3] As $s_8$ *increases*, digit 3 increasingly looks like digit 9.

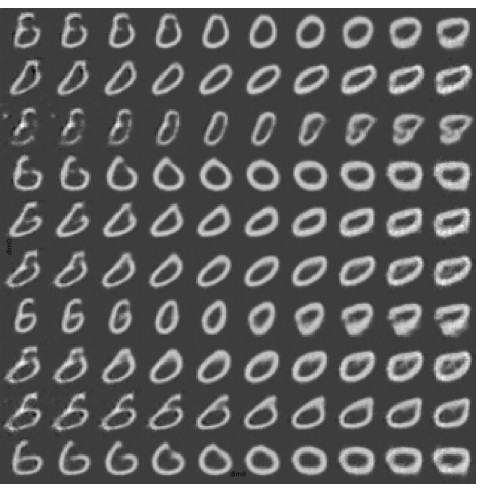

(b) [Anchor digit 0] As $s_9$ *decreases*, digit 0 increasingly looks like digit 6.

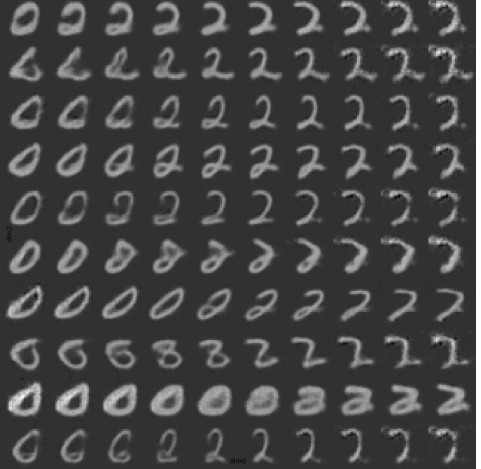

(c) [Anchor digit 2] As $s_1$ *decreases*, digit 2 increasingly looks like digit 0.

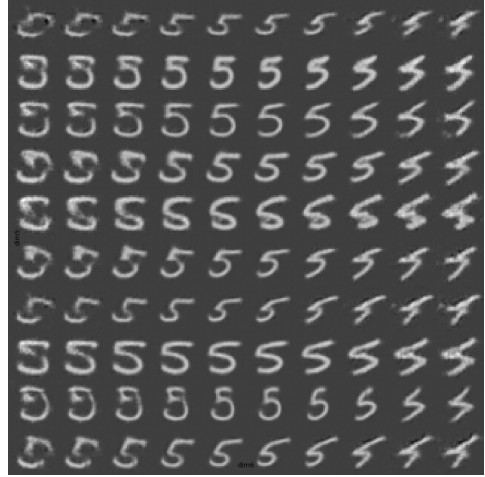

(d) [Anchor digit 5] As $s_7$ *increases*, digit 5 increasingly looks like digit 4.

Figure 3: Some resulting images obtained by varying a certain component $s_i$ by $\pm 4$ std (increasing from left to right) from the latent vector of an anchor image. Each row corresponds to one of 10 different anchor images sampled from test set, and each column is the resulting image by varing from the corresponding anchor image. We can see that some latent components correlate to the semantic meaning (i.e., digit) of output images: as some $s_i$ increases/decreases, the semantic digit of all 10 anchor images change uniformly towards another digit.

