# OpenReview forum: "Diverse Influence Component Analysis: A Geometric Approach to Nonlinear Mixture Identifiability"
_NeurIPS.cc/2025/Conference — NeurIPS 2025 poster_

### Official Review · Reviewer_pCZF · 2025-06-27

**Clarity:** 1
**Significance:** 3
**Originality:** 2
**Rating:** 4
**Confidence:** 4

**Summary:**

The article addresses the problem of identifying latent features from their nonlinear mixtures. The approach proposed is based on exploiting a data model where the the rows of the Jacobian matrix of the nonlinear mapping are assumed to lie in a scaled $\ell_1$ norm ball. The article proposes the use of two neural networks as parametric nonlinearities, whose parameters are obtained through a volume maximization on the gradients of the nonlinear mixing to exploit their pressumed scattering within the scaled $\ell_1$ norm ball. The article offers numerical experiments to illustrate the performance of the proposed approach which look convincing.

**Questions:**

- what is the motivation for the assumption about the Jacobian (inclusion in the l-1 norm ball) besides identifiability? Do you see that it is a practically valid assumption? can we also consider other sets alternative to l-1 norm ball (such as other polytopes in [48])?

**Ethical Concerns:**

["NO or VERY MINOR ethics concerns only"]

**Final Justification:**

As I noted in my original review the article addresses a noteworthy theoretical problem. However, the presentation style was confusing as it ignored almost all relevant background in deriving the proposed approach. This led to confusion about how the authors came up with their volume optimization based approach and the SDI condition. However, their response clarifies that  they will perform a significant improvement in that direction. Furthermore, they provided satisfactory answers for general review comments. Due to the significance of this work I increased my score to 4.

**Limitations:**

Yes.

**Paper Formatting Concerns:**

None.

**Quality:**

2

**Strengths And Weaknesses:**

## Strengths

 - The article addresses the hard problem of obtaining latent factors from their nonlinear mixtures and offers a solution with proven identifiability under some assumptions about the data model

## Weaknesses

A. One significant concern is the lack of proper attribution to prior work from which key definitions and methodology appear to be adopted. Although the referenced article is included in the general bibliography, the paper fails to cite it at the precise points where the reused ideas are introduced. As a result, the presentation gives a misleading impression of originality. This omission is not merely stylistic—it impairs the reader's ability to understand the origin and novelty of the contributions and violates standard scholarly norms of credit and transparency. Furthermore, this makes it difficult to assess whether the paper's contributions are genuinely novel or derivative.


The main reference under consideration is [48] G. Tatli et.al. “Polytopic Matrix Factorization: Determinant Maximization Based Criterion and Identifiability”. In: IEEE Transactions on Signal Processing 69 (2021), pp. 5431–5447.

Although the authors very briefly list this reference at the end of Section 4, the current work appears to closely extend the polytopic matrix factorization (PMF) introduced in [48] for linear mixtures to nonlinear mixtures. In this endaveour, the author(s) appeared to use definitions, criterion and the proof steps in [48] without any precise referals.

To clarify this point, I checked [48] and summarize main relevant points here: [48] proposes  PMF for the extraction of latent factors from their mixtures, which doesn't assume independence of latent factors but assumes a generative model (in Section II.A of [48]) for the input matrix $\mathbf{Y}$:

$$ \mathbf{Y}=\mathbf{H}_g\mathbf{S}_g $$

where $\mathbf{H}_g$ is unknown full-column-rank factor matrix and $\mathbf{S}_g$ is the unknown matrix whose columns are in a polytope $P$. [48] proposes the use of the following volume maximization  problem (in Section II.B)  to identify these factors  from the input matrix $\mathbf{Y}$:

*PMF Problem*
\begin{array}{ll}
\displaystyle \max_{\mathbf{H} \in \mathbb{R}^{M \times r},\, \mathbf{S} \in \mathbb{R}^{r \times N}} & \det(\mathbf{S}\mathbf{S}^T)  \tag{PMF}\\
\text{subject to} & \mathbf{Y} = \mathbf{H} \mathbf{S}, \\
& \mathbf{S}_{:,j} \in \mathcal{P}, \quad j = 1, \dots, N.
\end{array}


For the identifiability condition it introduces the *sufficient scattering condition* (in Section II.c) for the columns of $\mathbf{S}$ as

*{Definition II.2: Sufficiently Scattered Factor:}*
$\mathbf{S} \in \mathbb{R}^{r \times N}$ is called a sufficiently scattered factor corresponding to $\mathcal{P}$ if

  - (PMF.SS.i) $\mathcal{P} \supseteq \mathrm{conv}(\mathbf{S}) \supset \mathcal{E}_{\mathcal{P}},$ and
  - (PMF.SS.ii) $\mathrm{conv}(\mathbf{S})^{*, \mathbf{g}_{\mathcal{P}}} \cap \mathrm{bd}(\mathcal{E}_{\mathcal{P}}^{*, \mathbf{g}_{\mathcal{P}}}) = \mathrm{ext}(\mathcal{P}^{*, \mathbf{g}_{\mathcal{P}}}),$

where $\mathcal{E}_{\mathcal{P}}$ is the MVIE of $\mathcal{P}$, centered at $\mathbf{g}_{\mathcal{P}}$.

Section III.B of [48] focuses on the special case $\mathcal{P}=\mathcal{B}_1$ ($\ell_1$ norm ball), i.e. the sparse PMF case. Then Theorem 3 of [48] proves that the optimal solution of the (PMF) optimization problem above under sufficiently scattered condition (Definition II.2) recovers the factors of the PMF generative model upto sign and permutation ambiguity.

From this perspective, the contribution in the current article is an extension of the Sparse-PMF of [48]:

- The Sufficiently Diverse Influnce definition (Assumption 3.1) is exactly the sufficiently scattered condition (Definition II.1 above) in [48] applied to the rows of the Jacobian for the special case of polytope $\mathcal{P}=C \mathcal{B}_1$. Figure 1 is the two dimensional version of Fig. 5 in [48].

- Jacobian Volume Maximization problem in Page 5 is the extension of the Sparse PMF problem above, imposing sparsity to the Jacobian matrix, replacing the linear reconstruction constraint in Sparse-PMF with the nonlinear construction constraint.

- The proof of Theorem 3.2 (Identifiability of J-VolMax) in Appendix B, shares exact same steps as the proof of Theorem 3 for Sparse PMF in [48]:
    * the treatment between equations (22)-(27) in the proof of Theorem 3.2 are the same as the treatment around (15)-(17) in the proof of Theorem 3 in [48],
    *  Lemma B.1 in Appendix (from a nonexisting reference [55]) is the same as the second stage of the proof of Theorem 3 of [48],
    * the treatment between equations (28)-(33) in the proof of Theorem 3.2 are the same as the treatment in the third stage of the proof of Theorem 3 (around equation (24)) in [48].

In summary, the current article appears to be the extension of the sparse-PMF for linear mixing to nonlinear mixings.
Despite this tight  extension relationship, and  the reuse/generalization of results in  [48],

- there is no mention of the reference [48] in Section 2 (Background). Section 2 should also discuss the reference
K. Huang, N. D. Sidiropoulos, and A. Swami, “Non-negative matrix factorization revisited: Uniqueness and algorithm for symmetric decomposition,” IEEE Transactions on Signal Processing, vol. 62, no. 1, pp. 211–224, October 2013
which appears to be the first article where the sufficiently scattered condition for latent extraction, the root of the sufficient diverse influence condition, was introduced,
- there is no discussion about how the current work extends Sparse-PMF for linear mixings to  nonlinear mixings,
- there are no precise references at the appropriate points to the original determinant maximization based sparse PMF problem and the corresponding sufficiently scattered identifiability condition definitions, Theorem 3, and the proofs of Theorem 3 in [48].




B.  The assumptions about the data model (about the Jacobian), their applicability to practical problems could be better motivated and explained.


B. Section 5.2 "Inferring Transcription Factor" could be better explained in terms of the problem statement and the performance measure.

---

> ### Author Rebuttal · Authors · 2025-07-31
>
> ## Relation between DICA and matrix factorization literature
>
> ***[Relation to NMF/PMF - SDI]*** We would like to thank the reviewer for bringing up this issue. We sincerely acknowledge the importance of the work [48] (as well as [9, 17, 18]) and their contributions to matrix factorization’s identifiability analysis via volume-based criteria under scattering conditions. As we mentioned in “Related works”, the SDI is indeed a scattering-based condition as those in [9, 17, 18, 48], though formulated in the Jacobian domain. However, we understand the reviewer’s concern that the mentioning was too brief to fully clarify the connections. To address this, we plan to expand the related works and add remarks as follows:
>
> > “Volume minimization/maximization based approaches have been popular in structured matrix factorization (SMF) [9, 17, 18, 48], where scattering based conditions are used to establish identifiability for these criteria. The work [A4] first used the so-called sufficiently scattered condition (SSC) (the condition itself appeared in [A5] for NMF but no determinant was involved) to show that volume minimization of a latent factor identifies an SMF model under the SSC. The SSC means that the columns/rows of a factor matrix in an SMF model are widespread in the nonnegative orthant. This line of work later was generalized to SSC in different norm balls [48] (also see [17]). The SDI condition can be viewed as a functional extension of the column/row-based SSC condition [48, 17] to the Jacobian domain over a continuous manifold (in particular, the L1 norm ball based SSC in [48]). Compared to SSC, SDI admits rather different physical meaning as it is defined using the first-order derivatives. Nonetheless, the J-VolMax criterion can still be regarded as a nontrivial extension of volume-based SMF [A4, 9, 17, 48] (particularly [48]) into the nonlinear domain. In addition, although the contexts of nonlinear and linear unmixing (e.g., SMF) are drastically different, some proof techniques of volume based SMF [48] are found useful in our case (see Appendix B).”
>
> As the reviewer suggested, we will also revise the “Background” section accordingly to reflect the above.
>
> ***[Relation to NMF/PMF - Proof Contributions]*** We thank the reviewer again for raising this important point. While there was no intention to omit proper attribution to prior work, we agree in hindsight that key connections—particularly in parts of the proof—were not made sufficiently explicit. **We take full responsibility for this lack of clarity and are committed to improving it**.
>
> **To clarify, the non-existing [55] in the Appendix B was exactly [48], from which we cited parts of [Theorem 3, 48] as Lemma B.1 to use as a key component in the proof of Theorem 3.2**. As the main paper and the appendices were submitted separately (with one week apart), the bibliography was not cleanly managed and such missing references happened inadvertently. We apologize for this oversight.
> Regarding the proof contributions, note that the proof of Theorem 3.2 consists of the following components:
>
> 1. Lemma B.2 first relies on recasting the nonlinear identifiability problem J-VolMax into the first-order derivative domain. This leverages the smoothness of the nonlinear function and the availability of infinite data (so that the operations are in a continuous manifold). This part includes (12)-(21).
>
> 2. After establishing (21) in the Jacobian domain, the steps in (22)-(33) utilizes the SDI and a series of facts from convex analysis. In particular, these facts were presented in multiple prior works, particularly, [48] and [17].
>
> * The steps in (22)-(27) utilizes insights from convex geometry to characterize the Jacobian. Similar facts (in form of matrix columns/rows) were presented in prior works in polytopic matrix factorization [Theorem 3, 48] (and later dictionary learning [Lemma A.2, 17]);
>
> * In step (28), we apply a key insight from parts of [Theorem 3, 48], cited as Lemma B.1 from the non-existing [55] in our manuscript, to establish a condition for when an involved Jacobian matrix is orthogonal;
>
> * Finally, (29)-(33) are direct consequences of (28), i.e., Lemma B.1 from [48], and the SDI condition to establish that an involved Jacobian is not only orthogonal, but indeed a permutation matrix. This part again characterizes the Jacobian, but similar facts exist in [48] characterizing matrix factors.
>
> 3. From (34) to (53), the proof shows that the Jacobian volume must be maximized as in the learning criterion, such that (33) is not contradicted.
>
> 4. Finally, in Theorem 3.2, the proof is to utilize Lemma B.2 and continuity to unify permutation ambiguity over the smooth manifold.
>
> Therefore, we hope the reviewer agrees that, while our proof draws on known convex analysis tools in one segment, it goes substantially beyond previous work in both scope and technical content. In particular, the shift from factor matrices to Jacobians, and the treatment of identifiability over continuous manifolds, present challenges not encountered in linear models such as PMF or NMF. **That said, we fully agree with the reviewer that the use of known results—even when adapted or generalized—should be acknowledged explicitly**. It is unfortunate that the key cited lemma from [48] appeared as non-existing [55], because Lemma B.1 is an important step in the proof that we hoped to credit properly. We will revise Appendix B to include the correct citation, i.e., [48], and make the following remark after (22)–(33):
>
> > " The convex geometric arguments in Eqs. (22)–(33) are from tools developed in prior matrix factorization literature, particularly [48], where they were used to establish identifiability under scattering-based conditions.”
>
> *References*
>
> [A4] Fu et al., “Blind Separation of Quasi-Stationary Sources: Exploiting Convex Geometry in Covariance Domain”, IEEE TSP, 2015.
>
> [A5] Huang et al., “Non-negative matrix factorization revisited: Uniqueness and algorithm for symmetric decomposition”, IEEE TSP, 2013.
>
> ## SDI motivation and real-world applicability
>
> We thank the Reviewer for this question. Please find our answer in the following, which contain parts of a reply to Reviewer Sbtv.
>
> The motivation of DICA is to exploit the pattern of how sources $\mathbf{s}$ influence the observations $\mathbf{x}$ via the geometry structure of the mixing Jacobian matrix, in a manner similar to sparsity-based NMMI and the IMA. In particular, the latent variable model of DICA assumes a prior belief that
>
> > ***the way that distinct sources $s_1, …, s_d$ influence the observations are sufficiently different***,
>
> in the sense that the columns of Jacobian matrices spanned in different directions. **Therefore, the change to observations** $\mathbf{x}$ **by varying** $s_i$ **should be sufficiently different from that by varying** $s_j$. This can be described by how columns/rows of Jacobian are spread out.
>
> To quantitatively pin down this intuition to form the exact SDI condition, we follow the idea in the matrix factorization literature [9, 17, 18, 48] (particularly [48]), where sufficiently scattered conditions (SSCs) are often used to prove identifiability of linear mixture. SSC describes how column/row vectors of a factor matrix are scattered inside a convex polytope. Instead, we describe the scattering pattern of mixing Jacobian matrices at every point $\mathbf{s}$ over a continuous manifold. Despite the differences, we found properties of SSC helpful in underpinning the identifiability in the nonlinear mixing case after translation to SDI in our case (see Eqs (21)-(33) in Appendix B, which are based on SSC-based matrix factorization work [48]). To be specific, the nice mathematical properties of the SSC, coupled with a volume-maximizing criterion that find a set of maximally spread-out Jacobian columns as guided by our intuition, gives us the theoretical tool to establish an identifiability proof.
>
> We further found the SDI condition being applicable in real-world tasks such as single-cell transcriptomics analysis (Section 5.2) and semantic disentanglement in images (Appendix E). We hope to experiment other real applications of DICA in future.
>
> ## Alternative constraint sets to the scaled L1 norm ball in SDI
>
> We thank the Reviewer for their suggestion. While the weighted L1 norm constraint in SDI yields an arguably interesting nonlinear model, as elaborated in "Physical Meaning of SDI" section and the experiments, we believe there are other useful alternative constraint sets for SDI. This is an interesting direction that deserves attention in future works.
>
> However, we note that not all polytopes considered in NMF/PMF literature would yield a meaningful/useful data model in a nonlinear mixture context. This is because the physical meaning of polytope constraints in SDI is very different from the physical meaning of polytope constraints in PMF/NMF, due to the two being in totally different domains (Jacobian of nonlinear function vs. mixing matrix).
>
> For example, in the case of NMF [9], the polytope is a probability simplex. In Jacobian domain, this means the L1 norms of gradients are bounded by 1, and the gradients are non-negative. The non-negativity of gradients imposed by the NMF polytope is quite restrictive and *the physical meaning of a nonlinear mixing function with non-negative gradients at every point is quite unclear*. In contrast, in the linear data model as in NMF/PMF, constraining the rows/columns of a mixing matrix inside probability simplex yields a sensible data model, which found many applications.
>
> Hence, one should consider carefully whether a proposed set constraining the Jacobian of nonlinear mixing function would yield meaningful physical meanings for the data generating process. Since the physical meaning of the polytope in SDI of DICA and in SSC of NMF/PMF are very different, not all constraint sets in NMF/PMF would yield a meaningful Jacobian constraint for a nonlinear mixture.

---

> > ### Comment · Reviewer_pCZF · 2025-08-03
> > **Thanks for your response.**
> >
> > I would like to thank the authors for their detailed response. I have read their replies to all reviewers' comments and revisited the paper in light of their clarifications. Considering both the authors' responses/revisions and the significance of the results, I am raising my score to 4.

---

> > > ### Author Response · Authors · 2025-08-06
> > >
> > > We would like to thank the Reviewer for the discussions on clarifying the connections to PMF/NMF. These are important points that will make the contributions of the paper more articulated, which we will incorporate into our manuscript.

---

### Official Review · Reviewer_Sbtv · 2025-07-03

**Clarity:** 2
**Significance:** 3
**Originality:** 4
**Rating:** 5
**Confidence:** 4

**Summary:**

The paper studies the problem of nonlinear mixture model identification (NMMI), i.e., how to provably recover d-dim. latent sources $s$ from m-dim. observed nonlinear mixtures $x=f(s)$ thereof. (If the sources are jointly independent, this problem is known as nonlinear ICA.) NMMI is hard in general, but has been addressed in prior work under additional assumptions such as auxiliary information or sparsity.

The paper instead proposes a different approach based on the assumption of sufficiently diverse influences (SDI). SDI is a non-trivial condition involving the convex geometry of the gradients $\nabla f_1, ..., \nabla f_m$. Intuitively, it states that globally, i.e., for all $s$:
- (i) for each source $s_j$ there exists an $x_i$ that is positively influenced by $s_j$ and an $x_k$ that is negatively influenced by $s_j$
- (ii) for each source $s_j$ there exists an $x_l$ such that $\nabla f_l$ is dominated (in magnitude) by its $j$th component $\partial x_l / \partial s_j$

Algorithmically, the SDI condition suggests an approach based on maximising the volume of the Jacobian $J_f$ subject to invertibility of $f$ (enforced via an autoencoder) and a bounded norm constraint on the rows of the Jacobian, i.e., the gradients $\nabla f_i$.

Theoretically, it is shown that SDI enables identification of the latent sources up to permutation and element-wise reparametrization, without requiring statistical independence or sparsity. For a finite sample setting, it is shown that, under additional conditions, the identification error can be bounded.

Empirically, the proposed DICA approach is compared against a vanilla autoencoder and a baseline with sparsity regularization on the Jacobian. On synthetic and semi-synthetic gene expression data, DICA is shown to outperform these baselines in terms of latent variable recovery.

**Questions:**

### Questions
- How did you arrive at or come up with the SDI assumption? Can it be derived or motivated "from first principles"?
- Thm. 3.2 establishes that J-VolMax under Asm. 3.1 yields identifiability, but is there a more direct connection between the SDI condition and volume maximisation?
- For fixed magnitude vectors, the volume of a parallelotope is maximal iff the vectors are orthogonal, as enforced in IMA [11]. This seems related to the J-VolMax objective, but there the norm constraint in Eq. (7b) appears to be on the gradients (i.e., rows of the Jacobian), rather than on the partial derivatives (i.e., columns of the Jacobian) as in IMA. Could you please comment further on the relationship between the two approaches?
- is the SDI condition invariant to reparametrisation of the sources? i.e., if SDI holds for $f$ and $f$ is replaced by $f\circ \rho$ for an element-wise function $\rho$, does SDI still hold?
- what is relation between the notion of "one source dominating the derivatives" entailed by SDI to that of anchor features/hard sparsity [e.g., 41]? can one think of the former as a relaxation of the latter?
- Since [4] appears to be a key reference and is used as a baseline, are the authors aware of [R1] and [R2] which relax the sparsity assumption on the Jacobian of [4]? You may want to consider these works and discuss their relation to SDI as well.

### Comments and Suggestions
- There appear to be some inconsistencies regarding $\mathcal{B}_\infty^{1/w(s)}$ vs. $\mathcal{B}_\infty^{w(s)}$ in Asm. 3.1., the caption of Fig. 1, and lines 125-126. Is it possible that there is a typo here?
- Eq. (7a) seems to miss a closing bracket.
- $\hat{\Pi}$ is undefined in eq. (9)
- Thm 3.3 mentions "three regularity conditions" but four are given.
- The claim that IMA [11] does not have identifiability underpinning appears a little too strong: while full identifiability for IMA has not been established (to the best of my knowledge), e.g., [5] proves local identifiability.
- Since optimization of the Jacobian is highlighted as a computational challenge, the authors may be interested in [R3].
- The authors may be interested in [R4] which remarks on some potential issues with some proofs of reference [50] in their manuscript.

### References
[R1] Lachapelle, Sébastien, et al. "Additive decoders for latent variables identification and cartesian-product extrapolation." Advances in Neural Information Processing Systems 36 (2023): 25112-25150.

[R2] Brady, Jack, et al. "Interaction Asymmetry: A General Principle for Learning Composable Abstractions." International Conference on Learning Representations (2025).

[R3] Gresele, Luigi, et al. "Relative gradient optimization of the jacobian term in unsupervised deep learning." Advances in neural information processing systems 33 (2020): 16567-16578.

[R4] Buchholz, Simon. "Some remarks on identifiability of independent component analysis in restricted function classes." Transactions on Machine Learning Research (2023).

**Ethical Concerns:**

["NO or VERY MINOR ethics concerns only"]

**Final Justification:**

My initial concerns and questions regarding
- the intuition behind the SDI assumption and the motivation behind volume maximisation
- the relation of the proposed method to existing literature on nonlinear blind source separation
- limitations of the method, in particular the requirement for a large number of measurements compared to sources

have been satisfactorily addressed by the authors during the discussion, and the authors have agreed to include these clarifications in the updated manuscript.

I believe that this work is sound and a valuable addition to the literature on identifiable nonlinear representation learning.

After reading the other reviews, it seems that it was rightfully raised that the authors did not sufficiently credit a prior work in inspiring their approach, but that they have promised to rectify this. I do not consider computation issues in Jacobian computation (mentioned as a key weakness by another reviewer) sufficient grounds for rejection, seeing as this is transparently stated.

Therefore, I recommend that this paper be accepted.

**Limitations:**

yes

**Paper Formatting Concerns:**

No major concerns (minor vspacing may have been used after figure and table captions)

**Quality:**

3

**Strengths And Weaknesses:**

### Strengths
- the paper is well written and (apart from the mathematical complexity) easy to read
- the proposed approach is original and novel
- the paper provides contributions along several axes: a new principle, identifiability results, an estimation methods, and experiments
- the finite sample analysis (which is not common in the identifiability literature) is a nice addition
- the paper does a good job at covering related work and discussing its relations to the proposed approach
- the paper does not just consider sythetic data, but also provides results on a semi-synthetic dataset from a biological simulator

### Weaknesses
- the SDI assumption is quite difficult to understand: while the "Physical Meaning of SDI" paragraph is helpful to gain some intuition and interpret the condition, it is unclear how the authors came up with or arrived at the precise mathematical formulation in Assumption 3.1; it does not help that definitions and explanations of key concepts (convex hull, polar set, MVIE) are deferred to the appendix (these should be moved to the main paper);
- the precise relation between the SDI assumption and the Jacobian volume maximisation (J-VolMax) approach, as well as that between SDI and orthogonality maximization of the Jacobian columns (IMA) remain a little unclear;
- it seems that the proposed approach requires a very large number of observations relative to the number of latent sources (in section 3, it is stated that $m\geq 2d$ is required, but in the experiments it is much larger, $m\gg d$)
- the empirical evaluation is somewhat limited in that it only considers relatively naive baselines; comparison with other nonlinear ICA methods (e.g., IMA or based on sparsity) and reducing the number of observations $m$ (resp. increasing the number of sources) could be useful additions to get a better understanding of empirical performance
- quantifiers (for all, there exists) are used too sparingly and inconsistently; including these for all formal statements would improve clarity

---

> ### Author Rebuttal · Authors · 2025-07-31
>
> ## Motivations behind the SDI assumption
>
> The motivation of DICA is to exploit the pattern of how sources $\mathbf{s}$ influence the observations $\mathbf{x}$ via the geometry structure of the mixing Jacobian matrix, in a manner similar to sparsity-based NMMI and the IMA. In particular, while the Principles of Independent Causal Mechanisms [11, 45],
>
> > *“The causal generative process of a system’s variables is composed of autonomous modules that do not inform or influence each other.”*
>
> inspired the orthogonality of columns of the Jacobian matrix as in the IMA, we took a different view. Instead of assuming influence mechanisms being totally independent of each other, we assumed that
>
> > ***the ways that distinct sources $s_1, …, s_d$ influence the observations are sufficiently different***
>
> in the sense that the columns of Jacobian matrices spanned in different directions. **Therefore, the change to observations** $\mathbf{x}$ **by varying** $s_i$ **should be sufficiently different from that by varying** $s_j$. One can also think of SDI conditions as a relaxed version inspired by IMA, which describes geometrically the spread of Jacobian columns in space, without strict orthogonality.
>
> To quantitatively pin down this intuition to form the exact SDI condition, we follow the idea in the matrix factorization literature [9, 17, 18, 48] (particularly [48]), where sufficiently scattered conditions (SSCs) are often used to prove identifiability of linear mixture models. SSC describes how column/row vectors of a latent factor matrix are scattered inside a convex polytope. Instead, we describe the scattering pattern of mixing Jacobian matrices at every point $\mathbf{s}$ over a continuous manifold. Despite the differences, we found properties of SSC helpful in underpinning the identifiability in the nonlinear mixing case after translation to SDI in our case (see Eqs (21)-(33) in Appendix B, which are based on SSC-based matrix factorization work [48]). To be specific, the nice mathematical properties of the SSC, coupled with a volume-maximizing criterion that find a set of maximally spread-out Jacobian columns as guided by our intuition, gives us the theoretical tool to establish an identifiability proof.
>
> We will include more explanations on SDI along with related convex geometric notions in our updated main paper.
>
> ## Connection between SDI and Jacobian volume maximization
>
> Here we briefly explain the geometric intuition behind Jacobian volume-maximizing learning criterion. It is a generalization of geometric intuition of volume max/min based matrix factorization (see [48] and related works, e.g., [17, 9, 18]).
>
> Despite the technical definition of SDI being described in terms of the gradients $\nabla_{\mathbf{s}} x_i$, the implication is more clear in the partial derivative domain, as described in “Physical Meaning of SDI” section. The SDI condition at a point $\mathbf{x} = \mathbf{f}(\mathbf{s})$ implies that there exists some $x_i$ that $s_j$ dominantly regulates $x_i$, so $|\partial x_i / \partial s_j| \gg |\partial x_i / \partial s_k|$ for $k \neq j$. For the two partial derivative vectors $\partial \mathbf{x} / \partial s_j, \partial \mathbf{x} / \partial s_k \in \mathbb{R}^{m}$, this means that the two partial derivative vectors are spread out and point in quite different directions.
>
> Maximizing the Jacobian determinant as in the DICA objective would indeed encourage the partial derivative vectors to spread in different directions, aligning with the SDI assumption.
>
> ## Relationship between IMA and DICA
>
> As mentioned in the response to the Reviewer’s first question, both the IMA and the DICA framework seek to tackle the NMMI task via exploiting the geometric structure of mixing Jacobian, which describes how the sources $\mathbf{s}$ influence the observations $\mathbf{x}$. In particular, the learning criterions in both IMA and DICA include a Jacobian determinant term, which measures how spread-out the partial derivatives are.
>
> The difference between IMA and DICA lies in how the two data models assume the geometric structure of mixing Jacobian geometric structure, which dictates the goals of their respective learning criterions:
> * In IMA, it is the orthogonality of partial derivatives; hence, **IMA objective looks for a decoder with orthogonal partial derivatives, via maximizing the (normalized) Jacobian determinant**
> * In DICA framework, the SDI data model instead assumes a specific spread-out pattern of gradients in a $\ell_1$ norm ball, which entails that the partial derivatives are spread out in space (see the second reply as well as “Physical Meaning of SDI” section); hence, **DICA objective looks for a decoder with partial derivatives maximally spread out via maximizing Jacobian determinant** ***inside the constraint set from SDI***.
>
> The shared ground that DICA and IMA is indeed very intriguing, and we hope that the identifiability results presented in DICA can help shed light on a global identifiability analysis of IMA, in addition to its local identifiability property [5] as mentioned by the Reviewer.
>
> ## Invariance of SDI to reparametrization of sources
>
> Thank you for raising this point.
>
> If the sources $\mathbf{s}$ is reparametrized by an *invertible* element-wise transformation $\rho_i$, the new Jacobian function of $\mathbf{f}(\boldsymbol{\rho}(\mathbf{s}))$ is $J_{\mathbf{f}}(\boldsymbol{\rho}(\mathbf{s}))J_{\boldsymbol{\rho}}(\mathbf{s})$, where $J_{\boldsymbol{\rho}}(\mathbf{s}) = \text{Diag}(\frac{d \rho_1(s_1)}{d s_1}, …, \frac{d \rho_d(s_d)}{d s_d})$ is a diagonal matrix. Hence, the gradients $ \nabla_{\mathbf{s}} x_i $ are scaled by $J_{\boldsymbol{\rho}}(\mathbf{s})$. By Proposition A.1., the MVIE and the weighted $\ell_1$ norm ball $B_1$ containing the gradients are rescaled accordingly by $J_{\boldsymbol{\rho}}(\mathbf{s})$, and hence the SDI still holds (albeit with a different $B_1$). A similar argument was used in the proof of Theorem 3.1, particularly (18)-(21).
>
> As a result, the SDI is invariant to *invertible* element-wise transformation of sources. For *non-invertible* element-wise reparametrization, it is unclear.
>
> ## Relation between SDI and anchor features/hard sparsity
>
> The sparse influence structure such as anchor features (e.g., [41]) or hard Jacobian sparsity (e.g., [50, 51]) indeed have some overlap with SDI; for example, in the base case m = 2d of SDI.
>
> In fact, for the SDI cases of $|\partial x_i / \partial s_j| \gg |\partial x_i / \partial s_k| \approx 0$ where the non-dominant source $s_k$ has approximately absent influence, one can interpret these cases as a relaxation of the sparse influence. However, we note that the SDI also includes the cases where $|\partial x_i / \partial s_j| \gg|\partial x_i / \partial s_k| \gg 0$, where the non-dominant sources $s_k$ have an influence greater than 0. This departs from the sparse influence structure, which demands (approximately) zero entries in Jacobian matrices.
>
> Therefore, while some cases of SDI overlap with Jacobian sparsity, in general SDI is not a sparsity-based condition.
>
> ## Relation between [R1], [R2] and SDI
>
> We thank the reviewer for pointing out the two related works [R1] and [R2]. To our understanding, the focus of [R1], [R2] as well as [4] are on object-centric settings, which aims to recover disentangled representations of different objects in the presented scene. Therefore, [R1], [R2] and [4] mainly focus on identifying *blocks of latent variables* that represent objects. The goal of DICA aligns more closely with that of nonlinear ICA, which aims to recover the *individual latent variables* from an observed mixture.
>
> An exciting direction is to examine how the learning principle in DICA can contribute to object-centric representation learning, and we will discuss the relation between DICA and [R1], [R2] in future works.
>
> ## On number of observations required relative to number of sources
>
> Indeed, the SDI condition requires at least $m \geq 2d$, which is more than other nonlinear ICA methods (which typically requires $m = d$ or $m \geq d$ only). In practice, DICA favors $m \gg d$ to attain the SDI; see the “Discussions” in Section 3.1 and references therein for further details.
>
> However, high-dimensional data is prevalent and often lies inside a low-dimensional latent manifold. Hence, the SDI condition is still very applicable in real-world settings such as computer vision or single-cell biology. For example, we refer the Reviewer to Section E on a preliminary experiment on MNIST, where we showcase how DICA with a vanilla CNN-based autoencoder can discover some latent components that control the semantic digit of output.
>
> ## Experiments with IMA and reduced number of observations
>
> As the Reviewer suggested, we compare DICA with IMA. We also reduce the ratio of number of observations $m$ over number of sources $d$. For IMA, we use the local IMA contrast in [Def. 3.1, A6] as a regularizer, plus the autoencoder loss. We use a setting similar to Mixture C for testing. The nonlinear R2 scores are in the table below.
>
> |(d,m)|DICA|IMA|Sparse|Base|
> |---|---|---|---|---|
> |(3,40) | $0.90 \pm 0.10$ | $0.63 \pm 0.15$ | $0.80 \pm 0.13$ | $0.63 \pm 0.10$|
> |(3,30)| $0.89 \pm 0.07$ | $0.63 \pm 0.12$| $0.77 \pm 0.12$ | $0.63 \pm 0.14$|
> |(3,20)| $0.82 \pm 0.17$|$0.60 \pm 0.12$|$0.74 \pm 0.13$|$0.60 \pm 0.10$|
>
> * Since the setting of SDI-satisfying Mixture C does not favor the IMA assumption, i.e. orthogonal Jacobian, the IMA regularizer did not perform as well as DICA in this experiment.
>
> * Furthermore, the results show that while DICA requires a relatively large number of observations $m$ as predicted in the theory, the performance decays but is still acceptable when $m$ reduces towards $m = 2d$.
>
> *References*
>
> [A6] Ghosh et al., “Independent Mechanism Analysis and the Manifold Hypothesis”, Causal Representation Learning Workshop at NeuRIPS, 2023.

---

> ### Comment · Reviewer_Sbtv · 2025-08-04
>
> I thank the authors for their response.
>
> The clarifications in the first 5 five points are helpful and provide additional insights. I encourage the authors to incorporate some of these remarks into the updated manuscript.
>
> Regarding the connection to [R1] and [R2]: I would like to add that, while these works indeed focus mostly on recovering blocks of variables in the object-centric setting, they also have implications for the case of recovering singleton latents, i.e., with block size 1. The corresponding relaxations of sparsity in this case could still be interesting to compare to.
>
> Regarding the dimensionality of observations and the additional experiment: I appreciate the additional results and it is nice to see DICA perform very well compared to other methods when $m>>d$. However, an important ingredient of solid scientific work is to also showcase limitations and situations where the proposed methods fail. I therefore feel strongly that it is important to also include results for, e.g., $(d,m)=(3,10)$ and $(3,6)$ where DICA presumably stops working (well). It would also be interesting to look at $(d,m)=(5,10), (5,20)$ or $(10,20), (10,40)$ to get a better understanding of the relative size of $m$ and $d$ for DICA to be competitive.
>
> In practice, what happens when $m<2d$, i.e., the assumption is certainly violated. Will the procedure still run and produce an output?

---

> > ### Author Response · Authors · 2025-08-05
> >
> > We thank the Reviewer for the productive discussion. Please find our further comments below.
> >
> > ## Regarding the clarifications of DICA
> >
> > We are glad that the Reviewer finds the insights in our response to be helpful, and we will include these remarks in our manuscript.
> >
> > ## On connection between [R1], [R2], [4] and DICA
> >
> > We thank the Reviewer for their insight on a relationship between block-wise identifiability in object-centric learning and component-wise identifiability in DICA, when block size is $1$. Indeed, the connection is intriguing: in object-centric learning with sparse Jacobian [4], reducing the block size to $1$ would indeed correspond to the case of “extremely” sparse influence that are SDI-satisfying, i.e. when the gradients are some scaled basis vectors $\{\pm c_1 \mathbf{e}_1, …, \pm c_d\mathbf{e}_d\}$ (for $c_1,...,c_d > 0$). **We will thoroughly discuss the relationship between [R1], [R2], and [4], particularly the case when block size is $1$, in our updated manuscript.**
> >
> > ## Additional experiments with reduced $m$
> >
> > We appreciate the suggestion from the Reviewer. We will incorporate a more comprehensive ablation study of the number of latent dimensions vs. number of observations in our manuscript, so one can better understand when DICA succeeds and fails. Here, we provide preliminary experiments on Mixture C for $(d, m) = (3, 10), (3, 6), (3, 3)$ as well as $(d, m) = (5, 20), (5, 10), (5, 5)$ in the two following tables.
> >
> > | $(d, m)$ | DICA | IMA | Sparse | Base
> > |---|---|---|---|---|
> > | $(3, 10)$ | $0.71 \pm 0.17$ | $0.56 \pm 0.09$ | $0.75 \pm 0.12$ | $0.63 \pm ​​0.16$
> > | $(3, 5)$ | $0.64 \pm 0.09$ | $0.66 \pm 0.16$ | $0.78 \pm 0.16$ | $0.61 \pm 0.08$
> > | $(3, 3)$ | $0.55 \pm 0.07$ | $0.60 \pm 0.09$ | $0.58 \pm 0.16$ | $0.51 \pm 0.09$
> >
> > | $(d, m)$ | DICA | IMA | Sparse | Base
> > |---|---|---|---|---|
> > | $(5, 20)$ | $0.60 \pm 0.12$ | $0.53 \pm 0.07$ | $0.62 \pm 0.11$ | $0.54 \pm 0.07$
> > | $(5, 10)$ | $0.58 \pm 0.10$ | $0.49 \pm 0.07$ | $0.59 \pm 0.11$ | $0.47 \pm 0.09$
> > | $(5, 5)$ | $0.49 \pm 0.11$ | $0.44 \pm 0.07$ | $0.48 \pm 0.06$ | $0.43 \pm 0.09$
> >
> > We observe that the performance of DICA decreases towards that of the vanilla autoencoder baseline, when the ratio $m / d$ becomes smaller. Specifically, DICA performs significantly worse when $m$ approaches the limit $2d$. This confirms our theory that DICA works best when the number of observations is sufficiently large, as in high-dimensional datasets.
> >
> > ## Regarding what happens when $m < 2d$
> >
> > DICA will still output an estimation of the latent variables, albeit there’s no guarantee that those estimates match the ground-truth up to acceptable ambiguities. As we can see from the tables, for the $d = m = 3$ and $d = m = 5$ cases, DICA performs poorly and only as good as vanilla autoencoders.

---

> > > ### Comment · Reviewer_Sbtv · 2025-08-05
> > >
> > > I thank the authors for their continued engagement and further responses, which have satisfactorily addressed my concerns.
> > >
> > > I will increase my rating to 5 and will recommend acceptance, in the understanding that the authors will include:
> > > - additional intuition behind the SDI assumption and its relation to Jacobian volume maximisation
> > > - updated contextualisation w.r.t. prior work, in particular ref. [48] (as highlighted by reviewer `pCZF`) and the relation between DICA and {IMA, anchor features, object-centric learning (w. block size 1)}
> > > - additional experiments for $m\leq 2d$ and an honest and *prominent* (i.e., not buried in the appendix) discussion of the limitations of DICA, i.e., that it is designed for high-dim. settings and fails when the number of measurements is too small
> > >
> > > in the revised version of the manuscript, as discussed during the rebuttal period.

---

> > > > ### Author Response · Authors · 2025-08-06
> > > >
> > > > We thank the Reviewer for the constructive feedback. We will make sure to include all the points as discussed into our manuscript.

---

### Official Review · Reviewer_8yNX · 2025-07-14

**Clarity:** 2
**Significance:** 2
**Originality:** 3
**Rating:** 4
**Confidence:** 3

**Summary:**

The authors propose an alternative way to tackle the nonlinear mixture identifiability problem. The setting is as follows: unknown latent components are transformed by an unknown nonlinear function into observation and the challenge is to recover the unknown latent components from the observations. In general, the problem might be ill-posed, therefore most work are restricted to scenarios where identifiability can be shown. One way to do so is to make stronger assumptions on the nonlinear transformation, e.g. by imposing sparsity on the Jacobian matrix of the transformation. Instead, the authors build a geometric argument as to why it is sufficient to penalize the Jacobian log determinant. This allows them to treat functions with dense jacobians as well.

**Questions:**

- the proposed approach employs an unconstrained neural network and forces its inverse to exist with a loss that acts on training points. While this approach has been previously used in other contexts as well, it only requires the learnt map to be invertible locally. Especially in high-dimensional spaces, this doesn't enforce invertibility in any systematic way. This might not be visible in the experiments since a very small network is used (1 hidden layer with 64 units) in a very low-dimensional space
- Maybe I am missing something, but have you considered defining f as invertible by construction like normalizing flows? this would also make the proposed method computationally much more efficient when it comes to computing the Jacobian determinant (from cubic to quadratic/linear)
- If this is a viable approach, one could easily [1] even constrain the function to lie on a L1 norm ball by construction (in place of the regularization term of your approach)
- I have a general doubt concerning the experiment section. In the synthetic experiment the nonlinearity is chosen at random so I expect its jacobian not to be sparse (which is also a benefit of the proposed approach). On the other hand, for the single-cell experiment, do we know something about the degree of sparsity of the Jacobian of the "true" f to be learnt?
- If the underlying Jacobian is however sparse, should the method Sparse provide better results than the one obtained? Did you also try with higher sparsity? (I saw in the appendix you selected the best parameter via cross-validation)
- In case both experiment have a dense/non-sparse Jacobian, how does your method compare to current approaches that assume some form of sparsity?

[1] Negri et al., Injective flows for star-like manifolds, ICLR 2025

**Ethical Concerns:**

["NO or VERY MINOR ethics concerns only"]

**Final Justification:**

- I believe the work provides interesting theoretical insight to the mixture identifiability problem
- experimental results are encouraging even though not particularly extensive
- I remain a bit concerned about the true invertibility of the learnt function (which is invertible only on the training set) and about the Jacobian determinant computations, which are extremely expensive (if computed exactly) or only approximate (if an estimator is used)

**Limitations:**

Limitations are clearly stated and discussed in the conclusions. The main limitation is that in practice it is really expensive to compute the Jacobian log-determinant at every iteration, which makes the method not viable in practice.

**Paper Formatting Concerns:**

I haven't noticed any formatting issues.

**Quality:**

2

**Strengths And Weaknesses:**

Strengths:
- the mathematical foundation seems solid and the solution proposed is elegant

Weaknesses:
- even though it is left for future work, the proposed approach is not viable in practice because training requires computing the Jacobian determinant, which is cubic if the function is unconstrained (see questions below)

---

> ### Author Rebuttal · Authors · 2025-07-31
>
> ## Efficient computation of Jacobian log-determinant
>
> Note that our focus was on the identifiability analysis side, and thus we did not delve deep into the implementation side. **In practice, multiple methods can be used to avoid the cubic complexity**. Here we provide two options:
>
> 1. For example, one can use a trace-based formula as a surrogate [A1] (instead of using determinant) for volume optimization, **which would reduce the computational complexity to quadratic**. As a quick proof-of-concept test, we replace the log-det term with a trace-based surrogate in [A1] in our synthetic experiment similar to Mixture C with $d = 3$ and $m = 40$. We report the nonlinear R2 scores obtained from each method, as well as the average wall-clock time for gradient computation per epoch; as we can see, **the gradient computing time for this trace-based DICA is much lower than logdet-based DICA** while having similar performance in R2 score:
>
> | | Logdet-based DICA | Trace-based DICA | Sparse | Base |
> |---|---|---|---|---|
> | R2 Score | $0.94 \pm 0.08$ | $\boldsymbol{0.91} \pm 0.07$ | $0.79 \pm 0.12$ | $0.63 \pm 0.04$ |
> | Wall-clock Time per Epoch (ms) | $31.31 \pm 0.30$ | $\boldsymbol{24.89} \pm 0.26$ | $18.95 \pm 0.18$ | $6.90 \pm 0.18$ |
>
> 2. In addition, as pointed out by Reviewer Sbtv, there are techniques in the unsupervised learning literature [A2] that can efficiently deal with Jacobian determinant optimization. **The per-iteration complexity of this method is quadratic instead of cubic**. We leave the details for future follow-up works.
>
> We will add discussions on the computational aspects in the conclusion section.
>
> *References*
>
> [A1] Fu et al., “Robust Volume Minimization-Based Matrix Factorization for Remote Sensing and Document Clustering”, IEEE Transactions on Signal Processing, 2016.
>
> [A2] Gresele et al. "Relative gradient optimization of the Jacobian term in unsupervised deep learning" NeuRIPS, 2020.
>
> ## On inverting neural networks via autoencoder loss
>
> In practice, as pointed out by the reviewer, when the mixing function is not too complicated and there are enough data points, such an autoencoding loss in DICA objective can learn to approximately invert the decoder. However, when the mixing function is much more complicated or the dimensions are high, learning to invert the decoder might be much more complicated. Hence, the vanilla autoencoder presented in the manuscript is limited in this setting, which calls for more sophisticated models.
>
> Nonetheless, our focus is to demonstrate a learning principle for identifiable nonlinear mixture learning, and the vanilla autoencoder merely serves as proof-of-concept. We leave the investigations for implementing DICA learning criterion under more complex settings in follow-up works.
>
> ## On the use of normalizing flow in DICA framework
>
> We thank the reviewer for suggesting normalizing flow (NF), which is invertible by construction, as a possible alternative for the vanilla autoencoder. However, as far as we know, most of the current NF models require the latent dimension $d$ being equal to the number of observations $m$, whereas our SDI condition requires $m \geq 2d$.
>
> That being said, NF is a promising tool for efficiently implementing DICA framework, which we leave for further investigation in the future.
>
> ## On the single-cell transcriptomics experiment
>
> The purpose of our experiment on single-cell transcriptomics data is to test the performance of DICA in a setting where we have *no control of the mixing Jacobian* (which would show the usefulness of this framework in practice). To the best of our knowledge, a full ground-truth gene regulatory mechanism is yet available, and the transcription factor - gene expression interactions are only partially available. Therefore, we don’t know exactly how sparse gene regulatory mechanisms are, as there might be crosstalks (one transcription factor interacts with many genes) and various transcription noises [A3] such that the mixing Jacobian is dense/non-sparse. DICA can be considered as an addition to the data scientist’s toolkit for studying gene regulation in the case of non-sparse regulatory systems, on top of existing sparsity-based tools.
>
> *References*
>
> [A3] Tamar Friedlander et al. “Intrinsic limits to gene regulation by global crosstalk”. Nature
> Communications, 2016.
>
> ## Performance of DICA when the mixing Jacobian is sparse
>
> *Following the reviewer’s suggestion, we tried with higher sparsity of the Jacobian*. We used a setting similar to Mixture C in the paper. The nonlinear R2 scores are shown in the table below. One can see that both DICA and Sparse work reasonably well. We hope to remark that DICA’s identifiability condition covers some cases where the Jacobian is sparse, so it is not surprising that DICA could work well under higher sparsity levels. However, DICA is not proposed to outperform sparsity-based methods when the Jacobian is sparse, but as a complement to tackle more cases where the Jacobian is not sparse.
>
> | $(d, m)$ | DICA | Sparse | Base |
> |---|---|---|---|
> | $(3, 40)$ | $0.88 \pm 0.16$ | $0.85 \pm 0.07$ | $0.68 \pm 0.14$ |
> | $(4, 50)$ | $0.85 \pm 0.80$ | $0.80 \pm 0.12$ | $0.58 \pm 0.07$ |
>
> ## Performance of DICA vs. sparsity-based approaches, under dense/non-sparse mixing Jacobian
>
> We refer the reviewer to our experiments (in Section 5) as well as the table in our reply to your first question. We note that both the mixtures in the synthetic simulations and the single-cell transcriptomics experiment have Jacobian matrices that are (highly likely) non-sparse and SDI-satisfying. DICA outperforms Sparse, another approach that only exploits the Jacobian sparsity structure via a Jacobian L1 norm regularizer. These experiment results confirm the advantage of our proposed method in non-sparse settings.

---

> > ### Comment · Reviewer_8yNX · 2025-08-04
> >
> > I thank the authors for the detailed answers and for the additional experiments.
> >
> > I agree that the contribution is more on a theoretical side, but in order for it to be relevant I believe it is important that it is actionable. One could indeed approximate the Jacobian log determinant with Hutchinson's trace estimator  if the exact values are not needed. However, one would still require $f$ and $g$ to be invertible - and as the authors acknowledge, this it quite difficult in practice (independently of whether one uses an autoencoder or something else). Normalizing flows, on the other hand, would solve this problems at the expense of not having a universal function approximator (for functions, not densities). I believe that these shortcomings should be clearly acknowledged and discussed (not necessarily solved, since I believe there is no simple solution) and not just left as future work.
> >
> > Concerning the use of Normalizing Flows, I agree that the latent dimension must be left the same. Maybe an interesting reference can be found in [1], where the authors propose a similar approach to the one in the current paper. They approximate the Jacobian determinant with the trace estimator and learn an approximate inverse. I still believe that this approach wouldn't solve the shortcomings mentioned in the previous paragraph.
> >
> > > Following the reviewer’s suggestion, we tried with higher sparsity of the Jacobian. We used a setting similar to Mixture C in the paper.
> >
> > I am not sure I fully understood: Do you consider Mixture C to be a sparse settings? I would have expected a randomly initialized neural network to have a dense Jacobian.
> >
> > [1] Draxler et al., Free-form Flows: Make Any Architecture a Normalizing Flow, AISTATS 2024

---

> > > ### Author Response · Authors · 2025-08-04
> > >
> > > Thanks for this constructive discussion. We are glad that the Reviewer is familiar with Hutchinson's trace estimator and agrees that the trace-based volume estimator could work for large scale problems. We will also include the new reference as supporting literature for the estimator.
> > >
> > > ## Regarding invertibility
> > >
> > > Following the discussion and the suggestion, we will clearly articulate the limitations in our next version. Specifically, we will discuss the hardness to enforce exact invertibility of $\mathbf{f}$ and $\mathbf{g}$ using limited data. In addition, we will discuss limitations of several popular options for this task, e.g., stacked autoencoder (AE) and normalizing flows.
> > >
> > > Here’s our drafted version:
> > >
> > > > We remark on some limitations of DICA, particularly the challenge of enforcing $\mathbf{f}$ and $\mathbf{g}$ to be invertible. (a) One common approach is to use AE regularization, as in the DICA training loss in Section 5, which promotes invertibility under the assumptions of infinite samples and universal function approximators; see [Theorem 2, 4] and [Section V, A12]. This strategy is widely used in the literature, including classical deep clustering [A8], deep CCA [A9], CycleGAN [A7], and object-centric learning [4], as well as generative and foundation models like VQ-VAE [A10] and Latent Diffusion [A11]. Thus, stacked AE-type regularization—as used in DICA—is broadly applicable to large-scale data. That said, designing practical $\mathbf{f}$ and $\mathbf{g}$ architectures remains challenging due to the lack of true universal approximators, and invertibility is only approximate with finite samples. (b) Another way to ensure invertibility is through normalizing flows (NF), which enforce it by construction and are appealing for their connection to determinant computation. However, NF has two drawbacks in our setting: it requires the latent and data dimensions to match, whereas DICA assumes $m \geq d$; and it typically sacrifices universality, limiting the expressiveness of $\mathbf{f}$. These limitations pose interesting directions for future research.
> > >
> > > ## Regarding the experiment on sparse Jacobian
> > >
> > > We control the sparsity of the Jacobian by controlling $\partial x_i / \partial s_j$. When we generate the observations $x_i$ for $i=1,... m$, half of observations $x_i$ are generated from $d$ sources using a randomly initialized neural network: one source $s_k$ sampled from $U(-1, 1)$, and the other $d-1$ “suppressed” sources are set to $0$ (in the original Mixture C, these values were not exactly zero; here we force them to be zero). Hence, the observations are not influenced by the “suppressed” sources, so the corresponding Jacobian entries are zero, i.e. $\partial x_i / \partial s_j = 0$ for $j \neq k$. **The mixing function in this experiment would then have a sparse Jacobian.**
> > >
> > > *References*
> > >
> > > [A7] Zhu et al., "Unpaired Image-to-Image Translation Using Cycle-Consistent Adversarial Networks," ICCV, 2017.
> > >
> > > [A8] Yang et al., “Towards K-means-friendly Spaces: Simultaneous Deep Learning and Clustering”, ICML, 2017.
> > >
> > > [A9] Wang et al., “On Deep Multi-View Representation Learning”, ICML, 2015.
> > >
> > > [A10] van den Oord et al., “Neural Discrete Representation Learning”, NeuRIPS, 2017.
> > >
> > > [A11] Rombach et al., “High-Resolution Image Synthesis with Latent Diffusion Models”, CVPR, 2022.
> > >
> > > [A12] Lyu et al., “Identifiability-Guaranteed Simplex-Structured Post-Nonlinear Mixture Learning via Autoencoder”, IEEE Transactions on Signal Processing, 2021.

---

> > > > ### Comment · Reviewer_8yNX · 2025-08-05
> > > >
> > > > I would like to thank the authors for continuing the discussion. I still believe invertibility/Jacobian computation to be a significant limitation in practice, but I otherwise consider most of my concerns resolved. I will update my score accordingly.

---

> > > > > ### Author Response · Authors · 2025-08-06
> > > > >
> > > > > We would like to thank the Reviewer for the constructive discussions, and for considering updating the score.

---

### Decision · Program_Chairs · 2025-09-17

**Decision:**

Accept (poster)

**Comment:**

This paper proposes a novel approach to identifiability in  nonlinear source separation. Initially, the reviewers had a number of concerns. These included: intuition and motivation of the SDI assumption; invertibility of the learnt function; cubic computational cost of the Jacobian determinant; and relationship with the existing nonlinear identifiability literature.

Overall, all reviewers were satisfied with the author responses during the rebuttal period and raised their scores. For a final version of the manuscript, it will be important for the authors to incorporate the promised changes into the main paper.